# RECORD, a high-throughput, customizable system that unveils behavioral strategies leveraged by rodents during foraging-like decision-making
Raquel J. Ibáñez Alcalá [1,8], Dirk W. Beck[2,8], Alexis A. Salcido [1,8], Luis D. Davila[2,8], Atanu Giri[2,8], Cory N. Heaton [1,8], Kryssia Villarreal Rodriguez[1,8], Lara I. Rakocevic[2,8], Safa B. Hossain [1,8], Neftali F. Reyes [1,8], Serina A. Batson [1,8], Andrea Y. Macias[1,8], Sabrina M. Drammis[3,8], Kenichiro Negishi [4,8], Qingyang Zhang[5,8], Shreeya Umashankar Beck[1,8], Paulina Vara [1,8], Arnav Joshi[2], Austin J. Franco[1], Bianca J. Hernandez Carbajal[1], Miguel M. Ordonez[1], Felix Y. Ramirez[1], Jonathan D. Lopez[1], Nayeli Lozano[1], Abigail Ramirez[1], Linnete Legaspy[1], Paulina L. Cruz[1], Abril A. Armenta[1], Stephanie N. Viel[1], Jessica I. Aguirre[1], Odalys Quintanar[1], Fernanda Medina[1], Pablo M. Ordonez[1], Alfonzo E. Munoz[1], Gustavo E. Martínez Gaudier[1], Gabriela M. Naime[1], Rosalie E. Powers[1], Laura E. O'Dell[6], Travis M. Moschak[1], Ki A. Goosens [7] ✉ & Alexander Friedman [1,2] ✉

Translational studies benefit from experimental designs where laboratory organisms use human-relevant behaviors. One such behavior is decision-making, however studying complex decision-making in rodents is labor-intensive and typically restricted to two levels of cost/reward. We design a fully automated, inexpensive, high-throughput framework to study decision-making across multiple levels of rewards and costs: the REward-COst in Rodent Decision-making (RECORD) system. RECORD integrates three components: 1) 3D-printed arenas, 2) custom electronic hardware, and 3) software. We validated four behavioral protocols without employing any food or water restriction, highlighting the versatility of our system. RECORD data exposes heterogeneity in decision-making both within and across individuals that is quantifiably constrained. Using oxycodone self-administration and alcohol-consumption as test cases, we reveal how analytic approaches that incorporate behavioral heterogeneity are sensitive to detecting perturbations in decision-making. RECORD is a powerful approach to studying decision-making in rodents, with features that facilitate translational studies of decision-making in psychiatric disorders.

Studies of decision-making can quantify and parametrize otherwise nebulous concepts, such as cognition[1], subjective value[2], and help identify the biological correlates of decision-making related processes[3–7], thus many methods have been developed and employed to study decision-making in rodents (Supplemental Table 1). One example is the T-maze. T-mazes examine decision-making by offering a subject two options in branching arms at the end of the maze[3,5,6,8]. T-mazes are also used with virtual reality systems, allowing for two-photon calcium imaging during task performance[7]. Another common method used for studying decision-making in rodents is operant conditioning tasks[9–12]. Operant conditioning tasks have subjects perform actions (e.g., nose-pokes[13,14] or lever presses[15]) in response to cues. Rodent versions of the Iowa Gambling task are also used to

explore decision-making, since it mimics making decisions in uncertain conditions (typically by providing multiple options that have different probabilities of reward/cost being dispensed, depending on the magnitude of predictive stimuli), a common occurrence in day-to-day life[16].

These methods are excellent tools for studying different facets of decision-making, but we sought to provide a robust, easy-to-adopt system that increases the range of decision-making research. For example, decision-making tasks typically examine one or two levels of reward/cost[17–19]; so a system that enables the implementation of a broader range of reward/cost trade-offs within a single session with high experimental control would extend decision-making research. A range of trade-offs enhances granularity, facilitating the detection of individual differences, and exploration of decision-making phenotypes observed in psychiatric disorders[4,20]. Often decision-making research is interested in questions that involve large populations of animals, tracking behavior over task performance, and exploring different decision-making contexts[3,19,21]. We designed an automatic system[22] that provides these features within a single task-environment that is adaptable to many research spaces and experiments. Finally, we wanted our system to mimic ethologically relevant behaviors[23] which aids interpretation of neuronal recordings.

One frequent component of published decision-making tasks is the use of food[5,11,13,18,19,24] or water-restriction[7,9,12,17,21] to train/motivate rodents to perform the behavioral protocols. However, chronic deprivation may impact decision-making/behavior because it is a stressor that alters circulating hormones, blood glucose, and other physiological measures involved in appetitive processing[25]. Changes in these measures potentially alter neural activity or modify the activated neural circuits, thus impacting task performance and learning[26]. Tasks that function without requiring deprivation avoid these confounding variables and may better model human decision-making.

We aspired to design a decision-making system that: (1) is sensitive to individual variability in decision-making, (2) is adaptable to many experimental designs, (3) offers rodents multiple levels of cost and reward combinations, (4) and facilitates the study of biological mechanisms critical for decision-making. To accomplish these important objectives, we developed the Reward-Cost Rodent Decision-making (RECORD) system.

The RECORD system is a combination of five major components: 3D-printed parts, electronics, software (Fig. 1), behavioral protocols (Fig. 2), and modeling (Fig. 3). RECORD allows the identification of individual decision-making strategies (Fig. 4) and does not require food or water deprivation to

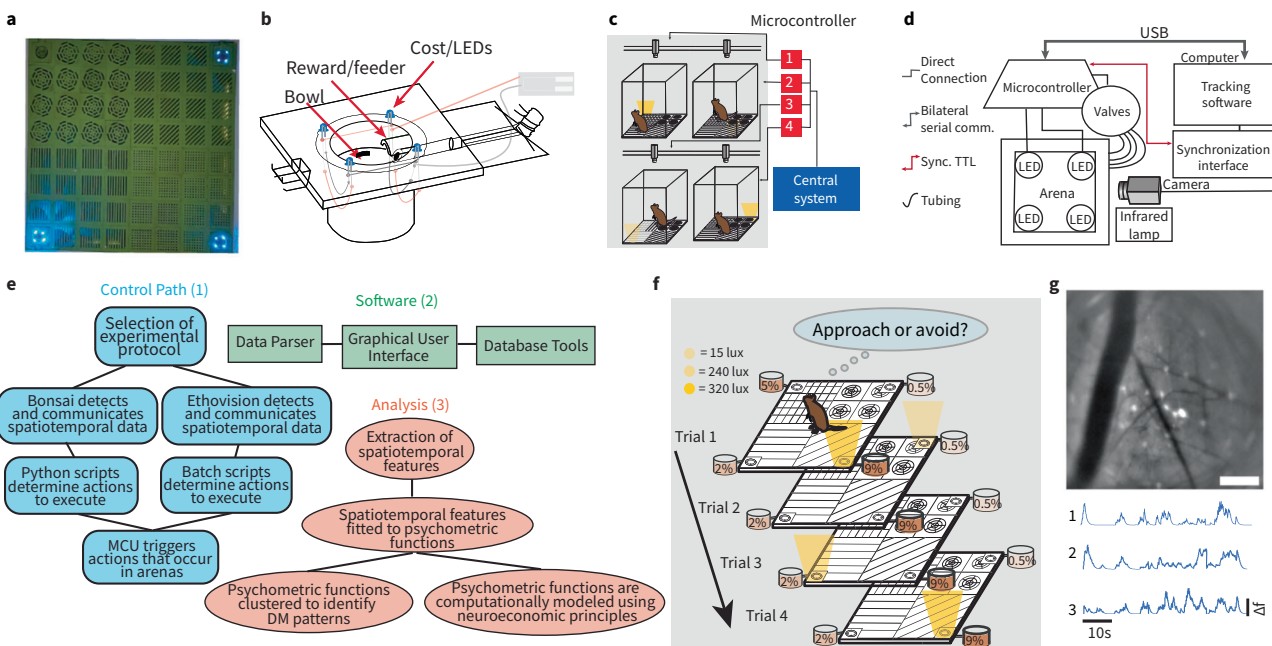

**Fig. 1 | Modular and integrated system design. a** The maze floor is 3D-printed with four different patterns to distinguish the spatial locations of four bowls. Each bowl delivers a specific concentration of liquid reward. Four examples of LED light intensities are shown, from lowest to highest (upper left, moving clockwise). **b** Design of the feeders that deliver liquid rewards (e.g., sucrose at pre-determined concentrations) and cost (e.g., different light levels) via a ring of LEDs. Rewards are only dispensed if the animal's location is detected to be close to the feeder (reward zone), meaning the animal must approach the illuminated LEDs to receive the sucrose solution. Light levels at each bowl can vary across trials and can be observed by the subject from a distance. **c** One microcontroller unit per maze (MCU; red boxes) allows each arena to run autonomously. Each MCU is connected to and communicates with one central system (blue box). This allows for flexibility in where the mazes are placed and increases the number of arenas that can be used concurrently. **d** Schematic of the complete maze set-up. Spatial information is collected through an infrared camera and used by NOLDUS or Bonsai to approximate an animal's location and posture. The computer sets behavioral programs for each MCU. Data from the MCUs is sent to the computer and stored. **e** We developed three sets of software for RECORD. (1) Depending on which experiment was selected, either Bonsai or NOLDUS was used to detect spatial information. Custom Python or batch scripts determine what actions should be executed, sending that information

to the MCU. The MCU then executes preprogrammed actions in the arenas (e.g., illuminate LEDs or dispense rewards). (2) Data collected from a session is sent to one of two custom parsers (which combine the trials and check for mismatches and consistency), depending on which experiment was run. The data can be accessed through a custom Graphical User Interface (GUI). Database tools were developed to expedite data retrieval and analysis. (3) Codes are used to extract features of behavior based on animal location, time, and choice. Using these features of animal behavior, we created modeling and analysis tools. We also developed synchronization scripts to allow our system to work with calcium imaging. **f** Cost–benefit Decision Making (decision-making) task. During each trial, the LEDs around one of the bowls are illuminated, signaling that the reward will be dispensed into that bowl. The reward at any spatial location is always the same (0.2–9% sucrose). The animal decides whether to approach the port and consume the reward while being exposed to the LED light or avoid the bowl. The illuminated bowl is randomly determined for each trial. LED intensity varies from 15 to 320 lx and depends on the behavioral protocol. **g** Mean projection image of in vivo GCaMP8f fluorescence measured in the anterior dorsomedial striatum over the course of one behavior session. Bright-colored regions indicate cells exhibiting active calcium dynamics during the recorded session. Scale bar equals 100 μm. Calcium activity (d$f$/$f$) trace of three example cells from the same session. The scale bar indicates 10 s.

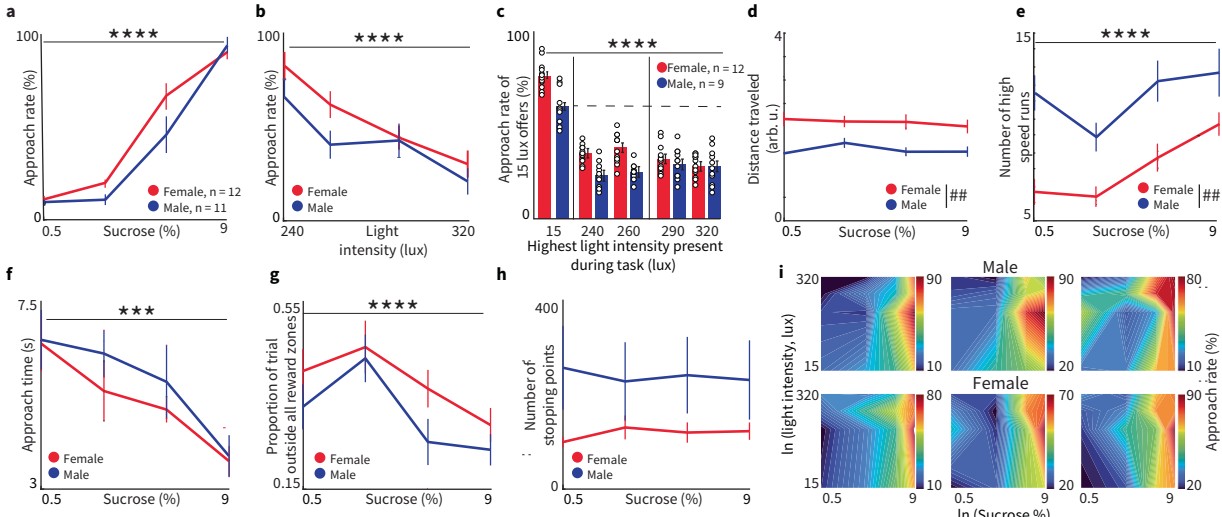

**Fig. 2 | Validation of the RECORD system. a** Approach rates increase as the sucrose concentration (SC) increases. Both sexes approach the reward more as SC increases, indicated by a main effect of SC in the repeated measures ANOVA (ANOVA_RM, ****$p < 0.0001$) and no main effect of sex ($p = 0.06$). Using 0.5–9% sucrose yields sigmoidal psychometric functions. Error bars = mean ± SEM for all plots. Additional statistical analysis is reported in the section "Methods: Statistics and reproducibility". * Is used to denote a significant effect of concentration (*$p ≤ 0.05$, **$p ≤ 0.01$, ***$p ≤ 0.001$, ****$p ≤ 0.0001$). **b** Approach rate decreases as light intensity increases (ANOVA_RM for light intensity ****$p < 0.0001$). Sex had no significant effect on approach rate as light intensity increased ($p = 0.153$). **c** Bayesian analysis examining approach rate of 15 lx offers across sessions where different maximum light intensities were used. The effect of light intensity on approach rate was significant ($p < 0.0001$). While post-hoc analysis found significant sex differences at 15 lx ($p = 0.0006$), 240 lx ($p = 0.0002$), and 260 lx ($p = 0.0002$), there was no main effect of sex on approach (ANOVA_RM, $p = 0.8$). **d–h** Females and males

demonstrate different patterns of time and movement dynamics between the initiation of a trial and when a choice is made. **d** Males travel shorter distances than females (ANOVA_RM, main effect of sex, ##$p = 0.003$) with no effect of SC ($p = 0.089$). **e** As SC increases, the number of high-speed runs increases (ANOVA_RM, main effect of SC, ****$p < 0.0001$). Males had significantly more high-speed runs than females (ANOVA_RM, main effect of sex, ##$p = 0.002$). **f** Male and female rats enter the reward zone faster when higher SCs are offered (ANOVA_RM, main effect of SC, ***$p = 0.0008$) but there was no sex difference (ANOVA_RM, main effect of sex $p = 0.4$). **g** For both sexes (ANOVA_RM, main effect of sex: $p = 0.233$), rats spent less time in the center of the maze and more of the trial in a reward zone for higher SCs (ANOVA_RM, main effect of SC, ****$p < 0.0001$). **h** Stopping points were unaffected by SC ($p = 0.98$) or sex ($p = 0.16$). # Is used to denote a significant effect of sex (# ≤0.05, ## ≤0.01, ### ≤0.001, #### ≤0.0001 **i** Individual cross-benefit integration maps demonstrate that approach rates for individual subjects are not linearly related to cost and reward and are heterogeneous across individual subjects.

motivate the animal (Fig. 5), making it ideal to study rodent models of psychiatric disorders (Figs. 6 and 7).

## Results

RECORD leverages 3D-printed, interchangeable, low-cost components allowing for flexibility in task design (Fig. 1, Supplemental Fig. 1, all 3D-prints, scripts, analytical/modeling codes, and data are deposited as described in Supplemental Table 2). Visual/tactile cues are 3D-printed into arena floors (Fig. 1a), allowing rodents to associate a spatial quadrant with a corresponding reward level. Rewards are dispensed through feeders into small bowls embedded into the arena floor; costs are delivered via LED rings surrounding each bowl (Fig. 1b). RECORD arenas are supported by pillars under the floor (Supplemental Fig. 1a) and wall-supporting pillars, which hold arena walls in a slit (Supplemental Fig. 1b). Modular components allow arenas to be built for an assortment of spaces, tasks, or subjects (e.g., mouse or rat).

Each RECORD arena is regulated by a Microcontroller Unit (MCU) and a custom Printed Circuit Board (PCB, Supplemental Fig. 1c). This allows multiple arenas to operate independently while communicating with a central system (Fig. 1c, d, Methods: 'Standalone microcontroller driven system'). The MCU, through the PCB, signals RECORD to administer cost (LED light) and reward (sucrose in bowls) with microsecond precision.

The electronic setup combines custom and pre-existing software. During decision-making sessions, Bonsai or Noldus Ethovision software tracks each animal's location (see the section "Methods: Spatial tracking for task execution"). Custom-developed programs use spatial information to execute trial events of a decision-making task. Since MCU output is task-dependent and RECORD generated ~1600 trials from ~40 rats run

individually in one of eight arenas per day, we developed parsers; one parses only behavioral data while the other parses behavioral and calcium-activity data (see the section "Methods: Data preprocessing and storage"). We also developed a graphical user interface (GUI) that served as an easy-to-use front-end application for interacting with RECORD data (Supplemental Fig. 1d). We scripted algorithms that extract spatiotemporal information (termed 'behavioral features') from each session for analysis/modeling. The features explored in this manuscript include approach time, approach rate, distance traveled, number of high-speed runs, number of rotations, number of stopping points, proportion of trials outside all reward zones, and reaction time (a description of each feature, their scripts, and scripts for features not used in this manuscript are provided in Methods: 'Spatiotemporal behavioral dynamics'). All data analysis tools, modeling tools, and components used to build the RECORD system are listed in a "key resource table" (Table 2), with details provided in supplemental documents.

Four main decision-making tasks were used for the initial validation of RECORD: reward/cost association, high-cost cost–benefit, low-cost cost–benefit, and alcohol trade-off. All tasks began with a tone signaling the start of a trial, followed by illumination of one feeder. Rats were given 6 s to reach the reward zone. If the rat was within the reward zone (an 'accept' trial), the port remained illuminated while the reward was dispensed and consumed (typically 7 s, Supplemental Fig. 1e). If the rat did not reach the reward zone (a 'reject' trial), the reward was not dispensed, and LEDs were turned off. The reward/cost association task was used to train rats to associate each reward level with a particular floor pattern (Supplemental Fig. 1f, g). The high-cost cost–benefit task presented light intensities of 15 or 320 lx while low-cost tasks only had 15 lx trade-offs. On average, it took ~9 weeks for a rat to be trained for reward/cost association, low-cost

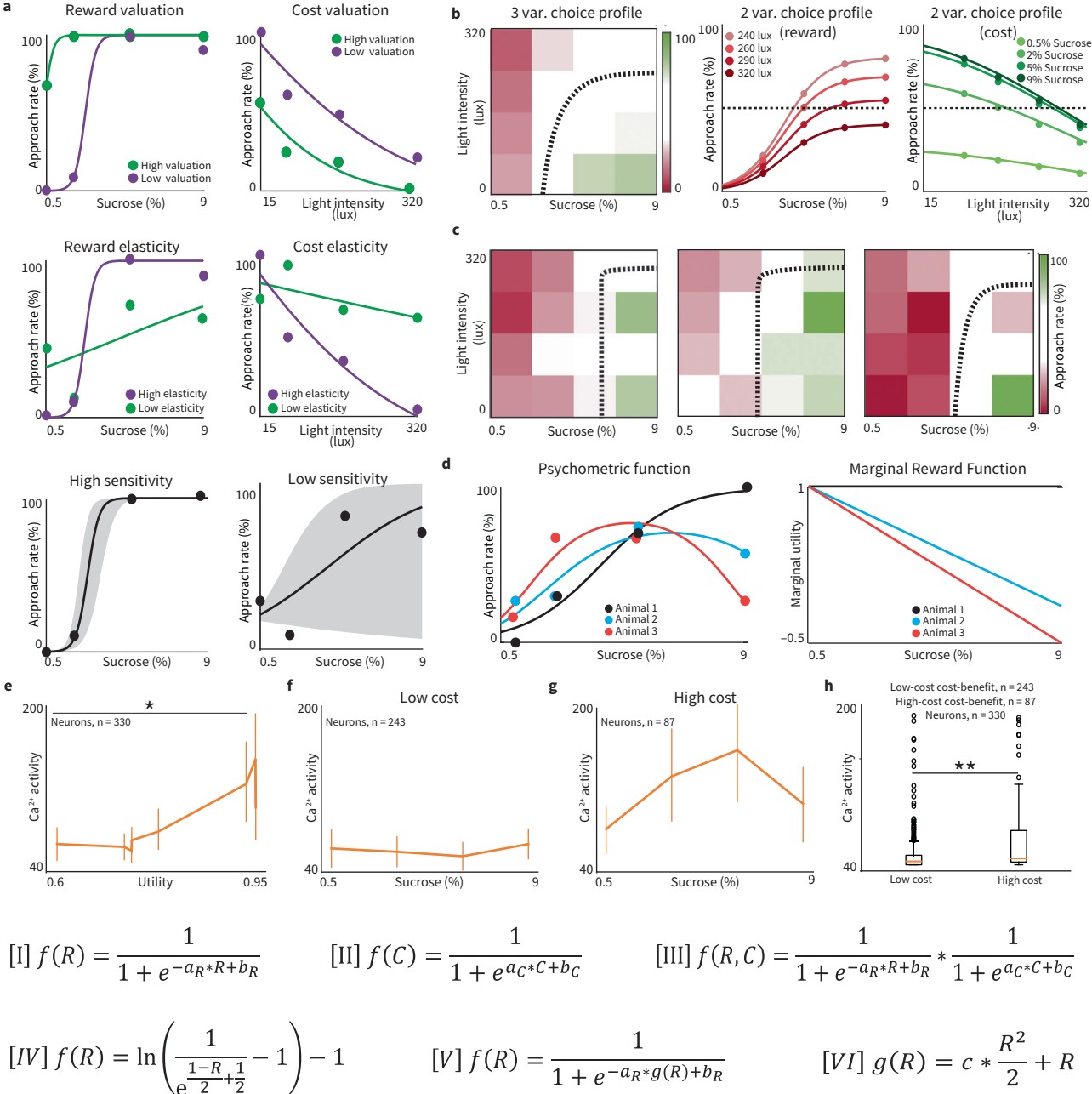

**Fig. 3 | RECORD generates data that can be modeled using neuroeconomic equations. a** Examples of neuroeconomic features used to model RECORD data, valuation (top), elasticity (middle), and sensitivity (bottom). **b** Data gathered from RECORD are also uniquely well-suited for modeling of psychometric functions in the three dimensions of reward, cost, and probability of a decision (left panel) with a dashed line representing where an animal switches from < or >50% approach rate termed a decision boundary. Alternative visualizations of this theoretical example are shown in the middle and right panels, where one horizontal (middle) or vertical (right) slice of a 3D function corresponds to a single classical 2D psychometric function. Fit based on Eq. [iii]. The dashed line represents the point at which the animal switches from more often avoiding to approaching, which can be represented with respect to (w.r.t.) reward by Eq. [iv]. **c** Examples of 3D decision-making profiles from different individuals. The dotted lines represent the decision-making boundaries. **d**. RECORD reveals that in individual sessions, some animals approach in a sigmoid-shaped fashion, but others approach less often as a reward increases, creating psychometric functions with parabolic shapes (left). This may be a manifestation of the law of diminishing returns/marginal utility in microeconomic

theory; the animals value each increment of the reward less as the number of increments increases. The marginal reward is fixed to 1 (black, sigmoid shaped, assumption of panels **a**–**c**) or allowed the freedom to decrease linearly (red and blue), revealing the animal's underlying marginal reward function (right). See Eqs. [v] and [vi]. **e** Calcium imaging of the dorsal medial striatum was performed using the low-cost and high-cost cost-benefit tasks. Calcium activity increases with increasing utility (one-way ANOVA *$p = 0.04$). Error bars = mean ± SEM for all plots. **f** and **g** Average cell trace activity in the dorsal medial striatum during high/low-cost sessions. **h** There is significantly more calcium activity on average during high-light intensity trials compared to low-light-intensity trials ($n = 330$ cellular traces, $n = 243$ low-cost cost-benefit, $n = 87$ high-cost cost–benefit; Main effect of cost: **$p = 0.0012$). Eqs. [I]–[VI]: where $R$ is reward level, $C$ is cost level, $a_R$, $b_R$ are parameters of fit to data where a reward is incremented, $a_C$, $b_C$ are parameters of fit to data where cost is incremented, $g(R)$ is a utility (defined in neuroeconomics as a model of the subjective valuation of reward and cost) function w.r.t. amount of reward consumed and $c$ is a third parameter of fit introduced to Eq. [V].

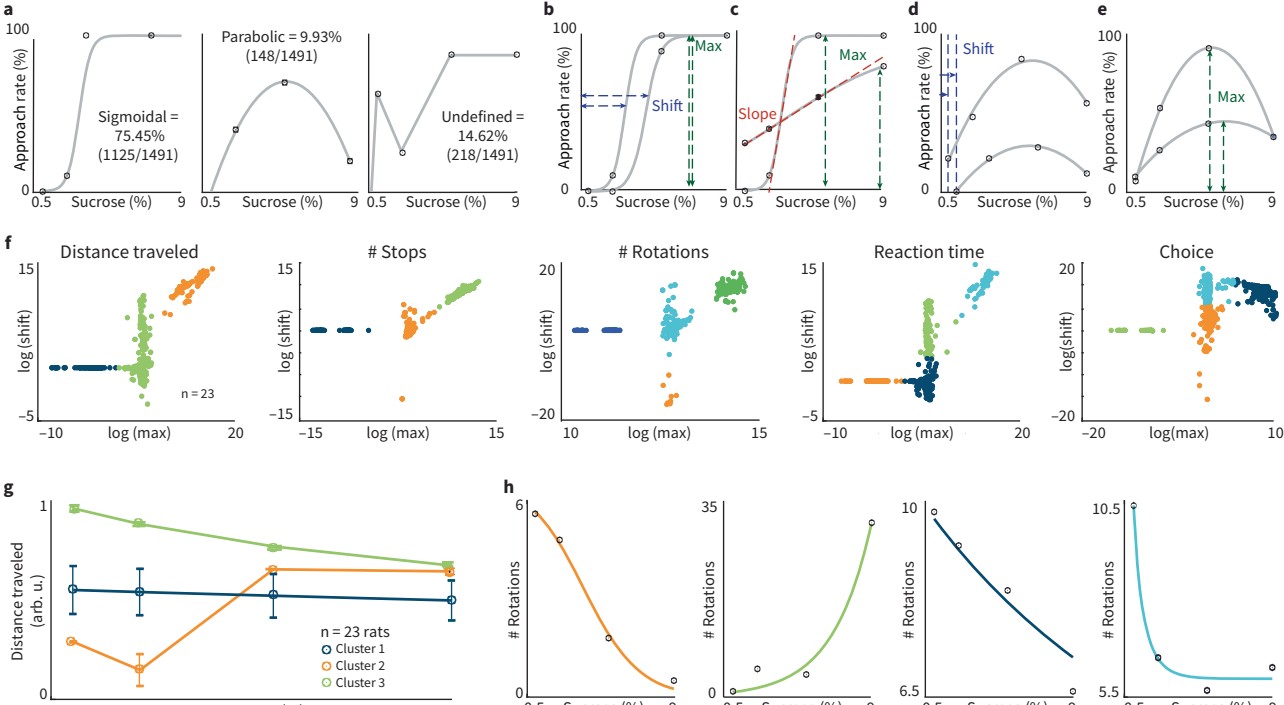

**Fig. 4 | RECORD generates functions that can be clustered to identify individual decision-making strategies. a** For every individual session, we plotted SC against one of the behavioral features (distance traveled, number of stops, number of rotations, reaction time or choice) (see the section "Methods: Psychometric function shape analysis' and 'Classifying decision-making strategies"). We found that all plots were best fit with one of two distinct shapes: sigmoid (left) or, parabolic (middle). Some plots (error >$0.4r^2$) did not fit any function and were classified as 'undefined' (right). Out of 1491 sessions (*n* = 23 rats), 75.45% were sigmoidal, 9.93% were parabolic, and 14.62% were undefined. We defined these as three decision-making patterns. Sigmoidal curves (**b** and **c**) were differentiated by sigmoidal shift, slope, and max. Parabolic curves (**d** and **e**) were differentiated by parabolic shift and max. **f** We sought to determine whether the relationships between the behavioral features operate on a continuum or were constrained in some manner. For sigmoidal curves, we calculated the shift, max and steepness of each sigmoid. Plotting these components against each other, we found discrete clusters (also see Supplementary Fig. 4a) by using the modified partition coefficient alongside a fuzzy *c*-means clustering algorithm. **g** Each cluster represents a distinct psychometric function, relating a feature to reward (3 clusters of Distance traveled in **f**). Error bars = mean ± SEM for all plots. **h** Psychometric functions for an individual rat from the four clusters in # Rotations depicted in panel **f** (see related example, Supplementary Fig. 4b).

cost–benefit, and high-cost cost–benefit tasks (Supplemental Fig. 1h). To demonstrate that RECORD can be used in tandem with in vivo imaging systems, we implanted a GRIN lens and injected calcium indicator virus GCaMP8f into the anterior dorsomedial striatum (DMS), a region central to cost-benefit computations[27] (see the section "Methods: Surgical procedures") and measured calcium activity while the animals performed the low cost and high-cost cost–benefit tasks (Fig. 1g and Supplemental Fig. 1i, j, see the section "Methods: Adaptable decision-making task batteries").

## Rats approach reward and avoid cost

To establish how rats would respond to the cost and reward trade-offs presented by the RECORD system, we administered behavioral tasks where different sucrose concentrations (SC, 0.5%, 2%, 5%, and 9%; "reward") were paired with LEDs of variable intensities (9, 10, 15, 20, 36, 42,160, 240, 260, 290 and 320 lx; "cost"). Unsurprisingly, rats approached reward/cost combinations more as SC increased (Fig. 2a, *p* < 0.0001, Supplemental Fig. 2a, for all statistics see 'Statistics and Reproducibility', Supplementary Note 6) and less as light intensity increased (Fig. 2b, *p* < 0.0001, Supplemental Fig. 2b). Analyzing approach rate along a spectrum of cost revealed context-dependent decision-making. The acceptance rate of rewards paired with 15 lx during sessions where only 15 lx was presented was ~80%. The acceptance rate of rewards paired with 15 lx in sessions with interspersed trials of higher light intensities was significantly lower (*p* < 0.0001, Fig. 2c).

The spatiotemporal characteristics we term 'behavioral features' can be analyzed in several ways. Behavioral features can be examined by cost and reward level, an approach that shows some sex differences. Across all trials, females traveled further than males (*p* = 0.0025, Fig. 2d) while

males had more high-speed runs than females (*p* = 0.0018, Fig. 2e); other features had no significant sex differences (Fig. 2f–h Supplemental Fig. 2c–h, see the section "Methods: Spatiotemporal behavioral dynamics"). Across both sexes, the number of high-speed runs increased with SC (*p* < 0.0001, Fig. 2e) while both approach time (*p* = 0.0008, Fig. 2f) and proportion of trial outside all reward zones (*p* < 0.0001, Fig. 2g) decreased as SC increased. Behavioral features were also compared across accept versus reject trials. Examining accept-only trials, females traveled further than males (*p* = 0.0038) and both sexes traveled less as SC increased (*p* = 0.0002, Supplemental Fig. 2i). Also, during accept-only trials, both the number of stopping points (*p* = 0.0015, Supplemental Fig. 2j) and number of high-speed runs (*p* < 0.0001, Supplemental Fig. 2k) increased as SC increased while the proportion of trial outside all reward zones decreased (*p* = 0.0003, Supplemental Fig. 2l). During reject-only trials, distance traveled was the only feature significantly affected by SC (*p* = 0.046) and sex (*p* = 0.014, Supplemental Fig. 2m). Number of stopping points had no significance (Supplemental Fig. 2n) while males ran more than females (*p* = 0.002, Supplemental Fig. 2o). There were no significant interactions detected for proportion of trial outside all reward zones (Supplemental Fig. 2q). Our data analysis tools allow behavioral features to be tracked across time within a trial or depicted in the spatial location where they were expressed (Supplemental Fig. 2q,r).

In summary, RECORD can generate cost-benefit maps displaying approach rates or behavioral features across multiple levels of rewards and costs. These can be averaged (Fig. 2a–h) or plotted for individuals, where the higher approach rates at higher light intensities suggest that for some rats, intense light may serve as a cue, rather than a deterrent, for the highest SCs, (Fig. 2i) and used to compare different experimental conditions.

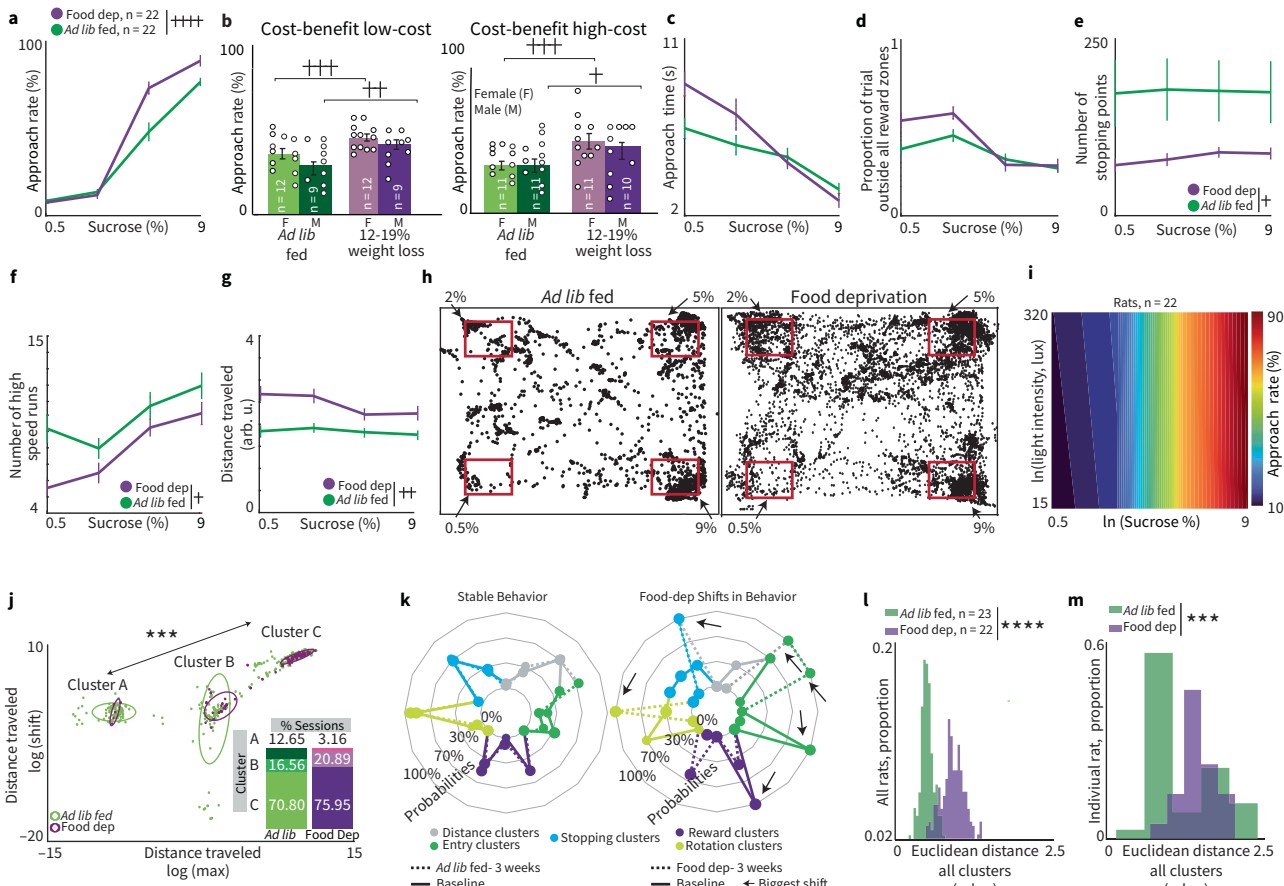

**Fig. 5 | Food deprivation alters decision-making. a** While food-deprived, rats will approach higher SCs when compared to ad libitum fed sessions (main effect of deprivation, ANOVA$_{RM}$, $++++p = 0.0001$, Tukey–Kramer honestly significant difference 5%: $p = 0.0005$, 9%: $p = 0.0049$). One rat developed a tumor during the alcohol task period and did not finish the full experiment, thus the incomplete data for that rat was not analyzed. + Is used to denote a significant effect of condition (+ ≤0.05, ++ ≤0.01, +++ ≤0.001, ++++ ≤0.0001). Error bars = mean ± SEM for all plots. **b** Food deprivation yielded a significant effect on both male and female for low cost (effect of condition, male **$p = 0.0032$, female ***$p = 0.0009$). In high-cost trials, approach rate was significantly different for both male and female groups (effect of condition, male *$p = 0.0126$, female ***$p = 0.0004$). **c–g** Food deprivation alters temporal and spatial aspects of behavior during decision-making. Approach times (**c**, $p = 0.195$) and proportion of trial spent outside of reward zones (**d** $p = 0.12$) were not significantly impacted by food deprivation. During food deprivation, stopping points decrease (**e**, ANOVA$_{RM}$ $+p = 0.04$), rats had less high-speed runs (**f** ANOVA$_{RM}$ $+p = 0.02$), and distance traveled during each trial is increased (**g** ANOVA$_{RM}$ $++p = 0.0016$). **h** Plots depicting an individual rat's spatial location across a baseline and food deprivation session. Each point represents the location of the rat every 100 ms during the session. The rat moved more during a session under

food deprivation (right) than ad libitum conditions (left). **i** During food deprivation, cross-benefit integration maps became homogenous straight lines and the non-linear relationship between reward and cost disappeared ($n = 21$: $F = 12$, $M = 9$). **j** Plot depicting cluster shifts between ad libitum fed and food deprived groups. Each data point represents two parameters for an individual session. We observed a significant difference in the count of sessions that belong to cluster A after food deprivation (chi-squared, ***$p = 0.0005$). Ellipses represent one standard deviation around cluster centroid. **k** Radar plots comparing ad libitum vs. food deprivation performance of individual rats vs. its baseline ("non-responder" on left plot, "responder" on right). Arrows point out the biggest visual differences between radar plots. **l** Using the average cluster distribution of ad libitum-fed rats, we created a distribution. We then calculated the Euclidean distance of each cluster of distance between ad libitum fed and food deprivation conditions and found that food deprivation shifted the peak completely outside of the baseline distribution. These shifts were statistically significant for both groups (****$p < 0.0001$, determined by two-sample Kolmogorov–Smirnov test). **m** Euclidian distances between individual rats for ad libitum fed and food deprivation conditions (***$p = 0.0006$, two-sample Kolmogorov–Smirnov test).

## RECORD enhances neuroeconomic modeling

RECORD's ability to offer numerous reward/cost trade-offs during a single session generates data well-suited for computational modeling. Several levels of rewards and costs enable the modeling of neuroeconomic principles such as subjective value or choice utility[2]. These models can be generated for and compared between multiple experiments to examine how a variable context/condition influences decision-making. All codes for the modeling done in this manuscript are provided in the section "Methods: 'Neuroeconomic modeling of decision-making'".

We first computed decision boundaries[20,28–30], which were determined by the linear regression of a subject's approach rate and indicated where a rat switched between approaching and avoiding reward/cost combinations

50% of the time in plots of reward versus cost levels. We then assessed three neuroeconomic features, which we termed 'valuation', 'elasticity', and 'sensitivity' (Fig. 3a). Valuation was identified by the point/s where the psychometric function begins to rise/fall in relation to changes in reward or cost level. Elasticity was determined by the slope of the function, representing how rapidly a rat shifts between approach and avoid behaviors as reward or cost is changed. Sensitivity was measured as variability along the decision-making boundary.

Comprehensive behavioral models depicting the relationship between rewards, costs, and choice (accept or reject) (Fig. 3b, left) can be separated into plots depicting individual levels of cost (Fig. 3b middle) or reward (Fig. 3b right), however, some behavioral trends identifiable in multilevel

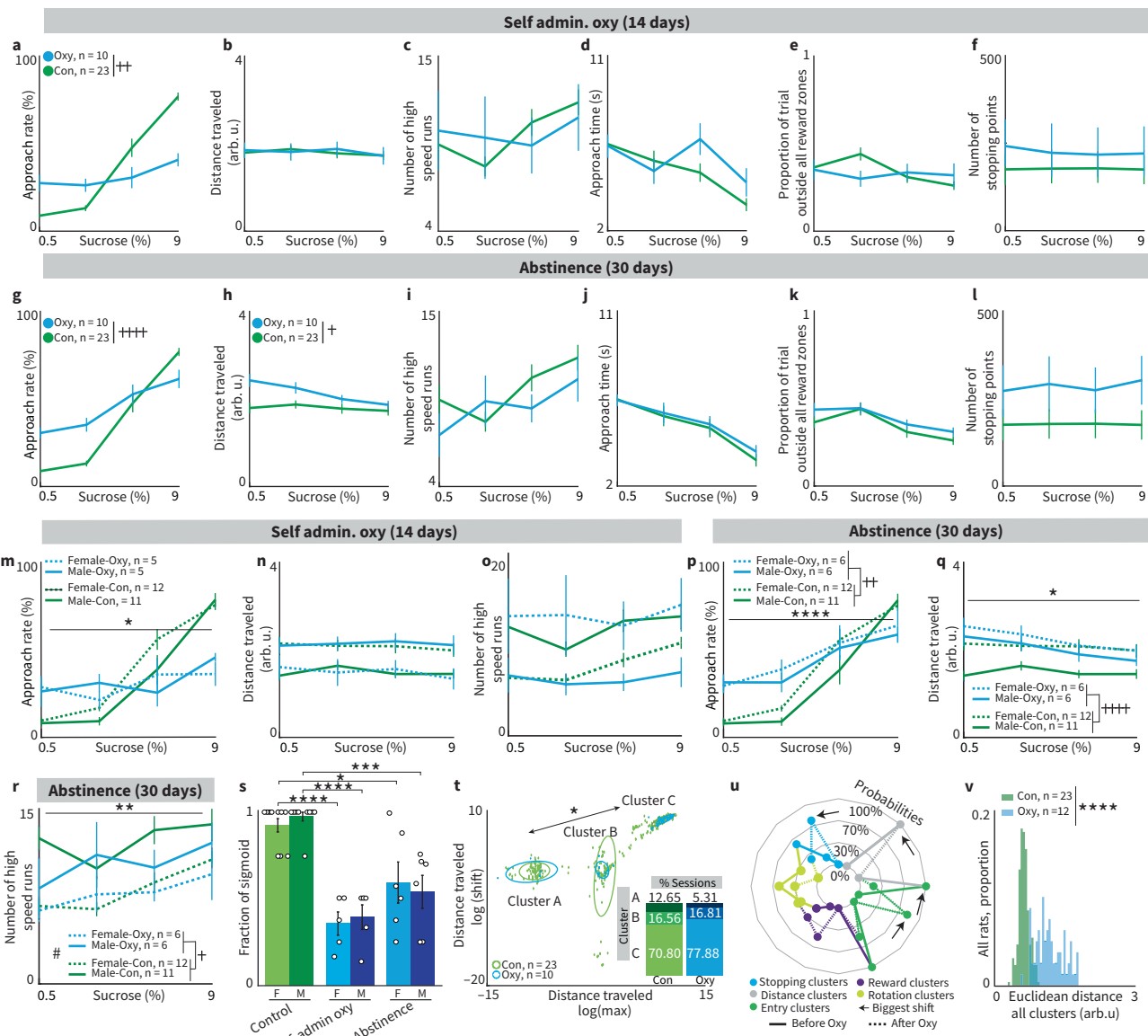

models are not visible when analyzing a single level of cost or reward (Fig. 3c). Cluster analysis was performed on multilevel behavioral models and decision boundaries to identify distinct decision-making patterns. We identified three subtypes of sigmoidal psychometric functions (Supplemental Fig. 3) for plots of approach rate versus reward level, which we term 'curve', 'corner', and 'vertical'. For curve functions, high-level rewards were rejected with gradually increasing frequency as costs increased (Supplemental Fig. 3a, b). Corner functions (Supplemental Fig. 3c, d) were like curves, but with a steep drop-off of acceptance of high-level rewards for higher costs; corner functions may have low elasticity and change decisions quickly after a "threshold" is reached (Supplemental Fig. 3g, h). Vertical functions are defined by more than 50% of high-level rewards accepted regardless of cost suggesting a high valuation of reward (Supplemental Fig. 3e, f). We also observed inverted u-shaped (parabolic) functions, which were best fit by a marginal utility function, and depict decreasing subjective values of higher SCs (Fig. 3d).

Due to our success modeling marginal utility (Fig. 3d) and building upon previous work where the DMS was implicated in encoding subjective value[3,21], we examined whether there was a correlation between DMS activity and total utility (an economic principle defined as the amount of satisfaction garnered through a transaction). Pairing RECORD with

Inscopix, we found that average calcium activity in the DMS (measured during each trial of a behavioral session) increased with total utility (Fig. 3e, $p = 0.04$, $n = 330$ cell traces). When we analyzed average calcium activity during the low-cost cost–benefit and high-cost cost–benefit tasks, there was no significant effect of SC (Fig. 3f, g). There were significant differences in the magnitude of calcium activity between tasks (Fig. 3h, $p = 0.001$). This suggests that calcium activity in the DMS is typically more sensitive to changes in cost.

Overall, an experimental system utilizing multiple levels of rewards and costs facilitates the examination of the correlation between brain activity and decision-making while also enabling the application of modeling methods.

### RECORD identifies individual/group decision-making patterns

RECORD is high-throughput, offers various reward/cost combinations, and allows rodents to move freely so that the interactions between reward, cost, and behavior during decision-making can be thoroughly examined. Plotting behavioral features as a function of SC led to two predominant psychometric shapes. After fitting individual curves generated during 1491 behavioral sessions across 23 rats, 75.47% were sigmoidal, 9.93% were parabolic (inverted-U), and 14.62% were undefined (Fig. 4a). Each sigmoid function was fitted using three parameters. 'Shift' was the horizontal distance from

**Fig. 6 | Decision-making with substance use: Oxycodone. a–f** Behavioral features compared between the control and oxycodone conditions during the 14 days of self-administration. Two rats (one male and one female) were incapable of running behavioral sessions after cocaine self-administration, and thus, their behavioral data were not collected. **a** Approach time was flattened during oxycodone self-administration ($++p = 0.0049$, Kolmogorov–Smirnov test), while the other features demonstrated no significant differences between conditions (**b** distance traveled: $p = 0.9132$, **c** number of high-speed runs: $p = 0.97$, **d** approach times: $p = 0.26$, **e** proportion of trial outside all reward zones: $p = 0.75$, **f** number of stopping points: $p = 0.5$). Error bars = mean ± SEM for all plots. **g–l** Behavioral features compared between the control and oxycodone conditions after 30 days of abstinence. **g** Approach time remained significantly different ($+p = 0.0187$) from the control condition. **h** Rats traveled significantly further after 30 days of abstinence from oxycodone ($+p = 0.045$). All other features remained were not different between groups (**i**, number of high-speed runs: $p = 0.44$, **j** approach time: $p = 0.74$, **k** proportion of trials outside all reward zone: $p = 0.32$, **l** number of stopping points: $p = 0.17$). **m** Oxycodone self-administration produced a unique set of sex differences in decision-making that differed from those observed in the control group ($p = 0.17$, Sex × Condition interaction, $*p = 0.04$, Sex × Concentration interaction). One male and one female rat were incapacitated throughout oxycodone self-administration and were unable to perform behavioral sessions. **n, o** While the magnitude of sex differences was not impacted by oxycodone self-administration, males became more female-like and vice versa for two behavioral features. **n** Whereas oxycodone increased the distance traveled for males, it decreased the distance traveled for females ($p < 0.0001$, Sex × Condition interaction). **o** Similarly, oxycodone increased the high-speed runs for females, while decreasing them for males ($p < 0.0001$, Sex × Condition interaction). **p–r** After 30 days of abstinence, behavioral features altered by oxycodone shift towards pre-oxycodone values for both males and females but remain impacted by prior self-administration. The approach rate remained significantly different between the OXY and CON conditions (**p**, main effect of condition, $++p = 0.0016$; Condition × Sucrose Concentration interaction,

$****p < 0.0001$). No sex differences were observed in the Oxy group for any reward level (post-hoc, $p < 0.08$ for all points). **q** For females, the distance traveled returned to pre-oxycodone levels, but males still showed significantly greater distance traveled during abstinence ($p = 0.4449$ and $p = 0.0093$, respectively, Sex × Condition interaction, $p = 0.0165$, $n$-way ANOVA$_{RM}$). Concentration ($****p < 0.0001$) and condition ($++++p < 0.0001$) effects were significant. **r** In contrast, the number of high-speed runs returned to pre-oxycodone levels for both sexes ($**p = 0.0057$ effect of concentration, $p = 0.0853$ effect of oxycodone, $p = 0.0506$ effect of sex, and $p = 0.2045$ Sex × condition interaction, $n$-way ANOVA$_{RM}$). **s** After analyzing the frequency of sigmoidal data for the approach rate measure, we found that oxycodone significantly reduced the percent of the session with sigmoidal psychometric functions (CON ~90% vs. Self-admin ~40%; one-way ANOVAs: female CON vs. female Self-admin and male CON vs. male Self-admin, $****p < 0.0001$). After 30 days of abstinence from oxycodone, sigmoid frequency recovered to ~55% but was still significantly lower than the levels observed in controls (female CON vs. female Abstinence, $p = 0.01$; male CON vs. male Abstinence, $***p = 0.0003$). **t** Plot depicting "macro-migration" between control and oxycodone groups. Each data point represents a fitting parameter for an individual session. We detected a significant population migration away from cluster A after oxycodone self-administration (chi-squared $p = 0.0235$). Ellipses represent one standard deviation around the cluster centroid. **u** Radar plot comparing baseline and abstinence conditions of a "responder" rat's behavioral clusters. A 'responder' was defined as an individual with a significant shift in Euclidian distances after oxycodone task performance. Arrows indicate the clusters within each behavioral feature that are shifted. **v** Using the average cluster distribution of baseline rats, we created a normal distribution. We then calculated the Euclidean distance of each cluster of the distance between baseline and oxy conditions and found that oxycodone shifted the peak completely outside of the baseline normal distribution. This shift in the Euclidean distance of oxycodone cluster distributions was statistically significant ($****p < 0.0001$, determined by the two-sample Kolmogorov–Smirnov test).

---

the $y$-axis, 'slope' was measured on the linear aspect of the sigmoid, and 'max' was the upper limit of the sigmoid (Fig. 4b, c). Parabolic functions were similarly parametrized by shift and max (Fig. 4d, e). To compare behavioral strategies across rats, we generated 2-D plots of sigmoidal parameters for each behavioral feature and observed discrete clusters within every plot (Fig. 4f, Supplemental Fig. 4a). This suggests decision-making behavior is variable in a finite number of ways, and the ways it varies is surprisingly consistent across individuals.

To further explore clusters, a psychometric function was calculated for each cluster by averaging the psychometric functions from all sessions within a cluster. Using distance traveled as an example (Fig. 4f, left), it is apparent that each cluster consists of sessions with distinct psychometric functions (Fig. 4g). Further, the psychometric functions of individuals within the clusters can be used to examine their preferred behavioral patterns [examples of an individual rat from each cluster in plots of the number of rotations (Fig. 4h) or reaction time (Supplemental Fig. 4b)]. The parabolic psychometric functions, which were mostly observed during initial stages of task performance (Supplemental Fig. 4c), did not exhibit distinct clusters across their parameters (Supplemental Fig. 4d).

We examined variability in decision-making behavior across animals by comparing how reward and cost impact different behavioral features (distance traveled, number of stops, etc.). Though we compared behavior across animals here, one could also use RECORD to identify an individual's preferred behavioral strategy by quantifying the number of sessions for a specific individual in each cluster. This detailed analysis of individual behavioral patterns can be used to identify decision-making sub-populations (groups of animals with similar behavioral preferences). RECORD enables this type of analysis by facilitating data collection on large numbers of subjects, with thousands of data points gathered across sessions fit psychometric functions, followed by clustering analysis of the parameterized functions longitudinally. This data shows that decision-making behaviors exhibit 'constrained heterogeneity' across individuals; this may be a behavioral correlate of the dimensionality reduction observed in neuronal activity patterns[31,32].

## Food deprivation alters decision-making

Food deprivation (FD) may be used to motivate rodents to learn and perform decision-making tasks, but this itself may shift decision-making outcomes. RECORD functions without any form of deprivation, enabling comparisons of decision-making between FD and non-deprived conditions in a within-subject manner (see the section "Methods: Food deprivation alters decision-making"). Approach rates increased after FD (Fig. 5a, $p < 0.0001$) driven by increased approach for high-sucrose rewards (Fig. 5a, post hoc, 5%: $p < 0.001$, 9%: $p = 0.005$). Rats who underwent FD accepted rewards more in low and high-cost conditions, regardless of sex (Fig. 5b).

Approach time and proportion of time spent outside the reward zone were not impacted by FD (Fig. 5c, d) but other behavioral features were altered: rats stopped less often ($p = 0.043$), accelerated less ($p = 0.025$), and traveled further during sessions ($p = 0.002$, Fig. 5e–g) compared to ad libitum conditions. These patterns were mostly recapitulated when analyzing accept-only or reject-only trials (Supplemental Fig. 5a–h). An averaged cost-benefit decision map demonstrates that approach rates were nearly linear across reward values (Fig. 5i vs. Fig. 2i), showing that FD induced robust insensitivity to cost (Fig. 5i, individual examples Supplemental Fig. 5n).

While the approach-enhancing impact of FD (Supplemental Fig. 5i, diet [ad libitum vs. FD]: female $p = 0.0042$, male $p = 0.003$) was observed in both sexes (Supplemental Fig. 5i), FD altered sex differences for some behaviors. Sex differences in distance traveled were exacerbated by FD, while sex differences in the number of high-speed runs were decreased (Supplemental Fig. 5j–m, interactions between sex and diet: distance traveled: $p = 0.012$, number of high-speed runs: $p = 0.02$). Sex differences in other measures were not impacted by FD.

We explored how behavioral patterns are altered by FD by comparing cluster distributions of the parameterized psychometric functions between FD and *ad libitum* conditions (Fig. 5j–m, and Supplemental Fig. 5o–r). There were subtle but significant shifts in cluster distribution (behavioral patterns) for distance traveled (Fig. 5j, Chi-square test, $p = 0.0005$). We compared individual changes in decision-making strategies across all features using radar plots (Fig. 5k, and Supplemental Fig. 5o, p). We found

behaviors to be significantly different between *ad libitum* fed and FD conditions by analyzing the difference in Euclidean distances between baseline and the two conditions (Fig. 5l, $p < 0.0001$, two-sample Kolmogorov–Smirnov test and Supplemental Fig. 5q, r, $p < 0.0001$,

$p < 0.0001$). Finally, we examined FD-induced shifts in Euclidean distances within single individuals. The impact of FD was observed even in the distribution shift of Euclidean distances for an *individual* (Fig. 5m, $p = 0.0006$, Kolmogorov–Smirnov test). By comparing whether an experimental

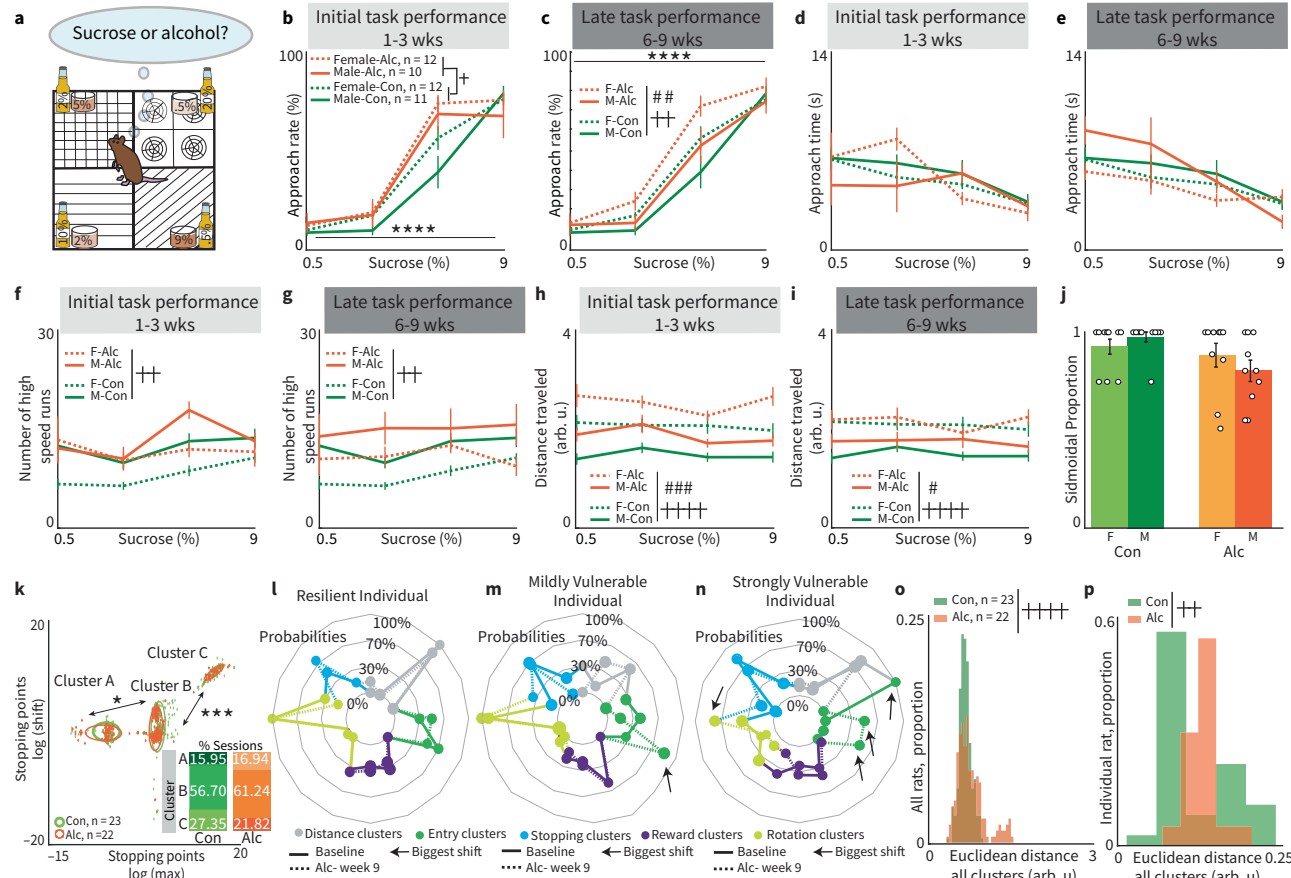

**Fig. 7 | Alcohol decision-making. a** During the alcohol task, rats were presented with a trade-off between sucrose and ethyl alcohol. 0.5% sucrose was delivered in 20% alcohol, 2% was delivered in 10% alcohol, 5% was delivered in 2% alcohol, and 9% was delivered in 0.5% alcohol. Outside of this change in reward offered, the task operated similarly to the high-cost and low-cost cost-benefit tasks. We examined how exposure to alcohol during task performance affects decision-making over time (9 weeks) relative to rats running on the version of the task without alcohol (Fig. 2). **b** When initially performing the alcohol trade-off task, male and female rats maintained preferential acceptance of offers with 5% and 9% sucrose while rejecting 0.5% and 2% sucrose (main effect of SC, ****$p < 0.0001$) with no sex differences (sex, $p = 0.4$). However, alcohol-exposed rats had significantly increased approach rates (main effect of task type, +$p = 0.0178$; group × SC interaction, $p = 0.0025$). Error bars = mean ± SEM for all plots. **c** After nine weeks of task performance, all three main effects were significant (sex ##$p = 0.0015$, condition ++$p = 0.01$, concentration ****$p < 0.0001$, n-way ANOVA_RM). The interaction between sex and SC was also significant with females accepting more than males and the control group overall ($p = 0.044$, *n*-way ANOVA_RM). Females approached significantly more than males ($p = 0.03$) when taking both initial and late alcohol task performance groups into consideration. **d** Approach time tracked during the initial (first 3 weeks) task performance had an overall significant main effect of concentration ($p < 0.0001$), but no other significant interactions when compared to the control group (sex × condition: $p = 0.254$, sex × concentration: $p = 0.2729$, task type × concentration: $p = 0.7127$) or main effects (sex = 0.9775, task type = 0.4457). **e** Approach time after prolonged (6–9 weeks) alcohol trade-off task performance continued to have no significant main effects or interactions when compared to the control group. However, there was still a significant response to concentration among the alcohol group ($p < 0.0001$). **f** and **g** For both initial and late task performance, the number of high-speed runs had no significant interactions for sucrose concentration or sex within alcohol groups. However, we detected a significant effect of the condition

when comparing alcohol and control groups (**f** ++$p = 0.0028$, **g** ++$p = 0.0031$), as well as a significant effect of sex (**f** $p = 0.0003$, **g** $p < 0.0001$). **h** and **i** Distance traveled during both initial (###$p = 0.0004$) and late (#$p = 0.045$, ANOVA_RM) task performance was significantly different between sexes, with females traveling further than males, with an overall effect of condition (++++$p < 0.0001$). There were no significant interactions while comparing initial or late task performance to the control group. **j** The alcohol trade-off task did not significantly impact sigmoid frequency, which we defined as the proportion of sessions where approach rate formed sigmoid-shaped data as opposed to parabolic or undefined data (one-way ANOVA male control $n = 10$ vs. male alcohol $n = 10$, $p = 0.36$, female control $n = 12$ vs. female alcohol $n = 10$, $p = 0.08$). **k** Plot depicting "macro-migration" between control and alcohol groups. Each data point represents a fitting parameter for an individual session. We detected a significant population migration in clusters A ($p = 0.0232$) and C ($p = 0.0003$) compared to control group clusters ($n = 20$, $M = 10$, $F = 10$). Ellipses represent one standard deviation around cluster centroid. **l–n** Examples of radar plots comparing baseline and alcohol conditions of the behavioral cluster of three individual rats who had increasing differences in Euclidean distances before and after alcohol trade-off task performance. Some rats maintained consistent cluster probabilities (e.g., left, low), some rats had moderate shifts in cluster probabilities (middle), and other rats had high differences between cluster probabilities measured using Euclidean distances. Arrow indicates a large shift. **o** Using the average cluster distribution of baseline rats, we created a distribution. We then calculated the Euclidean distance of each cluster of distance between baseline and alcohol conditions and found that alcohol shifted the peak completely outside of the baseline distribution. These shifts were statistically significant for both groups (****$p < 0.0001$, determined by two-sample Kolmogorov–Smirnov test).
**p** Euclidian distances between clusters for baseline and alcohol trade-off conditions (**$p = 0.0011$, two-sample Kolmogorov–Smirnov test).

manipulation shifts the distribution for an individual, one could potentially characterize the extent to which the individual is resilient (no shift in distribution) or vulnerable (significant shift) to the manipulation.

Collectively, our data demonstrate that FD non-specifically enhances the acceptance of cost-benefit tradeoffs by driving insensitivity to cost (Fig. 5b, i); FD also reshapes sex differences (Supplemental Fig. 5i–m). Since RECORD does not require any deprivation, it avoids the impact, bias and alterations FD causes across decision-making and its associated behaviors.

### Prior oxycodone usage impacts decision-making

Given that opioid overdose deaths increased 14% from 2020 to 2021[33], and that substance use impacts human decision-making[34], it is important to explore altered decision-making that accompanies substance use. To establish RECORD's sensitivity to differences in decision-making before, during, and after opioid administration, we conducted an experiment examining how oxycodone self-administration and abstinence impact decision-making. Rats were trained on the low-cost cost–benefit task and then exposed to a 14-day period of oxycodone self-administration in an operant chamber (see the section "Methods: Oxycodone self-administration and abstinence"). On average, rats self-administered 3.99 mg/session. Consistent with other studies[35], females administered roughly twice as much oxycodone as males (females: 5.6 mg/session; males: 2.7 mg/session). Three to four hours after self-administration, rats ran a RECORD cost-benefit task (see the section "Methods: Oxycodone behavioral task"). After the self-administration period, rats entered forced abstinence where they continued to perform the RECORD task. Throughout, their performance was compared to the control group of rats who trained and performed RECORD tasks without any manipulations (Fig. 2).

During 14 days of oxycodone self-administration, the relationship between SC and approach rate flattened (Fig. 6a, $p = 0.1051$ ANOVA, $p = 0.005$ KS test). This hyposensitivity to reward magnitude during opioid administration is consistent with several other studies[36–38]. When comparing task performance during the 14 days of self-administration to that of the control group, no other behavioral feature differed between the two groups (Fig. 6b–f). After 30 days of abstinence, the approach rate remained flattened compared to the control condition (Fig. 6g, $p = 0.019$), suggesting a persistent reward hyposensitivity across reward levels. Amongst other behavioral features, only the distance traveled differed between the two groups after abstinence (Fig. 6h–l, $p = 0.045$). These data might suggest a limited impact of oxycodone use on decision-making behavior; however, a different picture emerges when sex is considered.

There were reduced sex differences in approach rate (Fig. 6m, $p = 0.04$, Sex × Concentration interaction; $p > 0.05$ for all post-hoc comparisons between Female-oxy and Male-oxy) but not for approach times (Supplemental Fig. 6a). Importantly, sex differences in other behavioral features were impacted by oxycodone. For example, oxycodone self-administration increased the distance traveled by males but decreased the distance traveled by females (Fig. 6n, Sex × Condition interaction, $p < 0.0001$). Similarly, these opposing effects were observed for the number of high-speed runs (Fig. 6o), time spent in reward zones, and number of stopping points (Supplemental Fig. 6b, c). Collectively, oxycodone self-administration impacts decision-making behavior within sex, and altered sex differences in cost-benefit decision-making.

Many sex differences in behavioral features returned to pre-oxycodone levels after 30 days of abstinence, including approach rate (Fig. 6p, Sex × Condition interaction, $p = 0.65$). Distance traveled was the only behavioral feature to show perturbed sex differences after 30 days of abstinence (Fig. 6q, Sex × Condition interaction, $p = 0.0165$). All other features returned to pre-oxycodone levels including high-speed runs (Fig. 6r), approach time, proportion of trial spent outside feeder zones, and number of stopping points (Supplemental Fig. 6d–f).

During oxycodone self-administration, the percentage of sessions that displayed sigmoidal functions relating approach rate to reward levels was reduced to ~40%. After 30 days of abstinence from oxycodone, we found that the percentage of sessions with approach rate sigmoidal functions

increased to ~55%, which was still significantly below the pre-drug levels of ~95% (Fig. 6s). When comparing sigmoid session frequency across individual rats, we found that all rats were impacted by oxycodone self-administration, but there was significant inter-individual variation (Supplemental Fig. 6g). A significant correlation between the amount of oxycodone self-administered and the percentage of sessions that generate sigmoid shaped functions suggests that greater oxycodone consumption leads to the decreased percentage of sigmoidal functions used to relate costs to rewards (Supplemental Fig. 6h).

Radar plots and Euclidean distance analysis provide insight into behavioral shifts after oxycodone self-administration (Fig. 6t–v). Distance traveled had significant shifts in preferred clusters (behavioral patterns) when compared to baseline clusters (Fig. 6t, $p = 0.0235$, $p = 0.9455$ and $p = 0.1187$ for clusters left, middle, and right, chi-square test). Calculating the average psychometric function for each cluster within the oxycodone and baseline groups (Supplemental Fig. 6i), we identified three distinct ways that distance traveled relates to reward values. Euclidean distances were significantly different after oxycodone use (Fig. 6v, $p < 0.0001$, Kolmogorov–Smirnov), demonstrating significant changes to behavioral patterns. Individual radar plots comparing baseline cluster probabilities to the last week of oxycodone self-administration (Fig. 6u, Supplemental Fig. 6j, k) depict behavioral shifts. Finally, oxycodone dramatically increased *individual* Euclidean distances between the rat's baseline and abstinence periods (Supplemental Fig. 6l, $p = 0.001$, Kolmogorov–Smirnov). These analyses reveal that the "average" psychometric function (e.g. Fig. 6a) may not be representative of individual behavioral strategies (Fig. 6t). This failure of averaging has been previously described for neuronal responses[39,40] but is underappreciated in behavioral contexts.

### Sucrose alcohol trade-off

To further showcase the versatility of RECORD, we created a task to probe alcohol as a cost/reward. We developed a version of our decision-making task where alcohol was mixed with sucrose solutions to yield four trade-off solutions with inverse concentrations of the two substances: 0.5% sucrose and 20% alcohol, 2% sucrose and 10% alcohol, 5% sucrose and 2% alcohol, and 9% sucrose and 0.5% alcohol (Fig. 7a, see the section "Methods: Sucrose vs. alcohol trade-off").

Analyzing behavioral features between control and different stages of task performance revealed limited interactions. Approach rate during the first three weeks of task performance demonstrated that rats approached higher SCs more than the lower SCs, as expected. However, despite the aversive properties of ethanol[41,42], the approach rate was significantly higher for the alcohol task compared to the control group (Fig. 7b, Condition × Concentration, $p = 0.003$). After nine weeks of task performance, females exhibited higher approach rates than males (Fig. 7c, Sex × Concentration, $p = 0.044$). All other features had no significant interactions across initial-late task performance (Fig. 7d–i).

Unlike oxycodone self-administration, alcohol exposure did not alter the proportion of sessions with sigmoidal psychometric functions (Fig. 7j). However, like oxycodone, the percentage of sessions distributed across clusters varied greatly between the control and alcohol task (Fig. 7k, $p = 0.0232$, $p = 0.1753$, and $p = 0.0003$ for clusters left, middle, and right respectively, chi square test). Radar plots of individual rats with no, moderate and strong differences between initial and late alcohol task performance show how behavioral strategies are differentially shifted across individuals (Fig. 7l–n). Euclidean distances for the population (Fig. 7o, $p < 0.0001$) and an example of an individual rat (Fig. 7p, $p = 0.001$) demonstrate significant changes in decision-making behavior. This type of analysis may enable the identification of resilient or vulnerable subjects, depending on their Euclidean distances before and after substance use/other experimental conditions.

### Discussion

RECORD combines custom-designed hardware and software, providing a customizable tool that quantifies decision-making and behavioral correlates

across experimental conditions. RECORD is an automated system that can identify deviations in decision-making and behavior for sub-groups and individuals. Multiple levels of rewards and costs enable thorough comparisons of decision-making between sessions, conditions, contexts, and across time. RECORD's utility is enhanced by its adaptability and by implementing task series that progressively change one variable to isolate components of decision-making.

RECORD is a comprehensive behavioral system that does not implement food deprivation or water restriction to train rats for behavioral protocol performance. Food and water restriction alters behavior and task performance[26] (Fig. 5) along with reducing the detectability of sex differences in decision-making (Supplemental Fig. 5i–m). This is suboptimal since animal paradigms that limit sex differences are less conducive to developing meaningful clinical interventions[43,44]. Generally, sex differences observed using RECORD tends to align with prior behavioral research, like female rats approaching alcohol more than males[45], however there remain several questions to explore with the RECORD system including exploring sex differences during risky, probabilistic, or intertemporal decision-making in a foraging-like framework.

Another important consideration for research is detecting and examining how/why some individuals are resilient or vulnerable to disorder onset and progression. Experiments have demonstrated that a collection of individual behavioral traits expressed during decision-making can predict task performance[46]. Further, lesioning one of multiple cortical regions caused heterogenous effects on decision-making and associated behaviors depending on the individual[47]. These individual differences can be clustered into groups and can be influenced by external conditions like sleep deprivation or sucrose delivery[48]. Differences across individuals and clusters are quantifiable with RECORD and may be useful for identifying subjects resilient or susceptible to neuropsychiatric disorders by tracking behavioral shifts before, during, and after disorder onset.

Another strength of RECORD is its' sensitivity to detect behavioral differences, conferred by using multiple levels of reward and cost within each subject. For example, if an experiment used a different decision-making task (Table 1) with only rewards valued above 2% sucrose, one might erroneously conclude that oxycodone abstinence does not impact decision-making task performance (Fig. 6g). Our data reveal heterogeneity across individuals in how oxycodone self-administration and its abstinence impact decision-making (Fig. 6s–v, Supplemental Fig. 6g, h), making RECORD a powerful tool to study disorders.

RECORD's sensitivity and approach-avoid design make it well-suited for exploring neuroeconomic principles that underlie decision-making (Fig. 3 and Supplemental Fig. 3). Modeling RECORD data in the context of neuroeconomics enables further insight into how subjects value rewards/costs in varying contexts/conditions. While we focused on modeling subjective value/utility here, other neuroeconomic ideas could be studied using RECORD. For example, models of sensitivity, valuation, or elasticity can be analyzed in the contexts of loss aversion, delay discounting[24], and reward devaluation[49]. Other potential neuroeconomic principles that could be studied with RECORD include how salience[50], utility[2], probability[2], and contingency[2] shift as a consequence of the context. RECORD provides a method where quantifiable neuroeconomic principles can be linked with underlying biological mechanisms.

RECORD's foraging-like environment establishes a framework for studying ecologically relevant decision-making with high experimental control. The tone signaling trial initiation is unnatural, other aspects of naturalistic rodent behavior are included, like free movement, identification of reward locations using visual/tactile cues, and determining whether foraging is worthwhile[23]. This may contribute to the animals' willingness to perform the task without food/water deprivation. Future experiments could implement natural auditory cues[23] or burrows underneath the arena. An additional factor often present in nature is ambiguity. Tasks could incorporate ambiguity by (1) using multiple tones to signal trial onset, but different tones for differing probabilities of reward dispensation, (2) comparing bowls that deliver a particular concentration of sucrose with high-

probability to bowls that deliver the same sucrose concentration with low-probability, or (3) randomize session lengths. Since RECORD is modular, multiple connected arenas would allow rats to access different trade-offs depending on where they explore.

For more thorough behavioral analysis, RECORD could be used with B-SOiD, a software that provides measures of individual limb movements to delineate decision-making states or clusters more definitively[51]. Along with B-SOiD, RECORD could be used with behavioral software tools like A-SOiD[52], DeepLabCut[53–56], SLEAP[57], and BehaviorDEPOT[58], to investigate behavioral and decision-making links in various experimental setups. Overall, RECORD is a versatile tool that is capable of a wide array of decision-making experiments, providing a foundation for the thorough analysis of both decision-making and its associated behaviors.

## Methods
### Animal ethics statement
The project received approval for all protocols from the University of Texas at El Paso Institutional Animal Care and Use Committee and followed the Guide for Care and Use of Laboratory Animals (IACUC reference number: A-202009-1).

### RECORD system overview
In this paper, we outline a novel and customizable behavioral system designed to study decision-making in rodents, the Reward-Cost in Rodent Decision-making (RECORD) system. RECORD synergistically utilizes (1) 3D-printed arenas, (2) microcontroller-driven electronics, (3) camera-based animal tracking, (4) data acquisition software, (5) custom parsing software, (6) databasing software, and (7) analysis software. This system is high throughput while being sufficiently sensitive for the identification and quantification of sub-group/individual decision-making behaviors. The RECORD system can be used either alone or in conjunction with third-party hardware/software for the acquisition of other measures, such as neuronal recordings. Herein, assembly instructions, components used, suggested dimensions, and links to all pertinent GitHub repositories are provided.

### 3D-printed customizable RECORD arenas
The RECORD system is an open field environment that employs visual/spatial cues to signal stimuli location (see the section "Arena floor pattern layout", Fig. 1a) in the context of natural rodent behaviors (foraging). The arena was constructed using both fused deposition modeling (Fdecision-making) and stereolithography (SLA) 3D-printing to create a variety of printed pieces that combine to construct an affordable, modular, and customizable open-field arena. Some pieces were printed with polylactic acid (PLA) using Fdecision-making printing for production volume, other pieces required more precise details, increased sturdiness, and to be watertight, so SLA printing was used. The arenas were the test environment used for all behavioral tasks analyzed in the manuscript. The modular nature of the 3D printed components of RECORD allows arenas to be arranged into different shapes and sizes, as well as implement as many cost/reward offer regions (see the section "Feeders create cost/reward regions of interest") as the electronic hardware (see the section "Standalone microcontroller-driven system") and space allow.

Each arena is made up of individual 3D-printed pieces which act as building blocks that can be put together into different shapes and sizes. Each arena is composed of 9 types of 3D-printed pieces: (1) The feeder (https://github.com/rjibanezalcala/RECORD/blob/main/3d-prints/stl/Feeder%20v1.9.9.stl) which is used to deliver both cost and reward (see the section "Feeders create cost/reward regions of interest"), (2) Feeder base tiles (https://github.com/rjibanezalcala/RECORD/blob/main/3d-prints/stl/Feeder%20Base%20Tile.stl) which hold the feeder piece in place and allow it to be swapped out if needed, (3)–(7) Four different patterned floor tiles, namely Diagonal (https://github.com/rjibanezalcala/RECORD/blob/main/3d-prints/stl/Diagonal%20Floor%20Tile.stl), Grid (https://github.com/rjibanezalcala/RECORD/blob/main/3d-prints/stl/Grid%20Floor%20Tile.

stl), Horizontal (https://github.com/rjibanezalcala/RECORD/blob/main/3d-prints/stl/Horizontal%20Floor%20Tile.stl), and Radial (https://github.com/rjibanezalcala/RECORD/blob/main/3d-prints/stl/Radial%20Floor%20Tile.stl) which map different levels of reward to a location on the arena (see the section "Arena floor pattern layout"), (8) Basic support pillars (https://github.com/rjibanezalcala/RECORD/blob/main/3d-prints/stl/Basic%20Pillar%20v2.1.stl) to hold the arena floor together as well as raise it above the ground to allow electronics to run under the arena and allow drainage (see the section "Arena support pillars"), and (9) Wall support pillars (https://github.com/rjibanezalcala/RECORD/blob/main/3d-prints/stl/Wall%20Support%20Pillar%20v2.0.stl) to both raise the arena floor and hold the arena walls (see the section "Arena walls") in place.

## Arena floor pattern layout

The arena floor was designed to associate the locations of rewards with a corresponding floor pattern. We used four different patterns: Diagonal, Grid, Horizontal, and Radial patterns, plus one feeder base tile (Fig. 1b) per arena section. For the square arena validated in this paper, each corner is consistently associated with a specific reward level, thus allowing rats to make an informed decision to approach or avoid the offered reward/cost combination. Since the arena components are modular, the tiles can be arranged into different sizes and shapes; thus, one may design rectangular, T-shaped, labyrinthian, or other arenas for different experiments.

To avoid difficulties differentiating between similar patterns, diagonal and horizontal floor patterns were never located adjacent to each other on the arena floor (Fig. 1a). The arenas designed for these experiments measure 64 cm along each side.

A detailed guide on the construction of the arena can be found in the RECORD arena setup guide (Supplemental note 3, https://github.com/rjibanezalcala/RECORD/tree/main/documentation), and the ready-to-print (.stl) files used in the validated setup can be found in our ready-to-print GitHub repository (https://github.com/rjibanezalcala/RECORD/tree/main/3d-prints/stl). Modifiable AutoCAD (.dwg) files can be found in the dwg designs GitHub repository (https://github.com/rjibanezalcala/RECORD/tree/main/3d-prints/cad).

## Feeders create cost/reward regions of interest

The feeder piece (Fig. 1b) houses light-emitting diodes (LEDs) and enables the administration of sucrose solutions. The feeder is designed such that light is placed around an offered reward and directed upward, thus signaling the availability of a cost-benefit trade-off as well as serving as an aversive experience if the reward is consumed (cost). All wires and electronics are loaded from the bottom side of the feeder (under the arena floor), and the LEDs are mounted through 5mm diameter holes to protect all components from the animal. Due to RECORD's modularity, feeders may be added to or removed from the arena floor to alter the available decision-making points. Feeders are anchored to the arena floor by a base tile. Feeder pieces sit on the base tiles, held into place by a small hook on the front of the feeder to prevent it from being lifted during an experiment. The feeder can be lifted from the "tail" end to remove it for cleaning and maintenance. Situated at the "tail" of the feeder, which extends past the arena wall, is a protrusion where plastic tubing is attached to the feeder. Liquids are dispensed into the feeder bowl through this tubing and a passive drain at the bottom of the bowl ensures that the rodent is not "stockpiling" a reward from multiple trials. The hole that administers the liquid reward is covered with a hood to make it inaccessible to the rodent, preventing it from accessing any residual reward still in the tube. The feeder piece pictured in this paper was designed specifically with liquid reward in mind, however, if the same mounting surface dimensions on our feeder piece are kept, the same feeder base tile can be used to mount a different reward/incentive delivering apparatus.

## Arena support pillars

The RECORD arena is supported by two types of pillars: basic support pillars (Supplemental Fig. 1a) and wall support pillars (Supplemental Fig. 1b). The wall pillars hold the walls vertically and flush to the side of the arena. Flush walls reduce the shadows cast on the arena floor which could impede animal detection and tracking. These pillars are designed to be used on the edges of the arena with their "slot" facing outward; three wall supports are used per edge on the 8 by 8 tile arenas. The basic pillars, however, simply hold the arena floor together and feature small holes on the top that hold the floor tiles in place and are glued to the tiles. They elevate the arena floor 7 cm above the ground to give clearance for the necessary electronic components and tubing for stimulus delivery underneath the arena. For the arenas used in this manuscript, one basic pillar is used in every place where two or four tiles come together, as well as on the outside corners of the arena floor.

Although printing material selection is not critical for the basic support pillars due to the weight of both the arena and rodent being distributed across all the pillars. It is recommended to use a more durable material for the wall support pillars to support the weight of the wall material without breaking. The arenas described in this manuscript used polylactic acid (PLA) for the basic pillars, and Formlabs Tough 2000 resin for the wall supports.

## Arena walls

The arena walls are polyvinyl chloride (PVC) sheets (75 cm × 64.5 cm × 0.3175 cm) that prevent animals from jumping out of their enclosure, span the whole width of the arenas, and are thin enough to be held by the wall support pillars. The walls also provide isolation from adjacent arenas and the room. On the sides of the arena with feeder pieces, 8.25 cm × 2.3 cm slots were cut to allow the "tail" of the feeder piece to extend outward for access to the protruding feeder spout. All four walls are joined together using corner braces and screws, with any residual gaps between the walls covered by duct tape of a matching color (see Supplemental Note 3).

## Standalone microcontroller-driven system

The embedded electronics (microcontroller) and associated components (LEDs, cables, relays, valves, printed circuit board) enable the execution of programmable and automated cost/benefit decision-making tasks (Fig. 1c, d). Cost is administered through blue-colored LEDs, while solenoid valves deliver sucrose solution as a reward (see 'Cost and reward'). The embedded electronics system is one of the most customizable elements of the entire setup. The microcontroller allows cost and reward magnitude to be adjusted through minor changes to the microcontroller firmware. Another advantage of an embedded system is that one can add as many cost/reward delivery points to one arena as the microcontroller allows, while at the same time not needing extra synchronization hardware for the behavior tracking software or any additional external recording systems.

The Texas Instruments MSP430-EXPFR2355 Development Kit used with the RECORD system features a variety of on-board General-Purpose Input/Output (GPIO) pins, several of which can serve as an output for a timer-dependent pulse-width modulated (PWM) signal, suited for varying light intensity on LEDs. Additionally, this microcontroller features an enhanced universal serial communications interface (eUSCI) that supports the universal asynchronous receiver/transmitter (UART) protocol, which exchanges data between the microcontroller and another device such as a computer. Other pins serve as simple input or output pins for either TTL signals (see the section "Integrating the RECORD system with third-party hardware and software"), that can synchronize the microcontroller and another device, or to drive electronic relays which aid in delivering reward solution to the arena. A custom circuit board was designed as a method for routing signals to their respective locations (Supplemental Fig. 1c, see the section "A custom printed circuit board unifies electronic elements") while protecting the circuitry. The custom circuit board also relays microcontroller signals to the different types of cables that drive the electronics situated near or in the arenas (solenoid valves and cost LEDs, respectively; see the section "Simple cable assemblies carry signals between electronic elements").

All microcontroller code was developed using Code Composer Studio (https://www.ti.com/tool/CCSTUDIO) and is openly available in our

GitHub repository (https://github.com/rjibanezalcala/RECORD/tree/main/microcontroller) along with connection diagrams and guides to aid construction of the system (see Supplemental Note 2, https://github.com/rjibanezalcala/RECORD/blob/main/documentation/electronics_build_guide.pdf). A RECORD user manual is also available, containing setup instructions, extended functions, and additional relevant usage information, available for public access both within this paper and in a GitHub repository (see Supplemental Note 1 and https://github.com/rjibanezalcala/RECORD/blob/main/documentation/RECORD_User_Manual.pdf).

## Cost and reward

During cost/benefit conflict tasks, variable light intensities were used as aversive stimuli while sucrose solutions were used as rewarding stimuli. Blue (~470 nm wavelength) high-intensity light-emitting diodes (LEDs) were used since this color of light has been established as being aversive to rats[19,21,59]. Light also cued which feeder would dispense a reward for each trial. Each feeder was fitted with an assembly of four LEDs connected in parallel, forming a ring around the feeder bowl (see Supplemental Note 2). Varying concentrations of sugar diluted in water (sucrose solution, see 'Adaptable decision-making task batteries'), were deposited into the bowl of a feeder piece. For alcohol tasks, different concentrations of ethanol were mixed with sucrose solutions (see the section "Sucrose vs. alcohol trade-off").

## Integrating the RECORD system with third-party hardware and software

To execute the numerous events needed for a behavioral trial, the RECORD system's microcontroller uses the universal asynchronous receiver/transmitter (UART) protocol via serial USB COM Port communication to receive commands from a compatible operating system such as Windows. It also uses 5V TTL signals to communicate with third-party external hardware. These TTLs can be used as synchronization signals where the MCU might either control external hardware, be controlled by external hardware, or tell another system that a command has been executed. The microcontroller responds to each command it receives by sending an Acknowledgement TTL signal (ACK), signaling that a command has been executed and that it is ready to receive another. Conversely, the MCU can also send a programmable TTL whenever it is requested via command, and the behavior of this TTL signal can be configured to be a short pulse or toggled. The duration of this TTL pulse can also be configured in case a longer or shorter pulse is necessary. In this paper, we used the ACK signal at multiple wait-and-sync stopping points during a trial to signal the behavior-tracking system to continue trial execution based on the animal's location. This was done via the USB I/O hardware interface, though it is worth mentioning that this can be done through any external system's TTL interface. For communication with the RECORD microcontroller, a serial communications interface is needed. This is solved directly by custom Python libraries and scripts, used to run trials with the Inscopix nVista system for in vivo calcium imaging, or with third-party software such as PuTTY, which was used in combination Batch scripts (https://github.com/rjibanezalcala/RECORD/tree/main/microcontroller/batch_scripts) run by Noldus' Ethovision (https://www.noldus.com/ethovision-xt) for behavioral recordings.

## Microcontroller–driven relays and solenoid valves deliver reward to the arena

The RECORD system has an 8-channel, opto-coupled relay shield featuring 5V relay switches. These relays are driven by the microcontroller and isolated from the higher-voltage circuit needed to drive 24V normally closed (NC) solenoid valves. Only the normally open (NO) terminal on the relays were used, these relays open the circuit and cut current to the solenoid valves keeping them closed while the system is off or idle.

## A custom-printed circuit board unifies electronic elements

We designed a custom printed circuit board (PCB) (Extended Fig. 1c) that conveyed electrical signals between the microcontroller, cost LEDs, solenoid valves, and relay switches. The PCB also facilitated TTL signaling between the microcontroller and external hardware. One PCB supports up to 10 LED assemblies and up to 8 relay switches that share the same power source to drive a solenoid valve, thus each PCB can support up to two RECORD microcontroller systems. Flyback diodes are included on-board as protection circuitry to prevent damage to upstream electronic elements, specifically the microcontroller. Two optional voltage dividers are included to lower the voltage of input TTL signals that surpass the microcontrollers safe operating range. In addition, we included optional on-board LED indicators which can be wired to activate when a TTL signal is sent or received by the microcontroller for debugging purposes. Two types of connectors exist on the board; standard 2.54 mm board-to-board headers, used to interface with the microcontroller, and JST-XH connectors, which securely connect cable bundles to the PCB. Only through-hole components were utilized to allow different components to be employed. Gerber files and Autodesk Eagle (https://www.autodesk.com/products/eagle/overview) project files, along with connection diagrams can be found in our Github repository (https://github.com/rjibanezalcala/RECORD/tree/main/pcb/Revision%201.0).

## Simple cable assemblies carry signals between electronic elements

To distribute the numerous electronic signals needed to coordinate cost/benefit conflict decision-making tasks, a variety of specialized cable assemblies were designed and employed. All TTL signals are carried through two types of hardware synchronization cables (HSc): the Noldus HSc (N-HSc) and the Inscopix HSc (I-HSc), both are two conductor cables that attach to the PCB via a 2-pin female JST-XH connector. The same cable design as the N-HSc was used to make a power cable (Pc) to feed the relay switches through the PCB from an external power source. Finally, valve cables (Vc), and LED cables (Lc) were created to carry the relay-driving signals and the PWM signals for the lights, respectively.

The N-HSc carries signals to and from the microcontroller through a Cat-5 cable assembly. The first JST-XH pin connects directly to the "TTL input 1" terminal, and the second pin connects to "TTL output 1". The input pin sends TTL signals from the microcontroller to the EthoVision XT USB-IO box, whereas the output pin sends the signals back to the microcontroller. Similarly, the I-HSc carries TTL signals to and from the Inscopix nVista Data Acquisition (DAQ) box. This cable is composed of a coaxial cable, which includes only one live wire paired with grounding shielding around the body of the cable. The live coaxial wire was connected to pin 1 of the JST-XH connector and jumped the shielding wire directly to the ground. This can be repeated for a second coaxial cable connected to the second pin of the JST-XH connector. The Pc uses the same design as the N-HSc, with two conductors (live and ground), a 2-pin JST-XH connector for connection to the PCB, and an interface for an external power source.

Both the Vc and Lc are very similar in the way they are made. Both consist of multi-conductor cables with either a 5-pin (Vc) or a 6-pin (Lc) JST-XH female connector at the end. The Vc has 4 live wires that are connected to one connector pin each, while all ground wires are connected and soldered to the remaining connector pin. Similarly, the Lc cable is made in the same way but has 5 live wires and one shared ground wire. On the other end of both cables, 2-pin connectors are used to interface with their respective components. Valve and LED cables interface with their respective components via one live pin and a ground pin. Both cables can be detached from their respective devices in case servicing or replacing any component is required.

A guide on the construction of these components is provided in our GitHub repository (https://github.com/rjibanezalcala/RECORD).

## Spatial tracking for task execution

The RECORD system leverages contour-based animal behavior tracking to locate each rat in the arena throughout behavioral trials. Behavioral tasks were executed based on animal location such as whether a reward should be dispensed or not and spatiotemporal information was used for behavioral analysis. Animal location can also be used to program trial-based task

"batteries" tailored to study the dynamics of decision-making in the RECORD environment.

Data from camera-driven spatial tracking was saved as raw data in table file format. Spatial data was sampled for every frame captured by the camera, allowing for sampling rate adjustment limited only by hardware. This information was collected constantly during behavioral experiments. No additional sensors are needed for RECORD, making the task as simple and intuitive as possible. Additionally, infrared (IR) lighting was used with IR-sensitive cameras to conduct experiments in dim rooms since light was used as an aversive stimulus and rodents are more active in low-light conditions[60].

In this manuscript, the RECORD system is used with two separate setups: Ethovision XT to track animal location and react to its position in an arena. The other uses the Bonsai computer vision library to continuously record the animal's position. We chose these programs because they allow the creation of modular behavioral routines to build trials that implement the paradigms described in a later section. Since Ethovision requires a paid license, we also tested the system with a free alternative, Bonsai. Also of great importance is the ability to link the behavior-tracking software with our microcontroller-driven electronics. For this purpose, a series of short Windows Batch scripts were created to make a call to Plink, a command line executable for serial communications. This leverages Ethovision's ability to call executables and Batch scripts and creates the crucial bridge between tracking software and microcontroller hardware. We have also created RECORD-lib, a custom Python library specifically designed to script task batteries and interface with RECORD, making our system flexible to be used in conjunction with many of the available open-source software packages for animal tracking and behavior analysis.

Additionally, novel methods for tracking animal movement and biomechanics including but not limited to B-SOiD (https://github.com/YttriLab/B-SOID), DeepLabCut (https://github.com/DeepLabCut/DeepLabCut), and DeepLabStream (https://github.com/SchwarzNeuroconLab/DeepLabStream) have been developed to study animal behavior[51,53,57,61], thus a Python interface for RECORD was appropriate for integration with any of these software packages.

Ethovision experiment files (https://github.com/rjibanezalcala/RECORD/tree/main/ethovision_experiments), Batch scripts (https://github.com/rjibanezalcala/RECORD/tree/main/microcontroller/batch_scripts), Bonsai workflows (https://github.com/rjibanezalcala/RECORD/tree/main/bonsai_workflows), and the RECORD-lib package with code examples and documentation (https://github.com/rjibanezalcala/RECORD/tree/main/python/RECORD-lib) are available to download from our GitHub repository.

## Data preprocessing and storage
All data generated by both RECORD setups was parsed and saved in a remote PostgreSQL database for long-term storage and analysis. The behavioral setup produces one XLXS Microsoft Excel file per trial with various worksheets containing positional, hardware, and trial event data. In contrast, because the Inscopix setup includes RECORD-lib, Bonsai, and the nVista DAQ box all linked together, it produces three separate comma-separated value (CSV) files, each containing trial, positional, and calcium cell trace data respectively, along with an additional text file containing trial metadata.

Within one PostgreSQL database, four different database tables were used for storage; one to store data from our purely behavioral RECORD setup (behavior table), and three for data from our Inscopix RECORD setup (optogenetics table). Behavioral parameters were defined and tracked across all arenas, with each of them being assigned to a column in its respective table. Data was then searched and recalled from the database using PostgreSQL queries for further analysis.

To ensure that the data produced during behavioral tasks is accurately catalogued into the database, a custom MATLAB parser was developed to simultaneously track events and variables recorded in each RECORD arena. The parser sorts these events and variables by trail and arena from which the data was gathered. The parser is accessed through a software termed

Serendipity, an intuitive graphical user interface (GUI) that aims to minimize human error when uploading raw data collected from each animal during behavioral sessions (Fig. 1e, and Extended Fig. 1d). This app allows for multiple concurrent users to navigate a PostgreSQL database and to extract data for further analysis. It consists of three GUIs that rely on each other to perform the complex tasks of a) connecting to the PostgreSQL database, b) parsing and sorting raw data, and c) analyzing data that is already in the PostgreSQL database. These three elements of the app can be found in our GitHub repository (https://github.com/lddavila/UTEP-Brain-Computation-Lab-Remote-Databases-and-Serendipity-App/tree/main/App%20Deployment%20Folder).

Additionally, we created SerendiPYty, a Python library to parse, link, and upload data generated from the RECORD-Bonsai-Inscopix setup used for in vivo calcium imaging. This parser takes the four separate files and additional output files associated with the nVista DAQ box, links the data together using timestamps and then stores the organized data in a PostgreSQL database.

Guides are included on how to set up a PostgreSQL database, how to link the database to MATLAB, and injecting or retrieving data from the database, in our GitHub repository (https://github.com/lddavila/UTEP-Brain-Computation-Lab-Remote-Databases-and-Serendipity-App/blob/main/Supplemental%20Node%204%20Database.docx). Additionally, the SerendiPYty library, code examples, and documentation have also been included in this repository (https://github.com/rjibanezalcala/RECORD/tree/main/python/SerendiPYty).

## Adaptable decision-making task batteries
The behavioral tasks used in this manuscript are implemented through a series of trials that last for 30–50 s each (Fig. 1f, and Supplemental Fig. 1e). Because task variables can be extensively altered, the experimenter may modify certain cost, reward, and time variables such as implementing delays, and time spent calculating the position of the animal before taking an action. Doing so allows fine-tuning of the task structure to address any confounding animal behaviors observed during the trials.

Three types of tasks were used most in this manuscript, each with a different decision-making paradigm, but all following the same general structure. The first is the Reward/cost association task (see the section "Reward/cost association task"), where the animal is trained to associate light cues with reward, and reward with a spatial location differentiated by the four different floor patterns (see the section "Arena floor pattern layout"). This task lacks a choice action which may differentially activate multiple brain circuits, thus the high-cost cost–benefit and low-cost cost–benefit tasks (see the section "High and low-cost cost–benefit") were also designed, where a different reward/cost pairing was offered on each trial. The main difference between the *high*-cost and *low*-cost variations of this task is what cost level is assigned to the cost/reward offered over a set of *N* trials (which will be referred to as a "session" hereafter). It is important to note that there is no inherent sequential nature between the three tasks, rather each one is self-contained as a separate (and substantial) task all on its own. The association task may be used as a steppingstone toward the high- and low-cost cost–benefit tasks; however it is not a requirement. Generally, rats took around 9 weeks to complete training for the different tasks (Supplemental Fig. 1h).

These three tasks can also be applied to study how conditions such as substance use disorder (SUD), and hunger affect the decisions of rats, and how these differences are encoded in brain circuits. As a behavioral model for this, rats were exposed to variables to induce said conditions as a part of this manuscript. Because the RECORD system uses food as an incentive, it may be attractive to motivate animals to participate in the trials by means of food deprivation. For this reason, the effect of food deprivation was assessed by running high- and low-cost cost-benefit tasks after a period of food deprivation (see the section "Food deprivation impacts decision-making"). Also, SUD effects on behavior compared to baseline behavior were assessed. Using the same three tasks, an alcohol tradeoff task was created where ethanol was mixed into the sucrose reward (see the section "Sucrose vs.

alcohol trade-off"). A slightly different behavioral protocol was used for rats who underwent oxycodone self-administration (see the section "Oxycodone self-administration and abstinence").

Rats were habituated to human experimenters, the arenas and sucrose rewards, and the testing environment (arena, environmental sounds, light conditions, etc.) for 3–5 days. After two weeks, rats then continued to the association task.

We describe the protocol we followed to run our tasks using EthoVision in the documentation available in our RECORD repository (see Supplemental Note 6). (https://github.com/rjibanezalcala/RECORD/blob/main/documentation/noldus_ethovision_behavioural_protocol.pdf).

### Reward/cost association task

The reward/cost association task uses a combination of light and delivery of different concentrations of sucrose solutions to condition the animal to associate a reward to a location in the arena. In this task, light is not used as a cost, rather it is presented at very low intensity and only used to condition the rodent to associate the light and location of a reward. A reward is then delivered regardless of whether the animal approaches the light.

The task has three stages: (1) Tone Presentation, where a 4 s trial start tone (https://github.com/rjibanezalcala/RECORD/blob/main/trial_start_tone.wav) is played. This has the double purpose of both conditioning the rodent to the trial tone, and to signal the start of a trial. At the next stage, (2) the Cue Presentation stage, low-intensity light is presented at a predetermined location; the locations vary pseudo-randomly across trials. It is critical that the light is bright enough for the rodent to see, but not so bright that it stresses the animal; a range of 15–20 lx is used in the current study for this purpose. Finally, (3) the Reward Delivery, where a sucrose reward at a specific concentration (determined by the location of the feeder) is given to the rodent.

A total of 40 trials were run daily for 5 days a week over the span of 2–4 weeks until the rodents had learned the task. In this publication, male rats learned the task at ~2 weeks, and female rats took ~1 week. In addition, this task allowed for the identification of each rat's individual reward preferences based on approach rate vs. avoidance rate. Documentation for this reward/cost association task can be found in the Ethovision Experiments repository (https://github.com/rjibanezalcala/RECORD/tree/main/ethovision_experiments). Equal gender Long evans rats at age 10 weeks were utilized for the purposes of training. After training, animals were utilized for up to 1.5 years to complete the tasks outlined in this manuscript. We have complied with all relevant ethical regulations for animal use.

### High and low-cost cost–benefit

First, the trial start tone is presented (https://github.com/rjibanezalcala/RECORD/blob/main/trial_start_tone.wav). The offer location is signaled by turning the LEDs for that feeder on. The cost level at which this light is turned on is selected as CLv1 (cost-level 1) in the case of the low-cost cost-benefit variation of this task or selected at random between CLv1 and CLv3 in the case of the high-cost variation. CLv1 and CLv3 were selected with a uniform distribution (50% CLv1 and 50% CLv3), the distribution could be altered by the experimenter. After the offer is presented, the animal must decide whether to engage with the offer or not. To engage, the animal must approach the offer, stepping into the corresponding arena quadrant; if the animal stays outside of this quadrant, the trial is recorded as a 'reject' trial, and the light is turned off. After a variable amount of time (in this manuscript, 6 s) the system uses camera-based spatial tracking to determine whether the animal is within the quadrant where the offer was presented or is in another region. Reward is only dispensed if the animal lingers in the active quadrant or active region of interest (aROI). Finally, an inter-trial interval is applied at the beginning of each trial in which a rat waits until the next trial is started (in this manuscript, 28 s, Supplemental Fig. 1e, Supplemental Note 1).

Adjustments were made to cost-light intensity as there was high variability among individual rats; some seemed to perceive the light as stronger or weaker than the others, despite brightness being the same across

each cost level. Both high- and low-cost cost–benefit tasks can be found in our Ethovision Experiments repository (https://github.com/rjibanezalcala/RECORD/tree/main/ethovision_experiments).

### Food deprivation alters decision-making

Aberrant decision-making and behavioral changes were assessed in each rat through the implementation of a food deprivation paradigm. While running cost/benefit conflict tasks, rats were waned off their regular meals and weighed daily. Rats were initially given 20% of their body weight in daily food, then food availability was gradually reduced to only 5 g per rat per day over the course of three weeks. For each rat, every session was categorized in one of two ways: one where a weight reduction of 0−10% of their initial weight was observed, and another where a 10−20% weight reduction was observed. Food deprivation was halted after rats had lost around 20% of their pre-food deprivation weight, which occurred after about 3 weeks.

### Sucrose vs. alcohol trade-off

Animals ran high- and low-cost cost–benefit tasks for 9 weeks where cost levels presented during the task were rotated between level 1 (low-cost) only, level 1/level 3, and level 3 only (high-cost) five days a week. Animals were subjected to a sucrose vs. alcohol trade-off task based on the same cost-benefit tasks where different concentrations of alcohol were added to the existing levels of sucrose solutions. The amount of alcohol per solution increased as sugar concentration decreased (9% sucrose + 1% alcohol, 5% sucrose + 4% alcohol, 2% + 10% alcohol, and 0.5% sucrose + 20% alcohol).

### Oxycodone self-administration and abstinence

The self-administration task is based on one used in multiple published experiments[62–68]. Mildly water-deprived rats self-administered either oxycodone or water reward (with yoked saline) 6 h/day for 14 days, as described previously[19]. During each trial of the task, a nosepoke aperture was illuminated. Nosepokes into the illuminated aperture resulted in a bolus of oxycodone (0.05 mg/kg/infusion) or water/yoked saline (volume matched) coupled with a 20 s tone/house light stimulus. During the 20 s tone/house light stimulus, the nosepoke aperture was darkened and further nosepokes were recorded but did not result in drug delivery. Animals were also tested on extinction of self-administration both 1 day and 30 days following cessation of self-administration; this paradigm was sufficient to induce an 'incubation of craving' for oxycodone, that is, an increase in drug-seeking behavior at 30 days[21]. During extinction, nosepokes resulted in tone/house light stimulus, but no oxycodone or water/yoked saline.

### Oxycodone behavioral task

During RECORD behavioral sessions, rats performed a slightly modified version of the cost/benefit association task with the same rewards being dispensed, however only 280 lx was used as a cost as opposed to varying light intensities. This modification to the protocol was made due to a perceived hypersensitivity to the cost light, which we believed was due to the introduction of oxycodone. Additionally, we found that during self-administration and abstinence, some rats became increasingly aggressive toward experimenters and spent a large amount of time biting at the arena components instead of participating in the trial. These rats (n = 2) were removed from the study entirely.

### Spatiotemporal behavioral dynamics

To analyze individual decision-making strategies, a set of "features" were defined and extracted from data generated during behavioral trials (e.g., speed, orientation, position, etc.). After analysis, these features were recorded and stored in the PostgreSQL database in a separate table, then a psychometric function was plotted, with all four levels of reward along the $x$-axis and the feature tracked along the $y$-axis. The following is a list of the features extracted along with a definition of each one (Fig. 1e).

1. *Distance traveled*: Overall Euclidean distance traveled by the animal in the normalized trajectory (https://github.com/atanugiri/Feature-

Extraction/tree/main/Run%20Time), i.e. the two-dimensional plane created by the arena floor normalized to the camera's field of view.

2. *Travel pixel*: The number of pixels that were traveled by the animal between trial start and trial end in the normalized trajectory (https://github.com/atanugiri/Feature-Extraction/tree/main/Trajectory%20Plots, https://github.com/atanugiri/Feature-Extraction/tree/main/Run%20Time).

3. *Proportion of high-speed runs (bigaccelerationperunittravel)*: The total number of outliers present in a set of acceleration measurements. We calculated this based on the median and the standard deviation of the acceleration data divided by the distance traveled (https://github.com/atanugiri/Feature-Extraction/tree/main/Acceleration%20and%20Jerk%20Outliers).

4. *Stopping points*: Defined as the number of times the rat comes to a complete stop (moves < 0.1 units in both *X* and *Y* direction within a 3-s window) during the trial (https://github.com/atanugiri/Feature-Extraction/tree/main/Stop%20Time).

5. *Rotation points*: Defined as the number of rotations performed by the animal during each trial. A vector was defined from the center point of the rat to the head of the rat. Angular changes >180° in the vector within a 1.5 s window were defined as one rotation (https://github.com/atanugiri/Feature-Extraction/tree/main/Rotation%20Points).

6. *Approach time*: Refers to the time it takes the animal to reach an offer location after the "trial start" tone is presented (https://github.com/atanugiri/Feature-Extraction/tree/main/Run%20Time).

7. *Proportion of trial outside all reward zones*: Time the animal spent in the center of the arena divided by the number of trials in a session (https://github.com/atanugiri/Feature-Extraction/tree/main/Passing%20Central%20Zone).

All Matlab codes used for this analysis are available in the "feature extraction" Github repository (https://github.com/atanugiri/Feature-Extraction). Within the different functions used to extract each feature can be found.

## Neuroeconomic modeling of decision-making

The data collected by the RECORD system can be distilled into three features important for a neuroeconomic understanding of the animal's decision-making patterns: 1. valuation, the subjective value placed on reward and cost by individuals (top panel, revealed by parameter in Eqs. (1) and (2), green = high valuation of cost or rewards and purple = low valuation); 2. elasticity, the degree to which an animal changes its decision based on small changes to reward or cost (middle panel, parameter, green = low elasticity and purple = high elasticity); and 3. sensitivity, the responsiveness of the animal to the given levels of reward and cost (bottom panels, MSE of the fit, gray shaded region covers the curves of fit produced by the 80% confidence interval of parameter).

Psychometric functions are used to estimate the probability of approaching ($f(R)$ or $f(C)$) based on the independent variables of reward and cost presented to the animal. In plots, all fits optimize for least squares. For simplicity, data in plots were fit to levels (1–4) of the reward and cost concentrations.

Equation (1) (for the reward sigmoid functions) and 2 (for the cost sigmoid functions) are used in Fig. 3a. These each contain two parameters of fit: *a* and *b*, separately fitted to each set of points. The fit is a classical logistic function, where *b* is involved in the steepness of the sigmoid and *a* is involved in its shift along the *x*-axis. In Eq. (1), it is assumed that cost is held constant at some level, while in Eq. (2), reward is held constant at a given level.

Equation (3) is used for Fig. 3b, c, and Supplemental Fig. 3b–d. Here, we combine the sigmoidal fits in 1 and 2 to produce a single function relating any input level of reward and cost to the probability of approaching. When implemented, this equation is fitted to the grid of an animal's probability of approach data at each level of cost and reward. The boundary where the animal is expected to approach and avoid each 50% of the time can be

extracted (Eq. (4)) by setting the left-hand side of Eq. (3) to 0.5 and solving for reward in terms of cost.

Applying concepts of neuroeconomics to Eq. (1), we can replace the independent variable *R* with a $g(R)$ function. This substitution acknowledges that reward and utility may not have a linear match and that another function may better predict an animal's propensity to approach. Many possible $g(R)$ functions could be chosen here. We focus on simplicity to avoid overfitting. $g\prime(R)$ is set to begin at 1 (i.e., utility increases as quickly as reward concentration does) and then decrease at a rate proportional to *R* (see Eq. (7)); as *R* increases, due to diminishing utility returns of *R*, utility $g\prime(R)$ also decreases. Integrating, we get Eq. (8). The $-\frac{c_1}{2}$ term becomes *c*, and we make the logical assumption that utility is 0 when reward is not present.

$$f(R) = \frac{1}{1 + e^{-a_R*R+b_R}}, \tag{1}$$

$$f(C) = \frac{1}{1 + e^{a_C*C+b_C}}, \tag{2}$$

$$f(R, C) = \frac{1}{1 + e^{-a_R*R+b_R}} * \frac{1}{1 + e^{a_C*C+b_C}}, \tag{3}$$

$$f(R) = \ln\left(\frac{1}{e^{\frac{1-R}{2}+\frac{1}{2}}} - 1\right) - 1, \tag{4}$$

$$f(R) = \frac{1}{1 + e^{-a_R*g(R)+b_R}}, \tag{5}$$

$$g(R) = c * \frac{R^2}{2} + R, \tag{6}$$

$$g\prime(R) = 1 - c_1 * R, \tag{7}$$

$$g(R) = R - \frac{c_1}{2} * R^2 + c_2 \tag{8}$$

where *R* is reward level, *C* is cost level, $a_R$, $b_R$ are parameters to fit data where reward is incremented, $a_C$, $b_C$ are parameters to fit to data where cost is incremented, $g(R)$ is a utility function with reference to the amount of reward consumed, and *c* is a third parameter of fit introduced to Eq. (6). We have included all codes written for this analysis in our GitHub repository (https://github.com/rjibanezlcala/RECORD/tree/main/data_analysis/neuroeconomic_analysis).

## Surgical procedure

After successful training on the behavioral tasks, rats were injected with a calcium indicator virus, implanted with an Inscopix Gradient-Index lens (GRIN) and head-stage during a single surgery. All stereotaxic procedures were conducted on a Kopf stereotaxic instrument (Model 942). Flat skull position was obtained by matching the height of bregma and lambda along the same sagittal axis. All stereotaxic coordinates were made in relation to Bregma.

## Anesthesia and analgesia

For all surgical procedures, rats were anesthetized using an isoflurane anesthesia system (induction: 3%, maintenance 2%, oxygen flow rate 2 L/min). Meloxicam (1 mg/kg. SQ) and Enrofloxacin 2.27% (5 mg/kg, SQ) were mixed in 10 mL of Ringer's solution and administered (SQ) for 3 days post-surgical procedure, including the day of surgery. Topical antibiotic ointment and 4% Chlorhexidine were applied to the incision daily.

Following anesthesia administration, their heads were shaved, and the animals were situated into the stereotaxic frame by fixing their heads between ear bars. An iodine solution was applied to the surgical site, and Puralube ophthalmic ointment (MWI cat#027505) was applied to the eyes.

A heating pad was used to maintain core body temperature at 37 °C throughout the procedure.

**Bilateral viral injections**. Prior to injection, the stereotaxic coordinates for each injection site and the GRIN lens implant were marked on the skull surface using a stereotaxic manipulator. Five stainless steel screws were inserted epidurally surrounding the marked coordinates to provide stability for a head-mounted miniscope holder.

The calcium indicator AVV1-syn-jGCaMP8f-WPRE virus (Addgene, Cat# 162376-AVV1) was injected bilaterally into the dorsal striatum (ML ± 1.85, AP + 0.7, DV − 3.3, −4.3) via a 33-G needle attached to a Hamilton syringe. A craniotomy was performed followed by the removal of dura mater, allowing access for virus injection and lens implantation. Normal saline solution was used to clean the craniotomy holes and maintain pressure while ensuring that there would be no bleeding before starting the virus injection. To inject the virus, the needle was inserted and lowered to −4.4 from the dura mater, then raised to ensure that the needle remained free from obstruction. The needle was re-inserted and lowered to the first injection site (−4.3 from dura mater), and 600 nL of virus was administered/injected at a rate of 0.1 μL/min. After each injection, the needle remained in place for 6–10 min to allow the virus to fully disperse, then raised (1 mm/min) to check for any obstructions. If clear, the needle was reinserted to the second injection depth of −3.3, and 400 nL of virus was injected (0.1 μL/min), again allowing 6–10 min before raising the needle to allow full virus dispersal.

**Unilateral GRIN lens implant**. Once the viral injections were completed, the GRIN lens was implanted (Inscopix, ProView Integrated Lenses 1.0 mm × 9.0 mm, cat# 1050-00416). An 18-G needle was lowered past the corpus callosum to create a tract for the GRIN lens implant. To ensure the striatum was reached, a surgical microscope was used to observe the distinction in brain morphology during tract creation. After reaching the striatum (grey matter), exposed brain tissue was irrigated with sterile saline to maintain visibility. Using a 3D printed lens holder, the GRIN lens was lowered to 200 nm dorsal to the injection coordinates (speed: 200 nm/min, coordinates: ML + 1.85, AP + 0.7, DV − 4.1).

Once the lens reached the desired coordinates, craniotomy sites were sealed with a silicone adhesive (Kwik-Sil™, World Precision Instruments). The integrated headcap and five screws were then fixed to the skull using dental cement and allowed to cure for 5 min If needed, 5-0 sutures were applied, and the entire area was covered with topical ointments. Animals were given 7 days of recovery time.

**Freely moving calcium imaging**

Recordings began to be conducted 6–8 weeks after the virus injection. Neuronal activity was recorded using a commercially available miniaturized epifluorescence microscope and the nVoke 2.0 system (Inscopix, Inc., Palo Alto, CA). The miniscope docks to the baseplate-integrated lens implanted in the brain and calcium activity are sent to the Data AcQuisition system (DAQ). Recording parameters were all controlled using the Inscopix Data Acquisition Software (IDAS; Inscopix, Inc.) running on a web browser. Once dynamic calcium imaging was observed, recording parameters were maintained for all following recordings to consistently image the same cells and aid in longitudinal tracking of registered cells (frame rate: 20Hz, LED power: 1.6, Sensor Gain: 5.5, Electronic Focus: 360).

Combined imaging and behavior sessions began by holding the animal to allow for the removal of the baseplate-cover, followed by docking and securing the miniscope to the baseplate by tightening a set screw on the side. For each imaging behavioral session, the animal's motor behavior was recorded using the RECORD-Inscopix system setup (see the section "Spatial tracking for task execution"), while the calcium activity was monitored and recorded by TTL-triggered IDAS. The miniscope's excitation LED (wavelength: 455 nm) was activated by a TTL pulse synchronized to the behavioral task and remained on for the duration of each trial. A typical imaging and behavior session included 25 trials, with calcium imaging lasting a combined 6–15 min The session concluded with holding the animal to remove the miniscope, replacing the baseplate-cover, and securing it with the set screw.

**In vivo calcium imaging processing**

**Recordings preprocessing**. Inscopix Data Processing software (IDPS; Inscopix, Inc.) was used to process calcium imaging recordings. The processing workflow included steps to correct pixels, crop the field of view (FOV), and spatially downsample (2×) imaging frames to reduce the size of the data while maintaining an acceptable resolution (1280 × 800 pixels before downsampling and 590 × 350 pixels after downsampling). A Gaussian filter (cut-offs set to low: 0.005 pixel$^{-1}$, high: 0.5 pixel$^{-1}$) was then applied, followed by a motion correction step to account for and stabilize any brain movement artifacts associated with movement in the maze, by minimizing the differences between each frame and a "mean" reference frame. Normalization of the initial activation of calcium activity was accomplished by applying a Delta F/F filter. Neurons were identified using PCA/ICA analysis through their spatial footprint (cell's shape and location in the FOV, Fig. 1g). Identified cells were manually accepted, with possible cells with non-cell-like shape or non-calcium-like events discarded. Accepted cells and their activity traces were registered and exported to MATLAB (MathWorks, MA) for further analysis.

**Cell Maps and Longitudinal tracking**. Combined calcium imaging and behavior sessions allowed for the tracking and correlation of registered cells across multiple different sessions and task types. Using the enhanced correlation coefficient (ECC) image registration algorithm on IDPS, cell sets from several recording sessions were compared. Cell map alignment and cell matching were performed to identify the same cell in different recordings based on the similarity of the cell's spatial footprint. A global cell set was established as a reference session. Subsequent cell maps and cell sets were then compared to the established global cell set (Supplemental Fig. 1i, j).

**Cell activity analysis**

After calcium trace recordings were processed, the cell trace data was parsed using the SerendiPYty library and stored in the database (see the section "Data preprocessing and storage"). During analysis, calcium intensity ($\Delta F/F$) was arranged by utility (reward subtracted from cost, see the section "Neuroeconomic modeling of decision-making", Eq. (8)), or by reward level to evaluate the valuation of reward in terms of neuronal activity. A paired $T$-test was performed to evaluate how different cost levels affected neuronal activity. Additionally, an ANOVA repeated measure was performed to evaluate the significance of the effect of reward levels and utility neuronal activation. The code for this calcium trace analysis is available in our cell activity analysis GitHub repository (https://github.com/lrakocev/inscopix).

**Psychometric function shape analysis**

After plotting the psychometric functions, some took the shape of parabolic functions, others were shaped like sigmoidal functions, and some had no shape. Four different function models were used to classify the psychometric functions and ran a shape analysis was to fit them into one of the models. We chose three variations of the traditional sigmoid formula and a parabolic function model. The sigmoidal models are described by the three equations below:

$$f(x) = \frac{1}{1 + (b \cdot (e^{-cx}))}, f(x) = \frac{a}{1 + (b \cdot (e^{-cx}))}, \text{and } f(x)$$
$$= \frac{a}{1 + (b \cdot (e^{-c(x-d)}))}$$

where $a$ corresponds to the height of each sigmoid, $b$ and $d$ both correspond to the left and right shift of each sigmoid, respectively, and $c$ corresponds to steepness. The parabolic model followed the classical formula for a parabolic

equation, seen below:

$$f(x) = a \cdot (x - b)^2 + c$$

in this case, $a$ is the slope of the parabola, $b$ represents the horizontal shift of the function, and $c$ the vertical shift.

To fit psychometric functions to the models described above, we considered the coefficient of determination ($R^2$, ranging from 0 to 1 where 1 signifies the best fit possible) as a determining factor for shape analysis. For each sigmoidal model, an $R^2$ value was calculated, and any psychometric function whose $R^2$ resulted in a value of 0.4 or higher was considered to fit the model sufficiently. In the case that one psychometric function fits multiple sigmoidal models, it was sorted into an additional three-parameter sigmoid category. Psychometric functions that did not meet the threshold requirement were then fit to a parabola. If a function had an $r^2 < 0.4$ for either fit, it was determined to be undefined. All the parameters were stored in our PostgreSQL database for further processing. All code is contained within our data analysis GitHub repository (https://github.com/lddavila/UTEP-Brain-Computation-Lab-Remote-Databases-and-Serendipity-App/tree/main/Updated%20Analysis). Each folder within this directory contains both the code for our shape analysis and out classification analysis (see the section "Classifying decision-making patterns").

## Classifying decision-making patterns

Psychometric functions were characterized by maximum (highest point), steepness (slope), and shift (where on the $x$-axis a positive slope began). By comparing maximum vs. steepness, maximum vs. shift, or shift vs. steepness, discrete clusters could be identified using the fuzzy c-means clustering algorithm. To then derive the exact number of clusters formed by this analysis, clusters were partitioned until the highest possible modified partition coefficient (MPC) was achieved. MPC values range from 0 to 1 with values closer to 1 being preferred. During analysis, MPC scores were 0.8 or greater, demonstrating statistically distinct clusters.

No clusters were identified when analyzing parabolic-shaped functions. All analysis can be reproduced using the code provided in our data analysis GitHub repository (https://github.com/lddavila/UTEP-Brain-Computation-Lab-Remote-Databases-and-Serendipity-App/tree/main/Data%20Analysis). Each folder within this directory contains both the code for shape and classification analysis (see the section "Psychometric function shape analysis").

Clustering psychometric features provides a form of analysis that can parametrize the behavioral tendencies of each rat performing RECORD behavioral tasks. One method of cluster analysis compares the number of sessions a rat appeared within a cluster to the total number of sessions that rat ran. This allows for the direct comparison and analysis of individual differences in behavioral strategies exhibited during decision-making. Differences between Euclidean distances calculated for each rat found were normally distributed across the population of rats running RECORD tasks. This data was considered a control group and compared to the same rat's food deprivation, alcohol, and oxycodone datasets. These data showed a change in the probabilities calculated for each rat, revealing a change in decision-making behavior from baseline to both food deprivation. These probability tables can be created using the code contained in our probability table GitHub repository (https://github.com/lddavila/UTEP-Brain-Computation-Lab-Remote-Databases-and-Serendipity-App/tree/main/Updated%20Analysis).

## Statistics and reproducibility

A repeated measures analysis of variance (ANOVA) was conducted using the MATLAB ranova function to examine the effects of within-subject factors, such as sucrose concentration, and between-subject factors, including Sex and experimental conditions (control vs food deprivation). Additionally, pair-wise comparisons were conducted to further explore the differences between groups using a post-hoc analysis, specifically Tukey's

honestly significant difference method, implemented with the MATLAB multcompare function. To assess the between-subjects differences, a two-sample Kolmogorov-Smirnov test was also employed with the MATLAB kstest2 function. Three-way ANOVA was used to investigate the main effects of each of the three factors (e.g., sucrose concentration, sex, treatment groups) and any potential interactions between them.

The scripts for statistical analysis are located in our Github repository: https://GitHub.com/atanugiri/Data-Analysis/tree/main/Statistics.

A detailed explanation of the statistical analyses reported in each figure is provided below.

**Figure 2**

**Figure 2a: Control approach rate (FvM)**

Statistical significance was determined by Repeated measures analysis of variance. (Female $N = 12$, Male, $N = 11$). Effect of concentration: d.f. = 3, $F = 118.5268$, $p = 7.3397e-26$. Effect of sex: d.f. = 1, $F = 3.8265$, $p = 0.0639$. kstest2 results: $h = 0$, $p = 8.2894e-02$, ks2stat = 0.2557 (overall sex difference).

Post-hoc analysis: 0.5%: $p = 0.5994$, 2%: $p = 0.0203$, 5%: $p = 0.1014$, 9%: $p = 0.5338$. KStest2 and Wilcoxon rank sum test Results (complementary to post-hoc analysis). *KStest2: Conc*1: $h = 0$, $p = 0.3032$, ks2stat = 0.3788. RStest: Conc1: $h = 0$, $p = 0.4049$. KStest: Conc: $h = 1$, $p = 0.0087$, ks2stat = 0.6439. RStest: Conc2: $h = 1$, $p = 0.0187$. KStest2: Conc3: $h = 0$, $p = 0.2812$, ks2stat = 0.3864. RStest: Conc3: $h = 0$, $p = 0.1314$. KStest2: Conc4: $h = 0$, $p = 0.9465$, ks2stat = 0.2045. RStest: Conc4: $h = 0$, $p = 0.5156$.

**Figure 2b: Control Effect of cost on Approach rate (FvM)**

Statistical significance was determined by repeated measures analysis of variance. (Female $N = 12$, Male $N = 11$). $p$-value for concentration: 2.061e−10. $p$-value for sex: 0.15301. kstest2 results: $h = 0$, $p = 9.9819e-02$, ks2stat = 0.2481 (overall sex difference).

Post-hoc analysis: 240 lx: 1.8263e−01, 260 lx: 5.1534e−02, 290 lx: 8.8968e−01, 320 lx: 3.7194e−01.

KStest2 and Wilcoxon rank sum test Results (complementary to post-hoc analysis): KStest2: Conc1: $h = 0$, $p = 0.5833$, ks2stat = 0.3030. RStest: Conc1: $h = 0$, p = 0.2815. KStest2: Conc2: $h = 0$, $p = 0.2407$, ks2stat = 0.4015. RStest: Conc2: $h = 0$, $p = 0.0602$. KStest2: Conc2: $h = 0$, $p = 0.6484$, ks2stat = 0.2879. RStest: Conc2: $h = 0$, $p = 1.0000$. KStest2: Conc4: $h = 0$, $p = 0.7136$, ks2stat = 0.2727. RStest: Conc4: $h = 0$, $p = 0.4235$.

**Figure 2c: Bayesian analysis of cost**

Statistical significance was determined using the Statistical Package for the Social Sciences (SPSS) package ($F = 12$, $M = 9$) $p$-value for concentration: <0.0001.

Sex differences across all concentrations: $p = 0.8$.

Post-hoc analysis: 15lx%: $p = 0.000627$, 240lx%: $p < 0.0001893$, 260lx%: $p < 0.0001658$, 290lx%: $p < 0.2045$, 320lx%: $p = 0.405$.

**Figure 2d: Control Distance traveled (FvM)**

Statistical significance was determined by Repeated measures analysis of variance. (Female $N < 12$, Male $N < 11$).

Effect of concentration: d.f. = 3, $F = 2.2699$, $p = 8.9008e-02$.

Effect of sex: d.f. = 1, $F = 11.8146$, $p = 0.0025$. kstest2 results: $h = 1$, $p = 9.4199e-06$, ks2stat = 0.5019 (overall sex difference).

Post-hoc analysis: 0.5%: 1.2707e−03, 2%: 1.1033e−02, 5%: 3.5299e−03, 9%: 8.9459e−03.

KStest2 and Wilcoxon rank sum test Results (complementary to post-hoc analysis): *KStest2: Conc1:* $h = 1$, $p = 0.0258$, ks2stat = 0.5758. RStest: Conc1: $h = 1$, $p = 0.0051$, zval = 2.8003. KStest2: Conc2: h = 1, $p = 0.0361$, ks2stat = 0.5530. RStest: Conc2: h = 1, $p = 0.0151$, zval = 2.4311. KStest2: Conc2: h = 1, $p = 0.0230$, ks2stat = 0.5833. RStest: Conc2: h = 1, $p = 0.0051$, zval = 2.8003. KStest2: Conc4: $h = 1$, $p = 0.0361$, ks2stat = 0.5530. RStest: Conc4: $h = 1$, $p = 0.0062$, zval = 2.7388.

**Figure 2e: Control Number of high sp. runs (FvM)**

Statistical significance was determined by Repeated measures analysis of variance. (Female $N = 12$, Male $N = 11$). Effect of concentration: d.f. = 3, $F = 23.4392$, $p = 2.6239e-10$. Effect of sex: d.f. = 1, $F = 12.8352$, $p = 0.0018$.

kstest2 results: $h = 1$, $p = 3.0470e-05$, ks2stat = 0.4773 (overall sex difference).

Post-hoc analysis: 0.5%: 1.6629e−04, 2%: 2.5835e−03, 5%: 5.1037e−03, 9%: 6.0776e−02.

KStest2 and Wilcoxon rank sum test results (complementary to post-hoc analysis): KStest2: Conc1: $h = 1$, $p = 0.0003$, ks2stat = 0.8258. RStest: Conc1: $h = 1$, $p = 0.0006$, zval = −3.4158. KStest2: Conc2: $h = 1$, $p = 0.0059$, ks2stat = 0.6667. RStest: Conc2: $h = 1$, $p = 0.0042$, zval = −2.8619. KStest2: Conc2: $h = 1$, $p = 0.0323$, ks2stat = 0.5606. RStest: Conc2: $h = 1$, $p = 0.0106$, zval = −2.5541. KStest2: Conc4: $h = 0$, $p = 0.1213$, ks2stat = 0.4621. RStest: Conc4: $h = 0$, $p = 0.1481$, zval = −1.4463.

**Figure 2f: Control approach time (FvM)**

Statistical significance was determined by Repeated measures analysis of variance. (Female $N = 12$, Male $N = 11$). Effect of concentration: d.f. = 3, $F = 6.4355$, $p = 7.8859e−04$. Effect of sex: d.f. = 1, $F = 0.6365$, $p = 0.4348$. kstest2 results: $h = 0$, $p = 3.1096e−01$, ks2stat = 0.1986 (overall sex difference).

Post-hoc analysis: 0.5%: 8.9816e−01, 2%: 4.5069e−01, 5%: 5.3396e−01, 9%: 5.9227e−01.

KStest2 and Wilcoxon rank sum test Results (complementary to post-hoc analysis): KStest2: Conc1: $h = 0$, $p = 0.7358$, ks2stat = 0.2727. RStest: Conc1: $h = 0$, $p = 0.6458$, zval = −0.4597. KStest2: Conc2: $h = 0$, p = 0.4896, ks2stat = 0.3333. RStest: Conc2: $h = 0$, $p = 0.6682$, zval = −0.4286. KStest2: Conc2: $h = 0$, $p = 0.8286$, ks2stat = 0.2500. RStest: Conc2: $h = 0$, $p = 0.7169$, zval = −0.3627. KStest2: Conc4: $h = 0$, $p = 0.7136$, ks2stat = 0.2727. RStest: Conc4: $h = 0$, $p = 0.7350$, zval = −0.3385.

**Figure 2g: Control Prop. of trial out. all reward zones (FvM)**

Statistical significance was determined by Repeated measures analysis of variance. (Female $N = 12$, Male $N = 11$). Effect of concentration: d.f. = 3, $F = 14.3852$, $p = 3.0392e−07$. Effect of sex: d.f. = 1, $F = 1.5082$, $p = 0.2330$. kstest2 results: $h = 0$, $p = 3.3508e−01$, ks2stat = 0.1913 (overall sex difference).

Post-hoc analysis: 0.5%: 2.6980e−01, 2%: 7.5679e−01, 5%: 8.5789e−02, 9%: 3.0110e−01.

KStest2 and Wilcoxon rank sum test results (complementary to post-hoc analysis): KStest2: Conc1: $h = 0$, $p = 0.8067$, ks2stat = 0.2500. RStest: Conc1: $h = 0$, $p = 0.4044$, zval = 0.8338. KStest2: Conc2: $h = 0$, $p = 0.9982$, ks2stat = 0.1515. RStest: Conc2: $h = 0$, $p = 0.8292$, zval = 0.2157. KStest2: Conc2: $h = 0$, $p = 0.1006$, ks2stat = 0.4773. RStest: Conc2: $h = 1$, $p = 0.0483$, zval = 1.9743. KStest2: Conc4: $h = 0$, $p = 0.4595$, ks2stat = 0.3333. RStest: Conc4: $h = 0$, $p = 0.4219$, zval = 0.8031.

**Figure 2h: Control Number of stopping points (FvM)**

Statistical significance was determined by Repeated measures analysis of variance. (Female $N = 12$, Male $N = 11$). Effect of concentration: d.f. = 3, $F = 0.0544$, $p = 9.8312e−01$. Effect of sex: d.f. = 1, $F = 2.1682$, $p = 0.1557$.

kstest2 results: $h = 1$, $p = 2.5533e−05$, ks2stat = 0.4811 (overall sex difference) post-hoc analysis: 0.5%: 7.9690e−02, 2%: 2.5673e−01, 5%: 1.4691e−01, 9%: 2.0322e−01.

KStest2 and Wilcoxon rank sum test Results (complementary to post-hoc analysis): KStest2: Conc1: $h = 1$, $p = 0.0067$, ks2stat = 0.6591. RStest: Conc1: $h = 1$, $p = 0.0028$, zval = −2.9850. KStest2: Conc2: $h = 0$, $p = 0.2604$, ks2stat = 0.3939. RStest: Conc2: $h = 0$, $p = 0.1029$, zval = −1.6310. KStest2: Conc2: $h = 1$, $p = 0.0323$, ks2stat = 0.5606. RStest: Conc2: $h = 1$, $p = 0.0289$, zval = −2.1849. KStest2: Conc4: $h = 0$, $p = 0.0915$, ks2stat = 0.4848. RStest: Conc4: $h = 0$, $p = 0.0905$, zval = −1.6925.

**Figure 3**

**Figure 3e: Ca$^{2+}$ activity vs. utility**

Statistical significance was determined by one-way analysis of variance. (group 1 = 60, group 2 = 61, group 3 = 58, group 4 = 64, group 5 = 22, group 6 = 22, group 7 = 25, group 8 = 18) $p$-value for significance of difference between the groups (utility): 0.0429.

Post-hoc analysis by Tukey's HSD method: No group difference is statistically significant.

**Figure 3f: Ca$^{2+}$ activity at low cost**

Statistical significance was determined by one-way analysis of variance. (group 1 = 60, group 2 = 61, group 3 = 58, group 4 = 64) $p$-value for significance of difference between the groups (concentration): 0.9599. Post-hoc

analysis by Tukey's HSD method: No group difference is statistically significant.

**Figure 3g: Ca$^{2+}$ activity at high cost**

Statistical significance was determined by one-way analysis of variance. (group 1 = 22, group 2 = 22, group 3 = 25, group 4 = 18) $p$-value for significance of the difference between the groups (concentration): 0.5523. Post-hoc analysis by Tukey's HSD method: No group difference is statistically significant.

**Figure 3h: Ca$^{2+}$ activity low-cost vs. high-cost**

Statistical significance was determined by one-way analysis of variance. (group 1 = 243, group 2 = 87). $p$-value for significance of difference between the groups (concentration): 0.0012.

**Figure 5**

**Figure 5a: FD vs. control approach rate**

Statistical significance was determined by repeated measures analysis of variance (Control $N = 22$, FD $N = 22$).

Effect of concentration: d.f. = 3, $F = 281.8850$, $p = 1.0842e−55$. Effect of condition: d.f. = 1, $F = 19.0789$, $p = 0.0001$. kstest2 results: $h = 1$, $p = 1.0816e−02$, ks2stat = 0.2386 (overall difference in Control vs. FD). Post-hoc analysis: 0.5%: 6.8154e−01, 2%: 5.2118e−01, 5%: 5.0500e−04, 9%: 4.8848e−03.

KStest2 and Wilcoxon rank sum test Results (complementary to post-hoc analysis): KStest2: Conc1: $h = 0$, $p = 0.1746$, ks2stat = 0.3182. RStest: Conc1: $h = 0$, $p = 0.3038$, zval = 1.0283. KStest2: Conc2: $h = 0$, $p = 0.3320$, ks2stat = 0.2727. RStest: Conc2: $h = 0$, $p = 0.3820$, zval = 0.8743. KStest2: Conc3: $h = 1$, $p = 0.0138$, ks2stat = 0.4545. RStest: Conc3: $h = 1$, $p = 0.0011$, zval = −3.2659. KStest2: Conc4: $h = 1$, $p < 0.0001$, ks2stat = 0.6818. RStest: Conc4: $h = 1$, $p = 0.0003$, zval = −3.6583.

**Figure 5b (left): FD approach rate at low cost**

Statistical significance was determined using 1-way ANOVA ($F = 12$, $M = 9$). $p$-value for female control vs. food deprivation at lost cost is 9.3628e−04. $p$-value for male between two groups is 0.0032.

**Figure 5b (left): FD Approach rate at high cost**

Statistical significance was determined using 1-way ANOVA ($F = 12$, $M = 10$). $p$-value for females between the two groups is 4.4786e−04, and $p$-value for male between two groups is 0.0126.

**Figure 5c: Approach time (FD vs. control)**

Statistical significance was determined by Repeated measures analysis of variance. (Control $N = 22$, FD $N = 22$). Effect of concentration: d.f. = 3, $F = 20.1324$, $p = 6.3645e−10$. Effect of condition: d.f. = 1, $F = 1.7659$, $p = 0.1946$. kstest2 results: $h = 0$, $p = 3.5436e−01$, ks2stat = 0.1435 (overall difference in control vs. FD).

*Post-hoc analysis*: 0.5%: 1.0418e−02, 2%: 3.2611e−01, 5%: 7.3375e−01, 9%: 1.0336e−01. KStest2 and Wilcoxon rank sum test results (complementary to post-hoc analysis). KStest2: Conc1: $h = 0$, $p = 0.0647$, ks2stat = 0.4286. RStest: Conc1: $h = 1$, $p = 0.0127$, zval = −2.4921. KStest2: Conc2: $h = 1$, $p = 0.0395$, ks2stat = 0.4286. RStest: Conc2: $h = 0$, $p = 0.0883$, zval = −1.7044. KStest2: Conc2: $h = 0$, $p = 0.7388$, ks2stat = 0.1991. RStest: Conc2: $h = 0$, $p = 0.6885$, zval = 0.4009. KStest2: Conc4: $h = 0$, $p = 0.3320$, ks2stat = 0.2727. RStest: Conc4: $h = 0$, $p = 0.1625$, zval = 1.3966.

**Figure 5d: Prop. of trial out. all reward zones (FD vs. Control)**

Statistical significance was determined by Repeated measures analysis of variance. (Control $N = 22$, FD $N = 22$). Effect of concentration: d.f. = 3, $F = 51.5773$, $p = 8.1104e−22$. Effect of Condition: d.f. = 1, $F = 2.5126$, $p = 0.1204$.

kstest2 results: $h = 1$, $p = 1.7572e−02$, ks2stat = 0.2273 (overall difference in control vs. FD).

Post-hoc analysis: 0.5%: 5.6817e−03, 2%: 1.9624e−02, 5%: 5.4119e−01, 9%: 7.4789e−01. KStest2 and Wilcoxon rank sum test Results (complementary to post-hoc analysis). KStest2: Conc1: $h = 1$, $p = 0.0049$, ks2stat = 0.5000. RStest: Conc1: $h = 1$, $p = 0.0068$, zval = −2.7047. KStest2: Conc2: $h = 1$, $p = 0.0356$, ks2stat = 0.4091. RStest: Conc2: $h = 1$, $p = 0.0186$, zval = −2.3536. KStest2: Conc2: $h = 0$, $p = 0.3320$, ks2stat = 0.2727. RStest: Conc2: $h = 0$, $p = 0.2485$, zval = 1.1541. KStest2: Conc4: $h = 0$, $p = 0.3320$, ks2stat = 0.2727. RStest: Conc4: $h = 0$, $p = 0.9156$, zval = −0.1059.

**Figure 5e: Number of stopping points (FD vs. control)**

Statistical significance was determined by Repeated measures analysis of variance. (Control $N = 22$, FD $N = 22$). Effect of concentration: d.f. = 3, $F = 1.2986$, $p = 2.7791e-01$. Effect of Condition: d.f. = 1, $F = 4.3492$, $p = 0.0431$. kstest2 results: $h = 1$, $p = 7.5537e-08$, ks2stat = 0.4318 (overall difference in Control vs. FD).

Post-hoc analysis: 0.5%: 3.9999e−02, 2%: 3.3272e−02, 5%: 5.5036e−02, 9%: 5.9155e−02.

KStest2 and Wilcoxon rank sum test Results (complementary to post-hoc analysis). KStest2: Conc1: $h = 1$, $p = 0.0001$, ks2stat = 0.6364 RStest: Conc1: $h = 1$, $p = 0.0003$, zval = 3.6265. KStest2: Conc2: $h = 1$, $p = 0.0138$, ks2stat = 0.4545. RStest: Conc2: $h = 1$, $p = 0.0028$, zval = 2.9928. KStest2: Conc3: $h = 0$, $p = 0.0828$, ks2stat = 0.3636. RStest: Conc3: $h = 1$, $p = 0.0109$, zval = 2.5468. KStest2: Conc4: $h = 1$, $p = 0.0138$, ks2stat = 0.4545. RStest: Conc4: $h = 1$, $p = 0.0032$, zval = 2.9458.

**Figure 5f: Number of high sp. runs (FD vs. Control)**

Statistical significance was determined by Repeated measures analysis of variance. (Control $N = 22$, FD $N = 22$). Effect of concentration: d.f. = 3, $F = 45.2054$, $p = 6.6926e-20$. Effect of condition: d.f. = 1, $F = 5.4125$, $p = 0.0249$. kstest2 results: $h = 1$, $p = 1.0816e-02$, ks2stat=0.2386 (overall difference in Control vs. FD).

Post-hoc analysis: 0.5%: 7.3584e−04, 2%: 1.0250e−01, 5%: 2.0052e−01, 9%: 1.2278e−01.

KStest2 and Wilcoxon rank sum test Results (complementary to post-hoc analysis). KStest2: Conc1: $h = 1$, $p = 0.0015$, ks2stat = 0.5455. RStest: Conc1: $h = 1$, $p = 0.0012$, zval = 3.2275. KStest2: Conc2: $h = 0$, $p = 0.0828$, ks2stat = 0.3636. RStest: Conc2: $h = 0$, $p = 0.0689$, zval = 1.8191. KStest2: Conc2: $h = 0$, $p = 0.3320$, ks2stat = 0.2727. RStest: Conc2: $h = 0$, $p = 0.4455$, zval = 0.7629. KStest2: Conc4: $h = 0$, $p = 0.1746$, ks2stat = 0.3182. RStest: Conc4: $h = 0$, $p = 0.1424$, zval = 1.4670.

**Figure 5g: Distance traveled (FD vs. Control)**

Statistical significance was determined by Repeated measures analysis of variance. (Control $N = 22$, FD $N = 22$). Effect of concentration: d.f. = 3, $F = 12.6199$, p = 2.8777e−07. Effect of Condition: d.f. = 1, $F = 11.2464$, $p = 0.0017$. kstest2 results: $h = 1$, $p = 7.5537e-08$, ks2stat = 0.4318 (overall difference in Control vs. FD).

Post-hoc analysis: 0.5%: 3.7073e−04, 2%: 7.5759e−04, 5%: 3.3233e−02, 9%: 1.3234e−02. KStest2 and Wilcoxon rank sum test Results (complementary to post-hoc analysis). KStest2: Conc1: $h = 1$, $p = 0.0015$, ks2stat = 0.5455. RStest: Conc1: $h = 1$, $p = 0.0008$, zval = −3.3683. KStest2: Conc2: $h = 1$, $p = 0.0356$, ks2stat = 0.4091. RStest: Conc2: $h = 1$, $p = 0.0028$, zval = −2.9928. KStest2: Conc3: $h = 1$, $p = 0.0138$, ks2stat = 0.4545. RStest: Conc3: $h = 1$, $p = 0.0151$, zval = −2.4294. KStest2: Conc4: $h = 1$, $p = 0.0356$, ks2stat = 0.4091 RStest: Conc4: $h = 1$, $p = 0.0151$, zval = −2.4294.

**Figure 5j: Cluster shifts (Control vs. FD)**

Statistical significance was determined by Chi-squared test. The significance of difference in population in cluster 1, 2 and 3 is 0.0005, 0.1903 and 0.1904, respectively.

**Figure 5l: Baseline and food deprivation early vs late bins Euclidian distance**

Statistical significance $p = 1.8201e-76$, determined by two-sample Kolmogorov–Smirnov test. (Control $N = 23$, FD $N = 22$).

**Figure 5m: Baseline and food deprivation individual rat Euclidian distances**

Statistical significance $p = 0.00058$, determined by two-sample Kolmogorov–Smirnov test (Control $N = 23$, FD $N = 22$).

**Figure 6**

**Figure 6a. Approach rate (Control vs. self admin. Oxy)**

Effect of condition: d.f. = 1, $F = 2.7873$, $p = 0.1051$ kstest2 results: $h = 1$, $p = 0.0049$, ks2stat = 0.3196.

**Figure 6b. Distance traveled (Control vs. Self admin. Oxy)**

Effect of condition: d.f. = 1, $F = 0.0121$, $p = 0.9132$.

**Figure 6c. Number of high-speed runs (Control vs. Self admin. Oxy)**

Effect of condition: d.f. = 1, $F = 0.0015$, $p = 0.9698$.

**Figure 6d. Approach time (Control vs. Self admin. Oxy)**

Effect of condition: d.f. = 1, $F = 1.3040$, $p = 0.2628$.

**Figure 6e. Proportion of trials outside all reward zone (Control vs. Self admin. Oxy)**

Effect of condition: d.f. = 1, $F = 0.1051$, $p = 0.7480$.

**Figure 6f. Number of stopping points (Control vs. Self admin. Oxy)**

Effect of condition: d.f. = 1, $F = 0.5075$, $p = 0.4816$.

**Figure 6g. Approach rate (Control vs Abstinence)**

Effect of condition: d.f. = 1, $F = 6.1129$, $p = 0.0187$. kstest2 results: $h = 1$, $p = 0.0000$, ks2stat = 0.4257

**Figure 6h. Distance traveled (Control vs. Abstinence)**

Effect of condition: d.f. = 1, $F = 4.3279$, $p = 0.0453$.

**Figure 6i. Number of high-speed runs (Control vs. Abstinence)**

Effect of condition: d.f. = 1, $F = 0.6038$, $p = 0.4427$.

**Figure 6j. Approach time (Control vs. Abstinence)**

Effect of condition: d.f. = 1, $F = 0.1133$, $p = 0.7387$.

**Figure 6k. Proportion of trials outside all reward zone (Control vs. Abstinence)**

Effect of condition: d.f. = 1, $F = 1.0138$, $p = 0.3213$.

**Figure 6l. Number of stopping points (Control vs Abstinence)**

Effect of condition: d.f. = 1, $F = 1.9581$, $p = 0.1710$.

**Figure 6m: Self admin oxycodone Approach rate (FvM)**

Statistical significance was determined by repeated measures analysis of variance. (Female $N = 5$, Male $N = 5$). Effect of concentration: d.f. = 3, $F = 3.2073$, $p = 4.1083e-02$. Effect of sex: d.f. = 1, $F = 0.0521$, $p = 0.8251$. kstest2 results: $h = 0$, $p = 7.7095e-01$, ks2stat = 0.2000 (overall sex difference).

Post-hoc analysis: 0.5%: 8.0968e−01, 2%: 2.1173e−01, 5%: 4.2256e−01, 9%: 2.2622e−01.

KStest2 and Wilcoxon rank sum test Results (complementary to post-hoc analysis). KStest2: Conc1: $h = 0$, $p = 0.6974$, ks2stat = 0.4000. RStest: Conc1: $h = 0$, $p = 0.6429$. KStest2: Conc2: $h = 0$, $p = 0.2090$, ks2stat = 0.6000. RStest: Conc2: $h = 0$, $p = 0.2063$. KStest2: Conc2: $h = 0$, $p = 0.6974$, ks2stat = 0.4000. RStest: Conc2: $h = 0$, $p = 0.6349$. KStest2: Conc4: $h = 0$, $p = 0.6974$, ks2stat = 0.4000. RStest: Conc4: $h = 0$, $p = 0.3016$.

**Control vs. Self admin. Oxy Female:**

Statistical significance was determined by Repeated measures analysis of variance. (Control $N = 12$, Self admin. Oxy $N = 5$). $p$-value for Control vs. Self-admin. Oxy of female: 0.052962. kstest2 results: $h = 0$, $p = 5.1949e-02$, ks2stat = 0.3458.

Post-hoc analysis: 0.5%: 4.0723e−04, 2%: 3.0385e−01, 5%: 8.8902e−02, 9%: 1.2880e−05.

KStest2 and Wilcoxon rank sum test Results (complementary to post-hoc analysis). KStest2: Conc1: $h = 1$, $p = 0.0089$, ks2stat = 0.8000. RStest: Conc1: $h = 1$, $p = 0.0039$. KStest2: Conc2: $h = 0$, $p = 0.5074$, ks2stat = 0.4000. RStest: Conc2: $h = 0$, $p = 0.7757$. KStest2: Conc2: $h = 0$, $p = 0.1545$, ks2stat = 0.5500. RStest: Conc2: $h = 0$, $p = 0.1296$. KStest2: Conc4: $h = 1$, $p = 0.0004$, ks2stat = 1.0000. RStest: Conc4: $h = 1$, $p = 0.0003$.

**Male**: Statistical significance was determined by Repeated measures analysis of variance. (Control $N = 11$, Self admin. Oxy $N = 5$). $p$-value for Control vs Self admin. Oxy of male: 0.74837. kstest2 results: $h = 0$, $p = 5.2181e-02$, ks2stat = 0.3500.

Post-hoc analysis: 0.5%: 1.1302e−02, 2%: 5.2769e−04, 5%: 3.4051e−01, 9%: 7.5302e−05. KStest2 and Wilcoxon rank sum test Results (complementary to post-hoc analysis). KStest2: Conc1: $h = 1$, $p = 0.0313$, ks2stat = 0.7091. RStest: Conc1: $h = 1$, $p = 0.0124$. KStest2: Conc2: $h = 1$, $p = 0.0079$, ks2stat = 0.8182. RStest: Conc2: $h = 1$, $p = 0.0018$. KStest2: Conc2: $h = 0$, $p = 0.2005$, ks2stat = 0.5273. RStest: Conc2: $h = 0$, $p = 0.3608$. KStest2: Conc4: $h = 1$, $p = 0.0005$, ks2stat = 1.0000. RStest: Conc4: $h = 1$, $p = 0.0005$.

**3-way ANOVA results**

| Source | Sum Sq. | d.f. | Singular? | Mean sq. | F | Prob > F |
|---|---|---|---|---|---|---|
| Sex | 0.0122 | 1 | 0 | 0.0122 | 0.6013 | 0.4396 |
| Condition | 0.0762 | 1 | 0 | 0.0762 | 3.7568 | 0.055 |
| Concentration | 3.0662 | 3 | 0 | 1.0221 | 50.3587 | 0 |
| Sex*Condition | 0.0379 | 1 | 0 | 0.0379 | 1.8658 | 0.1745 |
| Sex*Concentration | 0.1742 | 3 | 0 | 0.0581 | 2.8615 | 0.0398 |
| Condition*Concentration | 1.3945 | 3 | 0 | 0.4648 | 22.9026 | 0 |
| Error | 2.4152 | 119 | 0 | 0.0203 | NaN | NaN |
| Total | 9.9283 | 131 | 0 | NaN | NaN | NaN |

KS test for the effect of condition: $h = 1$, $p = 0.0049$, KS statistic = 0.3196.

**Figure 6n: Self admin oxycodone Distance traveled (FvM)**

Statistical significance was determined by Repeated measures analysis of variance. (Female $N = 5$, Male $N = 5$). Effect of concentration: d.f. = 3, $F = 0.8971$, $p = 4.5703e-01$. Effect of sex: d.f. = 1, $F = 4.6420$, $p = 0.0633$. kstest2 results: $h=1$, $p=7.2529e-04$, ks2stat=0.6000 (overall sex difference).

Post-hoc analysis: 0.5%: 1.7344e−01, 2%: 9.6526e−02, 5%: 5.1225e−02, 9%: 4.6853e−02. KStest2 and Wilcoxon rank sum test Results (complementary to post-hoc analysis). KStest2: Conc1: $h = 0$, $p = 0.2090$, ks2stat = 0.6000. RStest: Conc1: $h = 0$, p = 0.2222. KStest2: Conc2: $h = 0$, $p = 0.2090$, ks2stat = 0.6000. RStest: Conc2: $h = 0$, $p = 0.0952$. KStest2: Conc2: $h = 1$, $p = 0.0361$, ks2stat = 0.8000. RStest: Conc2: $h = 0$, $p = 0.0556$. KStest2:Conc4: $h = 0$, $p = 0.2090$, ks2stat = 0.6000. RStest: Conc4: $h = 0$, $p = 0.0952$.

**Control vs. Self admin. Oxy Female**:

Statistical significance was determined by Repeated measures analysis of variance. (Control $N = 12$, Self admin. Oxy $N = 5$). $p$-value for Control vs Self admin. Oxy of female: 0.044575. kstest2 results: $h=1$, $p=7.5094e-04$, ks2stat=0.5083

Post-hoc analysis: 0.5%: 9.0352e−02, 2%: 3.8910e−02, 5%: 8.9572e−02, 9%: 3.1966e−02. KStest2 and Wilcoxon rank sum test Results (complementary to post-hoc analysis). KStest2: Conc1: $h = 0$, $p = 0.3153$, ks2stat = 0.4667. *RStest: Conc1: $h = 0$, p = 0.1946. KStest2: Conc2: $h = 0$, $p = 0.0950$, ks2stat = 0.6000.RStest: Conc2: $h = 0$, $p = 0.1037$. KStest2: Conc2: $h = 0$, $p = 0.2086$, ks2stat = 0.5167. RStest: Conc2: $h = 0$, $p = 0.0818$. KStest2: Conc4: $h = 0$, $p = 0.0671$, ks2stat = 0.6333. RStest: Conc4: $h = 1$, $p = 0.0365$.

**Male**: Statistical significance was determined by Repeated measures analysis of variance. (Control $N = 11$, Self admin. Oxy $N = 5$). $p$-value for Control vs. Self-admin. Oxy of male: 0.0032032. kstest2 results: $h=1$, $p=6.9107e-06$, ks2stat=0.6500. Post-hoc analysis: 0.5%: 5.9931e−03, 2%: 2.1350e−02, 5%: 1.3194e−03, 9%: 9.8642e−03.

KStest2 and Wilcoxon rank sum test Results (complementary to post-hoc analysis): KStest2: Conc1: $h = 1$, $p = 0.0313$, ks2stat = 0.7091. *RStest: Conc1: $h = 1$, p = 0.0275. KStest2: Conc2: $h = 0$, $p = 0.0703$, ks2stat = 0.6364. RStest: Conc2: $h = 0$, $p = 0.0517$. KStest2: Conc2: $h = 1$, $p = 0.0252$, ks2stat = 0.7273. RStest: Conc2: $h = 1$, $p = 0.0087$. KStest2: Conc4: $h = 0$, $p = 0.0848$, ks2stat = 0.6182. RStest: Conc4: $h = 1$, $p = 0.0275$.

**3-way ANOVA results**

| Source | Sum Sq. | d.f. | Singular? | Mean sq. | F | Prob > F |
|---|---|---|---|---|---|---|
| Sex | 0.0155 | 1 | 0 | 0.0155 | 0.0742 | 0.7858 |
| Condition | 0.033 | 1 | 0 | 0.033 | 0.1576 | 0.6921 |
| Concentration | 0.2361 | 3 | 0 | 0.0787 | 0.376 | 0.7705 |
| Sex*Condition | 10.5047 | 1 | 0 | 10.5047 | 50.1828 | 0 |
| Sex*Concentration | 0.313 | 3 | 0 | 0.1043 | 0.4985 | 0.684 |
| Condition*Concentration | 0.1143 | 3 | 0 | 0.0381 | 0.182 | 0.9084 |
| Error | 24.91 | 119 | 0 | 0.2093 | NaN | NaN |
| Total | 37.6947 | 131 | 0 | NaN | NaN | NaN |

**Fig. 6o: Self admin oxycodone number of high sp. runs**

Statistical significance was determined by Repeated measures analysis of variance. (Female $N = 5$, Male $N = 5$).

Effect of concentration: d.f. = 3, $F = 0.5993$, $p = 6.2169e-01$.

Effect of sex: d.f. = 1, $F = 3.7946$, $p = 0.0873$. kstest2 results: $h = 1$, $p = 8.1617e-03$, ks2stat=0.5000 (overall sex difference).

Post-hoc analysis: 0.5%: 1.9249e−01, 2%: 1.3288e−01, 5%: 5.9820e−02, 9%: 5.9578e−02.

KStest2 and Wilcoxon rank sum test Results (complementary to post-hoc analysis). KStest2: Conc1: $h = 0$, $p = 0.6974$, ks2stat = 0.4000. RStest: Conc1: $h = 0$, $p = 0.2222$. KStest2: Conc2: $h = 0$, $p = 0.6974$, ks2stat = 0.4000. RStest: Conc2: $h = 0$, $p = 0.2090$, ks2stat = 0.6000. RStest: Conc2: $h = 0$, $p = 0.0952$. KStest2: Conc4: $h = 0$, $p = 0.2090$, ks2stat = 0.6000. RStest: Conc4: $h = 0$, $p = 0.0952$.

**Control vs. Self admin. Oxy Female:**

Statistical significance was determined by Repeated measures analysis of variance. (Control $N = 12$, Self admin. Oxy $N = 5$). $p$-value for Control vs. Self-admin. Oxy of female: 0.026023. kstest2 results: $h=1$, $p=1.3666e-02$, ks2stat=0.4042. Post-hoc analysis: 0.5%: 2.8828e−02, 2%: 2.3955e−02, 5%: 8.6383e−02, 9%: 6.4387e−02.

KStest2 and Wilcoxon rank sum test Results (complementary to post-hoc analysis): KStest2: Conc1: $h = 0$, $p = 0.2086$, ks2stat = 0.5167. RStest: Conc1: $h = 0$, $p = 0.1296$. KStest2: Conc2: $h = 0$, $p = 0.2406$, ks2stat = 0.5000. RStest: Conc2: $h = 0$, $p = 0.0637$. KStest2: Conc2: $h = 0$, $p = 0.2086$, ks2stat = 0.5167. RStest: Conc2: $h = 0$, $p = 0.2786$. KStest2: Conc4: $h = 0$, $p = 0.5074$, ks2stat = 0.4000. RStest: Conc4: $h = 0$, $p = 0.3284$.

**Male**: Statistical significance was determined by Repeated measures analysis of variance. (Control $N = 11$, Self admin. Oxy $N = 5$). $p$-value for Control vs. Self-admin. Oxy of male: 0.0071297. kstest2 results: $h = 1$, $p = 1.6397e-05$, ks2stat = 0.6273.

Post-hoc analysis: 0.5%: 1.6520e−02, 2%: 1.9696e−02, 5%: 2.7185e−03, 9%: 1.8464e−02.

KStest2 and Wilcoxon rank sum test Results (complementary to post-hoc analysis): KStest2: Conc1: $h = 1$, $p = 0.0313$, ks2stat = 0.7091. RStest: Conc1: $h = 0$, $p = 0.0687$. KStest2: Conc2: $h = 0$, $p = 0.1019$, ks2stat = 0.6000. RStest: Conc2: $h = 1$, $p = 0.0380$. KStest2: Conc2: $h = 1$, $p = 0.0252$, ks2stat = 0.7273. RStest: Conc2: $h = 1$, $p = 0.0055$. KStest2: Conc4: $h = 0$, $p = 0.0848$, ks2stat = 0.6182. RStest: Conc4: $h = 1$, $p = 0.0380$.

**3-way ANOVA results**

| Source | Sum sq. | d.f. | Singular? | Mean sq. | F | Prob > F |
|---|---|---|---|---|---|---|
| Sex | 63.1409 | 1 | 0 | 63.1409 | 3.463 | 0.0652 |
| Condition | 0.0162 | 1 | 0 | 0.0162 | 0.0009 | 0.9763 |
| Concentration | 100.3972 | 3 | 0 | 33.4657 | 1.8354 | 0.1444 |
| Sex*Condition | 906.117 | 1 | 0 | 906.117 | 49.6964 | 0 |
| Sex*Concentration | 26.2414 | 3 | 0 | 8.7471 | 0.4797 | 0.697 |
| Condition*Concentration | 48.6865 | 3 | 0 | 16.2288 | 0.8901 | 0.4485 |
| Error | 2169.7317 | 119 | 0 | 18.233 | NaN | NaN |
| Total | 3345.3199 | 131 | 0 | NaN | NaN | NaN |

**Fig. 6p: Abstinence approach rate (FvM)**

Statistical significance was determined by Repeated measures analysis of variance. (Female $N = 6$, Male $N = 6$). Effect of concentration: d.f. = 3, $F = 19.9665$, $p = 2.6307e-07$. Effect of sex: d.f. = 1, $F = 0.2008$, $p = 0.6637$. kstest2 results: $h = 0$, $p = 6.2161e-01$, ks2stat = 0.2083 (overall sex difference).

Post-hoc analysis: 0.5%: 7.5735e−01, 2%: 3.7013e−01, 5%: 8.0930e−01, 9%: 6.4244e−01.

KStest2 and Wilcoxon rank sum test Results (complementary to post-hoc analysis). KStest2: Conc1: $h = 0$, $p = 0.8096$, ks2stat = 0.3333. RStest: Conc1: $h = 0$, $p = 1.0000$. KStest2: Conc2: $h = 0$, $p = 0.3180$, ks2stat = 0.5000. RStest: Conc2: $h = 0$, $p = 0.3095$. KStest2: Conc2: $h = 0$, $p = 0.8096$, ks2stat =

0.3333. RStest: Conc2: $h = 0$, $p = 0.8182$. KStest2: Conc4: $h = 0$, $p = 0.3180$, ks2stat = 0.5000. RStest: Conc4: $h = 0$, $p = 0.5887$.

**Control vs. Abstinence Female**: Statistical significance was determined by Repeated measures analysis of variance. (Control $N = 12$, Abstinence $N = 6$). $p$-value for Control vs. initial task of female: 0.18211. kstest2 results: $h = 1$, $p = 1.5846e-02$, ks2stat = 0.3750.

Post-hoc analysis: 0.5%: 9.8839e−05, 2%: 2.3068e−04, 5%: 8.7695e−01, 9%: 1.7166e−01.

KStest2 and Wilcoxon rank sum test Results (complementary to post-hoc analysis): KStest2: Conc1: $h = 1$, $p = 0.0007$, ks2stat = 0.9167. RStest: Conc1: $h = 1$, $p = 0.0002$. KStest2: Conc2: $h = 1$, $p = 0.0028$, ks2stat = 0.8333. RStest: Conc2: $h = 1$, $p = 0.0018$. KStest2: Conc2: $h = 0$, $p = 0.9290$, ks2stat = 0.2500. RStest: Conc2: $h = 0$, $p = 0.9636$. KStest2: Conc4: $h = 0$, $p = 0.1877$, ks2stat = 0.5000. RStest: Conc4: $h = 0$, $p = 0.4225$.

**Male**: Statistical significance was determined by Repeated measures analysis of variance. (Control $N = 11$, Abstinence $N = 6$). $p$-value for Control vs. initial task of male: 0.033933. kstest2 results: $h = 1$, $p = 8.7249e-04$, ks2stat = 0.4811.

Post-hoc analysis: 0.5%: 4.0393e−05, 2%: 6.3068e−04, 5%: 3.1164e−01, 9%: 5.9881e−03.

KStest2 and Wilcoxon rank sum test Results (complementary to post-hoc analysis): KStest2: Conc1: $h = 1$, $p = 0.0002$, ks2stat = 1.0000. RStest: Conc1: $h = 1$, $p = 0.0002$. KStest2: Conc2: $h = 1$, $p = 0.0033$, ks2stat = 0.8333. RStest: Conc2: $h = 1$, $p = 0.0031$. KStest2: Conc2: $h = 0$, $p = 0.5232$, ks2stat = 0.3788. RStest: Conc2: $h = 0$, $p = 0.5249$. KStest2: Conc4: $h = 1$, $p = 0.0042$, ks2stat = 0.8182. RStest: Conc4: $h = 1$, $p = 0.0074$.

**3-way ANOVA results**

| Source | Sum Sq. | d.f. | Singular? | Mean sq. | F | Prob > F |
|---|---|---|---|---|---|---|
| Sex | 0.0662 | 1 | 0 | 0.0662 | 3.0165 | 0.0848 |
| Condition | 0.228 | 1 | 0 | 0.228 | 10.3954 | 0.0016 |
| Concentration | 5.0851 | 3 | 0 | 1.695 | 77.2692 | 0 |
| Sex*Condition | 0.0045 | 1 | 0 | 0.0045 | 0.2061 | 0.6506 |
| Sex*Concentration | 0.0954 | 3 | 0 | 0.0318 | 1.4503 | 0.2314 |
| Condition*Concentration | 0.7475 | 3 | 0 | 0.2492 | 11.3581 | 0 |
| Error | 2.786 | 127 | 0 | 0.0219 | NaN | NaN |
| Total | 11.0015 | 139 | 0 | NaN | NaN | NaN |

KS test for the effect of condition: $h = 1$, $p < 0.0001$, KS statistic = 0.4257.

**Figure 6q: Abstinence distance traveled (FvM)**
Statistical significance was determined by Repeated measures analysis of variance. (Female $N = 6$, Male $N = 6$). Effect of concentration: d.f. = 3, $F = 16.2563$, $p = 1.8477e-06$. Effect of sex: d.f. = 1, $F = 1.0727$, $p = 0.3247$. kstest2 results: $h = 0$, $p = 2.1598e-01$, ks2stat = 0.2917 (overall sex difference).

Post-hoc analysis: 0.5%: 4.5320e−01, 2%: 4.5178e−01, 5%: 3.0428e−01, 9%: 2.1404e−01.

KStest2 and Wilcoxon rank sum test Results (complementary to post-hoc analysis): KStest2: Conc1: $h = 0$, $p = 0.8096$, ks2stat = 0.3333. RStest: Conc1: $h = 0$, $p = 0.4848$. KStest2: Conc2: $h = 0$, $p = 0.8096$, ks2stat = 0.3333. RStest: Conc2: $h = 0$, $p = 0.4848$. KStest2: Conc2: $h = 0$, $p = 0.0766$, ks2stat = 0.6667. RStest: Conc2: $h = 0$, $p = 0.3939$. KStest2: Conc4: $h = 0$, $p = 0.3180$, ks2stat = 0.5000. RStest: Conc4: $h = 0$, $p = 0.2403$

**Control vs. abstinence Female**:
Statistical significance was determined by Repeated measures analysis of variance. (Control $N = 12$, Abstinence $N = 6$). $p$-value for Control vs. initial task of female: 0.44479. kstest2 results: $h = 0$, $p = 1.0713e-01$, ks2stat = 0.2917. Post-hoc analysis: 0.5%: 1.4870e−01, 2%: 2.5012e−01, 5%: 9.4955e−01, 9%: 9.9174e−01.

KStest2 and Wilcoxon rank sum test Results (complementary to post-hoc analysis): KStest2: Conc1: $h = 0$, $p = 0.6693$, ks2stat = 0.3333. RStest:

Conc1: $h = 0$, p = 0.1797. KStest2: Conc2: $h = 0$, $p = 0.6693$, ks2stat = 0.3333. RStest: Conc2: $h = 0$, $p = 0.3355$. KStest2: Conc2: $h = 0$, $p = 0.1877$, ks2stat = 0.5000. RStest: Conc2: $h = 0$, $p = 0.4371$. KStest2: Conc4: $h = 0$, $p = 0.9290$, ks2stat = 0.2500. RStest: Conc4: $h = 0$, $p = 0.7503$.

**Male**: Statistical significance was determined by Repeated measures analysis of variance. (Control $N = 11$, Abstinence $N = 6$). $p$-value for Control vs. initial task of male: 0.0092961. kstest2 results: $h = 1$, $p = 1.2593e-04$, ks2stat = 0.5379. Post-hoc analysis: 0.5%: 1.2197e−03, 2%: 1.8008e−02, 5%: 2.3136e−02, 9%: 1.0416e−01. KStest2 and Wilcoxon rank sum test Results (complementary to post-hoc analysis).

KStest2: Conc1: $h = 1$, $p = 0.0125$, ks2stat = 0.7424. RStest: Conc1: $h = 1$, $p = 0.0071$. KStest2: Conc2: $h = 0$, $p = 0.1997$, ks2stat = 0.5000. RStest: Conc2: $h = 1$, $p = 0.0365$. KStest2: Conc2: $h = 1$, $p = 0.0480$, ks2stat = 0.6364. RStest: Conc2: $h = 0$, $p = 0.0616$. KStest2: Conc4: $h = 1$, $p = 0.0401$, ks2stat = 0.6515. RStest: Conc4: $h = 0$, $p = 0.0616$.

**3-way ANOVA results**

| Source | Sum Sq. | d.f. | Singular? | Mean Sq. | F | Prob > F |
|---|---|---|---|---|---|---|
| Sex | 5.1572 | 1 | 0 | 5.1572 | 27.9576 | 0 |
| Condition | 3.9704 | 1 | 0 | 3.9704 | 21.5238 | 0 |
| Concentration | 2.0282 | 3 | 0 | 0.6761 | 3.665 | 0.0142 |
| Sex*Condition | 1.0897 | 1 | 0 | 1.0897 | 5.9071 | 0.0165 |
| Sex*Concentration | 0.1722 | 3 | 0 | 0.0574 | 0.3112 | 0.8173 |
| Condition*Concentration | 1.1314 | 3 | 0 | 0.3771 | 2.0444 | 0.111 |
| Error | 23.427 | 127 | 0 | 0.1845 | NaN | NaN |
| Total | 38.3866 | 139 | 0 | NaN | NaN | NaN |

**Fig. 6r: Abstinence number of high sp. runs**
Statistical significance was determined by Repeated measures analysis of variance. (Female $N = 6$, Male $N = 6$). Effect of concentration: d.f. = 3, $F = 6.9387$, $p = 1.1029e-03$. Effect of sex: d.f. = 1, $F = 1.0006$, $p = 0.3408$. kstest2 results: $h = 0$, $p = 5.0588e-02$, ks2stat = 0.3750 (overall sex difference). Post-hoc analysis: 0.5%: 5.1260e−01, 2%: 3.1392e−01, 5%: 2.5092e−01, 9%: 3.6439e−01.

KStest2 and Wilcoxon rank sum test Results (complementary to post-hoc analysis): KStest2: Conc1: $h = 0$, $p = 1.0000$, ks2stat = 0.1667. RStest: Conc1: $h = 0$, $p = 0.8182$. KStest2: Conc2: $h = 0$, $p = 0.3180$, ks2stat = 0.5000. RStest: Conc2: $h = 0$, p = 0.2403. KStest2: Conc2: $h = 0$, $p = 0.3180$, ks2stat = 0.5000. RStest: Conc2: $h = 0$, $p = 0.1797$. KStest2: Conc4: $h = 1$, $p = 0.0122$, ks2stat = 0.8333. RStest: Conc4: $h = 1$, $p = 0.0411$.

**Control vs. Abstinence Female:**
Statistical significance was determined by Repeated measures analysis of variance. (Control $N = 12$, Abstinence $N = 6$). $p$-value for Control vs. initial task of female: 0.81023. kstest2 results: $h = 0$, $p = 1.0713e-01$, ks2stat = 0.2917.

Post-hoc analysis: 0.5%: 7.6146e−01, 2%: 3.6133e−01, 5%: 5.3077e−01, 9%: 4.9879e−01

KStest2 and Wilcoxon rank sum test Results (complementary to post-hoc analysis): KStest2: Conc1: $h = 0$, $p = 0.9994$, ks2stat = 0.1667. RStest: Conc1: $h = 0$, $p = 0.8916$. KStest2: Conc2: $h = 0$, $p = 0.6693$, ks2stat = 0.3333. RStest: Conc2: $h = 0$, $p = 0.4371$. KStest2: Conc2: $h = 0$, $p = 0.3842$, ks2stat = 0.4167. RStest: Conc2: $h = 0$, $p = 0.5532$. *KStest2: Conc4: $h = 0$, p = 0.0799*, ks2stat = 0.5833. RStest: Conc4: $h = 0$, $p = 0.1025$.

**Male**: Statistical significance was determined by Repeated measures analysis of variance. (Control $N = 11$, Abstinence $N = 6$). $p$-value for Control vs. initial task of male: 0.36093. kstest2 results: $h = 1$, $p = 6.6954e-03$, ks2stat=0.4129.

Post-hoc analysis: 0.5%: 8.8333e−02, 2%: 6.2386e−01, 5%: 1.2699e−01, 9%: 5.0279e−01.

KStest2 and Wilcoxon rank sum test Results (complementary to post-hoc analysis) KStest2: Conc1: KStest2: Conc2: KStest2: Conc2: KStest2: Conc4: h = 1, $p$ = 0.0125, ks2stat = 0.7424 RStest: Conc1: $h$ = 1, $p$ = 0.0477 h = 0, $p$ = 0.9495, ks2stat = 0.2424 RStest: Conc2: $h$ = 0, $p$ = 0.8075, $h$ = 0, $p$ = 0.1106, ks2stat = 0.5606 RStest: Conc2: h = 0, $p$ = 0.0983, $h$ = 0, $p$ = 0.5232, ks2stat = 0.3788. RStest: Conc4: $h$ = 0, $p$ = 0.4623.

**3-way ANOVA results**

| Source | Sum Sq. | d.f. | Singular? | Mean Sq. | *F* | Prob > *F* |
|---|---|---|---|---|---|---|
| Sex | 355.3947 | 1 | 0 | 355.3947 | 26.0124 | 0 |
| Condition | 41.0987 | 1 | 0 | 41.0987 | 3.0081 | 0.0853 |
| Concentration | 179.8368 | 3 | 0 | 59.9456 | 4.3876 | 0.0057 |
| Sex*Condition | 22.2205 | 1 | 0 | 22.2205 | 1.6264 | 0.2045 |
| Sex*Concentration | 10.9513 | 3 | 0 | 3.6504 | 0.2672 | 0.8489 |
| Condition*Concentration | 61.1791 | 3 | 0 | 20.393 | 1.4926 | 0.2197 |
| Error | 1735.1417 | 127 | 0 | 13.6625 | NaN | NaN |
| Total | 2545.4129 | 139 | 0 | NaN | NaN | NaN |

**Fig. 6s: Fraction of sigmoid (Control vs. Self admin vs. Abstinence)**

Statistical significance was determined by one-way analysis of variance. (Female Control = 10, Male Control = 10, Female Self admin. Oxy = 5, Male Self admin. Oxy = 5, Female Abstinence = 6, Male Abstinence = 6). Significance of difference between the groups: d.f. = 5, $F$ = 17.0600, $p$ = 1.2356e−08. Post-hoc analysis by Tukey's HSD method: Female Control and Male Control: 0.9862. Female Control and Female Self Admin: 9.5155e−06. Male Control and Male Self Admin: 6.2293e−06. Female Control and Female Abstinence: 0.0084. Male Control and Male Abstinence: 2.9341e−04.

**Figure 6t: Cluster shifts (Control vs. Self-admin oxy)**

Statistical significance was determined by Chi-squared test. The significance of the difference in population in clusters 1–3 is 0.0235, 0.9455 and 0.1187, respectively.

**Figure 6v: Baseline and Oxy individual rat Euclidian distances** Statistical significance $p$ = 6.8828e−38, determined by two-sample Kolmogorov–Smirnov test. (Control $N$ = 23, Oxy $N$ = 12).

**Figure 7**

**Figure 7b: Initial task Approach rate (FvM)**

Statistical significance was determined by Repeated measures analysis of variance. (Female $N$ = 12, Male $N$ = 10). Effect of concentration: d.f. = 3, $F$ = 49.9905, $p$ = 2.5457e−16. Effect of sex: d.f. = 1, $F$ = 0.5183, $p$ = 0.4799. kstest2 results: $h$ = 0, $p$ = 8.0438e−01, ks2stat = 0.1333 (overall sex difference).

Post-hoc analysis: 0.5%: 8.0114e−01, 2%: 8.8708e−01, 5%: 5.7254e−01, 9%: 5.0164e−01.

KStest2 and Wilcoxon rank sum test Results (complementary to post-hoc analysis): KStest2: Conc1: $h$ = 0, $p$ = 0.8848, ks2stat = 0.2333. RStest: Conc1: $h$ = 0, $p$ = 0.8621, zval = 0.1736. KStest2: Conc2: $h$ = 0, $p$ = 1.0000, ks2stat = 0.1167. RStest: Conc2: $h$ = 0, $p$ = 1.0000, zval = −0.0000. KStest2: Conc3: $h$ = 0, $p$ = 0.9304, ks2stat = 0.2167. RStest: Conc3: $h$ = 0, $p$ = 0.9467, zval = 0.0668. KStest2: Conc4: $h$ = 0, $p$ = 0.8848, ks2stat = 0.2333. RStest: Conc4: $h$ = 0, $p$ = 1.0000, zval = −0.0000.

**Control vs. Initial Task Female:**

Statistical significance was determined by Repeated measures analysis of variance. (Control $N$ = 12, Initial Task $N$ = 12). $p$-value for Control vs. initial task of female: 0.17699. kstest2 results: $h$ = 0, $p$ = 4.8027e−01, ks2stat = 0.1667.

Post-hoc analysis: 0.5%: 4.7801e−01, 2%: 8.2100e−01, 5%: 2.6400e−02, 9%: 9.4445e−01.

KStest2 and Wilcoxon rank sum test Results (complementary to post-hoc analysis): KStest2: Conc1: $h$ = 0, $p$ = 0.1862, ks2stat = 0.4167. RStest: Conc1: $h$ = 0, $p$ = 0.5008. KStest2: Conc2: $h$ = 0, $p$ = 0.0656, ks2stat = 0.5000. RStest: Conc2: $h$ = 0, $p$ = 0.3827. KStest2: Conc3: $h$ = 0, $p$ = 0.1862, ks2stat =

0.4167. RStest: Conc3: $h$ = 0, $p$ = 0.0526. KStest2: Conc4: $h$ = 0, $p$ = 0.4333, ks2stat = 0.3333. RStest: Conc4: $h$ = 0, $p$ = 0.7708.

**Male**: Statistical significance was determined by Repeated measures analysis of variance. (Control $N$ = 11, Initial Task $N$ = 10). $p$-value for Control vs. initial task of male: 0.066116. kstest2 results: $h$ = 0, $p$ = 6.6144e−01, ks2stat = 0.1545.

Post-hoc analysis: 0.5%: 3.4193e−01, 2%: 2.3440e−01, 5%: 2.4745e−02, 9%: 3.3059e−01.

KStest2 and Wilcoxon rank sum test Results (complementary to post-hoc analysis): KStest2: Conc1: $h$ = 0, $p$ = 0.2890, ks2stat = 0.4000. RStest: Conc1: $h$ = 0, $p$ = 0.9153. KStest2: Conc2: $h$ = 0, $p$ = 0.6490, ks2stat = 0.3000. RStest: Conc2: $h$ = 0, $p$ = 0.7219. KStest2: Conc3: $h$ = 1, $p$ = 0.0259, ks2stat = 0.6000. RStest: Conc3: $h$ = 1, $p$ = 0.0150. KStest2: Conc4: $h$ = 0, $p$ = 0.2890, ks2stat = 0.4000. RStest: Conc4: $h$ = 0, $p$ = 0.8039.

**3-way ANOVA results**

| Source | Sum Sq. | d.f. | Singular? | Mean Sq. | *F* | Prob > *F* |
|---|---|---|---|---|---|---|
| Sex | 0.0929 | 1 | 0 | 0.0929 | 2.8631 | 0.0925 |
| Condition | 0.1859 | 1 | 0 | 0.1859 | 5.7305 | 0.0178 |
| Concentration | 13.402 | 3 | 0 | 4.4673 | 137.6882 | 0 |
| Sex*Condition | 0.0067 | 1 | 0 | 0.0067 | 0.2063 | 0.6503 |
| Sex*Concentration | 0.0808 | 3 | 0 | 0.0269 | 0.8303 | 0.4789 |
| Condition*Concentration | 0.4846 | 3 | 0 | 0.1615 | 4.9791 | 0.0025 |
| Error | 5.4184 | 167 | 0 | 0.0324 | NaN | NaN |
| Total | 19.8036 | 179 | 0 | NaN | NaN | NaN |

**Fig. 7c: Late task Approach rate (FvM)**

Statistical significance was determined by Repeated measures analysis of variance. (Female $N$ = 10, Male $N$ = 10). Effect of concentration: d.f. = 3, $F$ = 65.1004, $p$ = 6.1134e−18. Effect of sex: d.f. = 1, $F$ = 6.0131, $p$ = 0.0246. kstest2 results: $h$ = 0, $p$ = 3.6131e−01, ks2stat = 0.2000 (overall sex difference).

Post-hoc analysis: 0.5%: 8.8484e−01, 2%: 1.0123e−01, 5%: 7.5526e−02, 9%: 3.3494e−01.

KStest2 and Wilcoxon rank sum test Results (complementary to post-hoc analysis): KStest2: Conc1: $h$ = 0, $p$ = 0.9748, ks2stat = 0.2000. RStest: Conc1: $h$ = 0, $p$ = 0.5004, zval = 0.6739. KStest2: Conc2: $h$ = 0, $p$ = 0.3129, ks2stat = 0.4000. RStest: Conc2: $h$ = 0, $p$ = 0.1315, zval = 1.5083. KStest2: Conc3: $h$ = 0, $p$ = 0.6751, ks2stat = 0.3000. RStest: Conc3: $h$ = 0, $p$ = 0.1233, zval = 1.5411. KStest2: Conc4: $h$ = 0, $p$ = 0.9748, ks2stat = 0.2000. RStest: Conc4: $h$ = 0, $p$ = 0.4201, zval = 0.8062.

**Control vs. Late Task Female:**

Statistical significance was determined by Repeated measures analysis of variance. (Control $N$ = 12, Late Task $N$ = 10). $p$-value for Control vs. initial task of female: 0.024429. kstest2 results: $h$ = 0, $p$ = 2.2658e−01, ks2stat = 0.2167.

Post-hoc analysis: 0.5%: 3.7745e−01, 2%: 1.4453e−01, 5%: 6.0921e−02, 9%: 2.1423e−01.

KStest2 and Wilcoxon rank sum test Results (complementary to post-hoc analysis): KStest2: Conc1: $h$ = 0, $p$ = 0.5564, ks2stat = 0.3167. RStest: Conc1: $h$ = 0, $p$ = 0.5960. KStest2: Conc2: $h$ = 0, $p$ = 0.0706, ks2stat = 0.5167. RStest: Conc2: $h$ = 0, $p$ = 0.0703. KStest2: Conc3: $h$ = 0, $p$ = 0.1582, ks2stat = 0.4500. RStest: Conc3: $h$ = 0, $p$ = 0.0688. KStest2: Conc4: $h$ = 0, $p$ = 0.0873, ks2stat = 0.5000. RStest: Conc4: $h$ = 0, $p$ = 0.2595.

**Male**: Statistical significance was determined by Repeated measures analysis of variance (Control $N$ = 11, Late Task $N$ = 10). $p$-value for Control vs. initial task of male: 0.2364. kstest2 results: $h$ = 0, $p$ = 3.0854e-01, ks2stat = 0.2045.

Post-hoc analysis: 0.5%: 5.0440e−01, 2%: 4.2998e−01, 5%: 2.9060e−01, 9%: 5.8461e−01.

KStest2 and Wilcoxon rank sum test Results (complementary to post-hoc analysis): KStest2: Conc 1: $h = 0$, $p = 0.0978$, ks2stat=0.5000 RStest: Conc 1: $h = 0$, $p = 0.3548$ KStest2: Conc 2: $h = 0$, $p = 0.6114$, ks2stat=0.3091 RStest: Conc 2: $h = 0$, $p = 0.9151$ KStest2: Conc 3: $h = 0$, $p = 0.4673$, ks2stat=0.3455 RStest: Conc 3: $h = 0$, $p = 0.2594$ KStest2: Conc 4: $h = 0$, $p = 0.8888$, ks2stat=0.2364 RStest: Conc 4: $h = 0$, $p = 0.5716$.

**3-way ANOVA results**

| Source | Sum Sq. | d.f. | Singular? | Mean Sq. | F | Prob > F |
|---|---|---|---|---|---|---|
| Sex | 0.2624 | 1 | 0 | 0.2624 | 10.4186 | 0.0015 |
| Condition | 0.1716 | 1 | 0 | 0.1716 | 6.8126 | 0.0099 |
| Concentration | 13.1391 | 3 | 0 | 4.3797 | 173.9008 | 0 |
| Sex*Condition | 0.0181 | 1 | 0 | 0.0181 | 0.7167 | 0.3985 |
| Sex*Concentration | 0.209 | 3 | 0 | 0.0697 | 2.7663 | 0.0437 |
| Condition*Concentration | 0.1103 | 3 | 0 | 0.0368 | 1.4593 | 0.2278 |
| Error | 4.0044 | 159 | 0 | 0.0252 | NaN | NaN |
| Total | 17.9639 | 171 | 0 | NaN | NaN | NaN |

**Initial Task vs. Late Task:**
**3-way ANOVA results**

| Source | Sum sq. | d.f. | Singular? | Mean Sq. | F | Prob > F |
|---|---|---|---|---|---|---|
| Sex | 0.1824 | 1 | 0 | 0.1824 | 4.6943 | 0.0318 |
| Condition | 0.0001 | 1 | 0 | 0.0001 | 0.0013 | 0.971 |
| Concentration | 12.9611 | 3 | 0 | 4.3204 | 111.1738 | 0 |
| Sex*Condition | 0.0448 | 1 | 0 | 0.0448 | 1.1539 | 0.2844 |
| Sex*Concentration | 0.085 | 3 | 0 | 0.0283 | 0.7288 | 0.5363 |
| Condition*Concentration | 0.1284 | 3 | 0 | 0.0428 | 1.1015 | 0.3504 |
| Error | 6.0235 | 155 | 0 | 0.0389 | NaN | NaN |
| Total | 19.5583 | 167 | 0 | NaN | NaN | NaN |

**Fig. 7d: Initial task Approach time (FvM)**

Statistical significance was determined by Repeated measures analysis of variance. (Female $N = 12$, Male $N = 10$). Effect of concentration: d.f. = 3, $F = 3.0746$, $p = 5.4017e{-}02$. Effect of sex: d.f. = 1, $F = 0.7937$, $p = 0.4073$. kstest2 results: $h = 0$, $p = 4.6263e{-}01$, ks2stat=0.2016 (overall sex difference).

Post-hoc analysis: 0.5%: 7.1363e−01, 2%: 7.6787e−01, 5%: 7.5464e−02, 9%: 6.6465e−01.

KStest2 and Wilcoxon rank sum test Results (complementary to post-hoc analysis): KStest2: Conc1: $h = 0$, $p = 0.2141$, ks2stat = 0.5000. RStest: Conc1: $h = 0$, $p = 0.3462$. KStest2: Conc2: $h = 0$, $p = 0.3180$, ks2stat = 0.5000. RStest: Conc2: $h = 0$, $p = 0.3701$. KStest2: Conc3: $h = 1$, $p = 0.0032$, ks2stat = 0.7167. RStest: Conc3: $h = 1$, $p = 0.0192$. KStest2: Conc4: $h = 0$, $p = 0.2503$, ks2stat = 0.4545. RStest: Conc4: $h = 0$, $p = 0.5360$.

**Control vs. Initial Task Female:**

Statistical significance was determined by Repeated measures analysis of variance. (Control $N = 12$, Late Task $N = 12$). $p$-value for Control vs. initial task of female: 0.95393. kstest2 results: $h = 0$, $p = 1.0029e{-}01$, ks2stat = 0.2570.

Post-hoc analysis: 0.5%: 6.0495e−01, 2%: 7.3330e−02, 5%: 1.0644e−02, 9%: 6.9342e−01.

KStest2 and Wilcoxon rank sum test Results (complementary to post-hoc analysis): KStest2: Conc1: $h = 0$, $p = 0.0876$, ks2stat = 0.5091. RStest: Conc1: $h = 0$, $p = 0.5968$. KStest2: Conc2: $h = 0$, $p = 0.0799$, ks2stat = 0.5833. RStest: Conc2: $h = 1$, $p = 0.0320$. KStest2: Conc3: $h = 0$, $p = 0.0656$, ks2stat =

0.5000. RStest: Conc3: $h = 0$, $p = 0.0606$. KStest2: Conc4: $h = 0$, $p = 0.2812$, ks2stat = 0.3864. RStest: Conc4: $h = 0$, $p = 0.3099$.

**Male:** Statistical significance was determined by Repeated measures analysis of variance. (Control $N = 11$, Late Task $N = 10$). $p$-value for Control vs. initial task of male: 0.3337. kstest2 results: $h = 0$, $p = 3.8605e{-}01$, ks2stat = 0.2110.

Post-hoc analysis: 0.5%: 7.3182e−01, 2%: 1.6792e−01, 5%: 9.4946e−01, 9%: 6.1620e−01.

KStest2 and Wilcoxon rank sum test Results (complementary to post-hoc analysis): KStest2: Conc1: $h = 0$, $p = 0.2290$, ks2stat = 0.4848. RStest: Conc1: $h = 0$, $p = 0.2696$. KStest2: Conc2: $h = 0$, $p$.

**3-way ANOVA results**

| Source | Sum sq. | d.f. | Singular? | Mean Sq. | F | Prob > F |
|---|---|---|---|---|---|---|
| Sex | 0.0057 | 1 | 0 | 0.0057 | 0.0008 | 0.9775 |
| Condition | 4.1585 | 1 | 0 | 4.1585 | 0.5848 | 0.4457 |
| Concentration | 213.9496 | 3 | 0 | 71.3165 | 10.0285 | 0 |
| Sex*Condition | 9.3398 | 1 | 0 | 9.3398 | 1.3134 | 0.2537 |
| Sex*Concentration | 27.986 | 3 | 0 | 9.3287 | 1.3118 | 0.2729 |
| Condition*Concentration | 9.7491 | 3 | 0 | 3.2497 | 0.457 | 0.7128 |
| Error | 1024.0377 | 144 | 0 | 7.1114 | NaN | NaN |
| Total | 1296.7682 | 156 | 0 | NaN | NaN | NaN |

**Fig. 7e: Late task Approach time (FvM)**

Statistical significance was determined by Repeated measures analysis of variance. (Female $N = 10$, Male $N = 10$). Effect of concentration: d.f. = 3, $F = 1.0838$, $p = 3.8602e{-}01$. Effect of sex: d.f. = 1, $F = 0.1609$, $p = 0.7049$. kstest2 results: $h = 0$, $p = 1.7336e{-}01$, ks2stat = 0.2745 (overall sex difference).

Post-hoc analysis: 0.5%: 5.2868e−01, 2%: 4.4209e−01, 5%: 5.2319e−01, 9%: 4.8906e−01.

KStest2 and Wilcoxon rank sum test Results (complementary to post-hoc analysis): KStest2: Conc1: $h = 1$, $p = 0.0204$, ks2stat = 0.8000. RStest: Conc1: $h = 1$, $p = 0.0303$. KStest2: Conc2: $h = 0$, $p = 0.5070$, ks2stat = 0.4250. RStest: Conc2: $h = 0$, $p = 0.2844$. KStest2: Conc3: $h = 0$, $p = 0.4892$, ks2stat = 0.3556. RStest: Conc3: $h = 0$, $p = 0.3562$. KStest2: Conc4: $h = 0$, $p = 0.1076$, ks2stat = 0.5417. RStest: Conc4: $h = 1$, $p = 0.0274$.

**Control vs. Late Task Female:**

Statistical significance was determined by Repeated measures analysis of variance. (Control N = 12, Late Task $N = 10$). $p$-value for Control vs. initial task of female: 0.58585. kstest2 results: $h = 0$, $p = 2.9353e{-}01$, ks2stat = 0.2134.

Post-hoc analysis: 0.5%: 6.6412e−01, 2%: 7.3010e−01, 5%: 9.4083e−02, 9%: 2.9966e−01.

KStest2 and Wilcoxon rank sum test Results (complementary to post-hoc analysis): KStest2: Conc1: $h = 1$, $p = 0.0204$, ks2stat = 0.8000 RStest: Conc1: $h = 1$, $p = 0.0303$. KStest2: Conc2: $h = 0$, $p = 0.5070$, ks2stat = 0.4250 RStest: Conc2: $h = 0$, $p = 0.2844$. KStest2: Conc3: $h = 0$, $p = 0.4892$, ks2stat = 0.3556 RStest: Conc3: $h = 0$, $p = 0.3562$. KStest2: Conc4: $h = 0$, $p = 0.1076$, ks2stat = 0.5417, RStest: Conc4: $h = 1$, $p = 0.0274$.

**Male**: Statistical significance was determined by Repeated measures analysis of variance. (Control $N = 11$, Late Task $N = 10$). $p$-value for Control vs. initial task of male: 0.74017. kstest2 results: $h = 0$, $p = 2.2511e{-}01$, ks2stat = 0.2487. Post-hoc analysis: 0.5%: 6.3673e−01, 2%: 4.9689e−01, 5%: 8.2142e−01, 9%: 5.9077e−01.

KStest2 and Wilcoxon rank sum test Results (complementary to post-hoc analysis): KStest2: Conc1: $h = 0$, $p = 0.1019$, ks2stat = 0.6000. RStest: Conc1: $h = 0$, $p = 0.1149$. KStest2: Conc2: $h = 0$, $p = 0.5402$, ks2stat = 0.4000. RStest: Conc2: $h = 0$, $p = 0.3710$. KStest2: Conc3: $h = 0$, $p = 0.4114$, ks2stat =

0.3778. RStest: Conc3: $h = 0$, $p = 0.4470$. KStest2: Conc4: $h = 0$, $p = 0.2147$, ks2stat = 0.4545. RStest: Conc4: $h = 0$, $p = 0.1518$.

### 3-way ANOVA Results

| Source | Sum Sq. | d.f. | Singular? | Mean Sq. | F | Prob > F |
|---|---|---|---|---|---|---|
| Sex | 23.2272 | 1 | 0 | 23.2272 | 4.5491 | 0.0347 |
| Condition | 0.258 | 1 | 0 | 0.258 | 0.0505 | 0.8225 |
| Concentration | 255.2297 | 3 | 0 | 85.0766 | 16.6626 | 0 |
| Sex*Condition | 3.2778 | 1 | 0 | 3.2778 | 0.642 | 0.4244 |
| Sex*Concentration | 28.1551 | 3 | 0 | 9.385 | 1.8381 | 0.1431 |
| Condition*Concentration | 11.3002 | 3 | 0 | 3.7667 | 0.7377 | 0.5313 |
| Error | 699.4998 | 137 | 0 | 5.1058 | NaN | NaN |
| Total | 1000.7338 | 149 | 0 | NaN | NaN | NaN |

**Fig. 7f: Initial task Number of high sp. runs (FvM)**

Statistical significance was determined by Repeated measures analysis of variance. (Female $N = 12$, Male $N = 10$). Effect of concentration: d.f. = 3, $F = 1.8924$, $p = 1.4050e{-}01$. Effect of sex: d.f. = 1, $F = 0.8500$, $p = 0.3675$. kstest2 results: $h = 0$, $p = 1.2139e{-}01$, ks2stat = 0.2458 (overall sex difference). Post-hoc analysis: 0.5%: 7.7450e{-}01, 2%: 9.1269e{-}01, 5%: 3.6064e{-}03, 9%: 5.6181e{-}01.

KStest2 and Wilcoxon rank sum test Results (complementary to post-hoc analysis): KStest2: Conc1: $h = 0$, $p = 0.9304$, ks2stat = 0.2167. RStest: Conc1: $h = 0$, $p = 0.7667$. KStest2: Conc2: $h = 0$, $p = 0.8286$, ks2stat = 0.2500. RStest: Conc2: $h = 0$, $p = 0.7667$. KStest2: Conc3: $h = 1$, $p = 0.0101$, ks2stat = 0.6500. RStest: Conc3: $h = 1$, $p = 0.0111$. KStest2: Conc4: $h = 0$, $p = 0.1072$, ks2stat = 0.4833. RStest: Conc4: $h = 0$, $p = 0.3734$.

**Control vs. Initial Task Female:** Statistical significance was determined by Repeated measures analysis of variance. (Control $N = 12$, Initial Task $N = 12$). $p$-value for Control vs. initial task of female: 0.013. kstest2 results: $h = 1$, $p = 4.8054e{-}02$, ks2stat = 0.2708. Post-hoc analysis: 0.5%: 8.0719e{-}02, 2%: 1.3341e{-}02, 5%: 3.6748e{-}02, 9%: 7.0275e{-}01.

KStest2 and Wilcoxon rank sum test Results (complementary to post-hoc analysis): KStest2: Conc1: $h = 0$, $p = 0.0656$, ks2stat = 0.5000. RStest: Conc1: $h = 1$, $p = 0.0351$. KStest2: Conc2: $h = 1$, $p = 0.0191$, ks2stat = 0.5833. RStest: Conc2: $h = 1$, $p = 0.0086$. KStest2: Conc3: $h = 0$, $p = 0.0656$, ks2stat = 0.5000. RStest: Conc3: $h = 1$, $p = 0.0262$. KStest2: Conc4: $h = 0$, $p = 0.1862$, ks2stat = 0.4167. RStest: Conc4: $h = 0$, $p = 0.5067$.

**Male:** Statistical significance was determined by Repeated measures analysis of variance. (Control $N = 11$, Initial Task $N = 10$). $p$-value for Control vs. initial task of male: 0.49907. kstest2 results: $h = 0$, $p = 5.7150e{-}01$, ks2stat = 0.1659. Post-hoc analysis: 0.5%: 8.6648e{-}01, 2%: 7.4849e{-}01, 5%: 1.6437e{-}02, 9%: 7.9965e{-}01.

KStest2: Conc1: $h = 0$, $p = 0.2646$, ks2stat = 0.4091 RStest: Conc1: $h = 0$, $p = 0.5035$. KStest2: Conc2: $h = 0$, $p = 0.6114$, ks2stat = 0.3091 RStest: Conc2: $h = 0$, $p = 0.6985$. KStest2: Conc3: $h = 1$, $p = 0.0198$, ks2stat = 0.6182 RStest: Conc3: $h = 1$, $p = 0.0183$ KStest2: Conc4: $h = 0$, $p = 0.8290$, ks2stat = 0.2545 RStest: Conc4: $h = 0$, $p = 0.9719$.

### 3-way ANOVA results

| Source | Sum Sq. | d.f. | Singular? | Mean sq. | F | Prob > F |
|---|---|---|---|---|---|---|
| Sex | 380.4236 | 1 | 0 | 380.4236 | 13.5543 | 0.0003 |
| Condition | 258.1331 | 1 | 0 | 258.1331 | 9.1971 | 0.0028 |
| Concentration | 336.9166 | 3 | 0 | 112.3055 | 4.0014 | 0.0088 |
| Sex*Condition | 73.4858 | 1 | 0 | 73.4858 | 2.6183 | 0.1075 |
| Sex*Concentration | 76.5402 | 3 | 0 | 25.5134 | 0.909 | 0.438 |
| Condition*Concentration | 88.7222 | 3 | 0 | 29.5741 | 1.0537 | 0.3704 |
| Error | 4687.1338 | 167 | 0 | 28.0667 | NaN | NaN |
| Total | 5899.4658 | 179 | 0 | NaN | NaN | NaN |

**Fig. 7g: Late task Number of high sp. runs (FvM)**

Statistical significance was determined by Repeated measures analysis of variance. (Female $N = 10$, Male $N = 10$). Effect of concentration: d.f. = 3, $F = 0.5016$, $p = 6.8275e{-}01$. Effect of sex: d.f. = 1, $F = 1.8180$, $p = 0.1943$. kstest2 results: $h = 0$, $p = 1.3925e{-}01$, ks2stat = 0.2500 (overall sex difference). Post-hoc analysis: 0.5%: 4.4173e{-}01, 2%: 1.5443e{-}01, 5%: 4.3851e{-}01, 9%: 8.8666e{-}02.

KStest2 and Wilcoxon rank sum test Results (complementary to post-hoc analysis). KStest2: Conc1: $h = 0$, $p = 0.3129$, ks2stat = 0.4000. RStest: Conc1: $h = 0$, $p = 0.3075$. KStest2: Conc2: $h = 0$, $p = 0.6751$, ks2stat = 0.3000. RStest: Conc2: $h = 0$, $p = 0.1620$. KStest2: Conc3: $h = 0$, $p = 0.6751$, ks2stat = 0.3000. RStest: Conc3: $h = 0$, $p = 0.9698$. KStest2: Conc4: $h = 0$, $p = 0.1108$, ks2stat = 0.5000. RStest: Conc4: $h = 0$, $p = 0.0890$.

**Control vs. Late Task Female:** Statistical significance was determined by Repeated measures analysis of variance. (Control $N = 12$, Late Task $N = 10$). $p$-value for Control vs. late task of female: 0.062741. kstest2 results: $h = 0$, $p = 1.6019e{-}01$, ks2stat = 0.2333. Post-hoc analysis: 0.5%: 1.4081e{-}01, 2%: 8.1374e{-}03, 5%: 1.3291e{-}02, 9%: 4.2351e{-}01.

KStest2 and Wilcoxon rank sum test Results (complementary to post-hoc analysis). KStest2: Conc1: $h = 0$, $p = 0.6259$, ks2stat = 0.3000. RStest: Conc1: $h = 0$, $p = 0.3390$. KStest2: Conc2: $h = 1$, $p = 0.0076$, ks2stat = 0.6667. RStest: Conc2: $h = 1$, $p = 0.0062$. KStest2: Conc3: $h = 0$, $p = 0.1902$, ks2stat = 0.4333. RStest: Conc3: $h = 1$, $p = 0.0321$. KStest2: Conc4: $h = 0$, $p = 0.2270$, ks2stat = 0.4167. RStest: Conc4: $h = 0$, $p = 0.2485$.

**Male:** Statistical significance was determined by Repeated measures analysis of variance. (Control $N = 11$, Late Task $N = 10$). $p$-value for Control vs. late task of male: 0.36436. kstest2 results: $h = 0$, $p = 1.1714e{-}01$, ks2stat = 0.2523. Post-hoc analysis: 0.5%: 6.9166e{-}01, 2%: 5.0927e{-}02, 5%: 5.4199e{-}01, 9%: 5.5656e{-}01.

KStest2 and Wilcoxon rank sum test Results (complementary to post-hoc analysis): KStest2: Conc1: $h = 0$, $p = 0.2646$, ks2stat = 0.4091, RStest: Conc1: $h = 0$, $p = 0.4597$. KStest2: Conc2: $h = 0$, $p$

### 3-way ANOVA results

| Source | Sum sq. | d.f. | Singular? | Mean sq. | F | Prob > F |
|---|---|---|---|---|---|---|
| Sex | 744.2571 | 1 | 0 | 744.2571 | 21.8951 | 0 |
| Condition | 306.8216 | 1 | 0 | 306.8216 | 9.0263 | 0.0031 |
| Concentration | 117.5248 | 3 | 0 | 39.1749 | 1.1525 | 0.3298 |
| Sex*Condition | 0.0267 | 1 | 0 | 0.0267 | 0.0008 | 0.9777 |
| Sex*Concentration | 8.8943 | 3 | 0 | 2.9648 | 0.0872 | 0.967 |
| Condition*Concentration | 109.5021 | 3 | 0 | 36.5007 | 1.0738 | 0.3619 |
| Error | 5404.726 | 159 | 0 | 33.992 | NaN | NaN |
| Total | 6729.4241 | 171 | 0 | NaN | NaN | NaN |

**Fig. 7h: Initial task Distance traveled (FvM)**

Statistical significance was determined by Repeated measures analysis of variance. (Female $N = 12$, Male $N = 10$). Effect of concentration: d.f. = 3, $F = 2.4366$, $p = 7.3371e{-}02$. Effect of sex: d.f. = 1, $F = 17.9034$, $p = 0.0004$. kstest2 results: $h = 1$, $p = 2.4759e{-}06$, ks2stat = 0.5417 (overall sex difference). Post-hoc analysis: 0.5%: 1.2714e{-}02, 2%: 5.9126e{-}02, 5%: 8.5187e{-}04, 9%: 1.1225e{-}03.

KStest2 and Wilcoxon rank sum test Results (complementary to post-hoc analysis): KStest2: Conc1: $h = 1$, $p = 0.0220$, ks2stat = 0.6000 RStest: Conc1: $h = 1$, $p = 0.0092$ KStest2: Conc2: $h = 0$, $p = 0.0567$, ks2stat = 0.5333 RStest: Conc2: $h = 0$, $p = 0.0806$ KStest2: Conc3: $h = 1$, $p = 0.0076$, ks2stat = 0.6667 RStest: Conc2: $h = 1$, $p = 0.0022$ KStest2: Conc4: $h = 1$, $p = 0.0076$, ks2stat = 0.6667 RStest: Conc4: $h = 1$, $p = 0.0041$.

**Control vs. Initial Task Female:**

Statistical significance was determined by Repeated measures analysis of variance. (Control $N = 12$, Initial Task $N = 12$). $p$-value for Control vs initial task of female: 0.01429. kstest2 results: $h = 1$, $p = 4.6080e{-}05$, ks2stat

= 0.4583. Post-hoc analysis: 0.5%: 6.7266e−02, 2%: 1.6070e−02, 5%: 3.1394e−01, 9%: 9.0362e−03.

KStest2 and Wilcoxon rank sum test Results (complementary to post-hoc analysis): KStest2: Conc1: $h = 0$, $p = 0.0656$, ks2stat = 0.5000. RStest: Conc1: $h = 0$, $p = 0.0999$. KStest2: Conc2: $h = 1$, $p = 0.0191$, ks2stat = 0.5833. RStest: Conc2: $h = 1$, $p = 0.0304$. KStest2: Conc3: $h = 0$, $p = 0.0656$, ks2stat = 0.5000. RStest: Conc3: $h = 0$, $p = 0.1749$. KStest2: Conc4: $h = 0$, $p = 0.0656$, ks2stat = 0.5000. RStest: Conc4: $h = 1$, $p = 0.0086$.

**Male**: Statistical significance was determined by Repeated measures analysis of variance. (Control $N = 11$, Initial Task $N = 10$). $p$-value for Control vs. initial task of male: 0.013586. kstest2 results: $h = 1$, $p = 1.3560e−03$, ks2stat = 0.4045. Post-hoc analysis: 0.5%: 1.3573e−02, 2%: 3.4304e−02, 5%: 4.5088e−02, 9%: 5.5697e−02.

KStest2 and Wilcoxon rank sum test Results (complementary to post-hoc analysis): KStest2: Conc1: $h = 0$, $p = 0.0782$, ks2stat = 0.5182. RStest: Conc1: $h = 1$, $p = 0.0221$. KStest2: Conc2: $h = 1$, $p = 0.0259$, ks2stat = 0.6000. RStest: Conc2: $h = 1$, $p = 0.0265$. KStest2: Conc3: $h = 0$, $p = 0.2006$, ks2stat = 0.4364. RStest: Conc3: $h = 0$, $p = 0.0528$. KStest2: Conc4: $h = 0$, $p = 0.2418$, ks2stat = 0.4182. RStest: Conc4: $h = 0$, $p = 0.1300$.

**3-way ANOVA results**

| Source | Sum Sq. | d.f. | Singular? | Mean sq. | F | Prob > F |
|---|---|---|---|---|---|---|
| Sex | 17.7228 | 1 | 0 | 17.7228 | 75.0599 | 0 |
| Condition | 8.437 | 1 | 0 | 8.437 | 35.7328 | 0 |
| Concentration | 1.1351 | 3 | 0 | 0.3784 | 1.6025 | 0.1907 |
| Sex*Condition | 0.0679 | 1 | 0 | 0.0679 | 0.2874 | 0.5926 |
| Sex*Concentration | 0.6284 | 3 | 0 | 0.2095 | 0.8872 | 0.4491 |
| Condition*Concentration | 0.6088 | 3 | 0 | 0.2029 | 0.8595 | 0.4634 |
| Error | 39.4312 | 167 | 0 | 0.2361 | NaN | NaN |
| Total | 68.6863 | 179 | 0 | NaN | NaN | NaN |

**Fig. 7i: Late task distance traveled (FvM)**

Statistical significance was determined by Repeated measures analysis of variance. (Female N = 10, Male $N = 10$). Effect of concentration: d.f. = 3, $F = 0.7077$, $p = 5.5161e−01$. Effect of sex: d.f. = 1, $F = 4.6430$, $p = 0.0450$. kstest2 results: $h = 1$, $p = 1.0793e−02$, ks2stat = 0.3500 (overall sex difference).

Post-hoc analysis: 0.5%: 9.3255e−02, 2%: 8.9902e−02, 5%: 5.4953e−01, 9%: 6.7305e−03.

KStest2 and Wilcoxon rank sum test Results (complementary to post-hoc analysis): KStest2: Conc1: $h = 1$, $p = 0.0310$, ks2stat = 0.6000. RStest: Conc1: $h = 0$, $p = 0.0890$. KStest2: Conc2: $h = 0$, $p = 0.3129$, ks2stat = 0.4000. RStest: Conc2: $h = 0$, $p = 0.1212$. KStest2: Conc3: $h = 0$, $p = 0.3129$, ks2stat = 0.4000. RStest: Conc3: $h = 0$, $p = 0.6232$. KStest2: Conc4: $h = 1$, $p = 0.0069$, ks2stat = 0.7000. RStest: Conc4: $h = 1$, $p = 0.0113$.

**Control vs. Late Task Female:**

Statistical significance was determined by Repeated measures analysis of variance. (Control $N = 12$, Late Task $N = 10$). $p$-value for Control vs. late task of female: 0.72859. kstest2 results: $h = 0$, $p = 3.6205e−01$, ks2stat = 0.1917.

Post-hoc analysis: 0.5%: 8.6636e−01, 2%: 5.2765e−01, 5%: 4.3035e−01, 9%: 2.4643e−01.

KStest2 and Wilcoxon rank sum test Results (complementary to post-hoc analysis): KStest2: Conc1: $h = 0$, $p = 0.7647$, ks2stat = 0.2667 RStest: Conc1: $h = 0$, $p = 0.6682$. KStest2: Conc2: $h = 0$, $p = 0.6259$, ks2stat = 0.3000 RStest: Conc2: $h = 0$, $p = 0.5310$. KStest2: Conc3: $h = 0$, $p = 0.4896$, ks2stat = 0.3333 RStest: Conc3: $h = 0$, $p = 0.5752$. KStest2: Conc4: $h = 0$, $p = 0.3689$, ks2stat = 0.3667 RStest: Conc4: $h = 0$, $p = 0.1985$.

**Male**: Statistical significance was determined by Repeated measures analysis of variance. (Control $N = 11$, Late Task $N = 10$). $p$-value for Control vs. late task of male: 0.13682. kstest2 results: $h = 1$, $p = 1.1240e−02$, ks2stat = 0.3409.

Post-hoc analysis: 0.5%: 9.0183e−02, 2%: 5.1266e−01, 5%: 9.7800e−02, 9%: 2.6146e−01. KStest2 and Wilcoxon rank sum test Results (complementary to post-hoc analysis). KStest2: Conc1: $h = 0$, $p = 0.0876$, ks2stat = 0.5091. RStest: Conc1: $h = 0$, $p = 0.0845$. KStest2: Conc2: $h = 0$, $p = 0.2890$, ks2stat = 0.4000. RStest: Conc2: $h = 0$, $p = 0.5035$. KStest2: Conc3: $h = 0$, $p = 0.0978$, ks2stat = 0.5000. RStest: Conc3: $h = 0$, $p = 0.2178$. KStest2: Conc4: $h = 0$, $p = 0.5742$, ks2stat = 0.3182. RStest: Conc4: $h = 0$, $p = 0.4181$.

**3-way ANOVA results**

| Source | Sum Sq. | d.f. | Singular? | Mean Sq. | F | Prob > F |
|---|---|---|---|---|---|---|
| Sex | 10.6559 | 1 | 0 | 10.6559 | 46.8601 | 0 |
| Condition | 1.0952 | 1 | 0 | 1.0952 | 4.8163 | 0.0296 |
| Concentration | 0.376 | 3 | 0 | 0.1253 | 0.5512 | 0.6481 |
| Sex*Condition | 0.3553 | 1 | 0 | 0.3553 | 1.5624 | 0.2131 |
| Sex*Concentration | 0.2714 | 3 | 0 | 0.0905 | 0.3978 | 0.7548 |
| Condition*Concentration | 0.1164 | 3 | 0 | 0.0388 | 0.1706 | 0.9161 |
| Error | 36.1564 | 159 | 0 | 0.2274 | NaN | NaN |
| Total | 49.1756 | 171 | 0 | NaN | NaN | NaN |

**Fig. 7j: Fraction of sigmoid (Control vs Alcohol)**

Statistical significance was determined by one-way analysis of variance. (Female Control = 10, Male. Control = 10, Female Late task = 10, Male Late task = 10). Significance of difference between the groups: d.f. = 3, $F = 2.4814$, $p = 0.0766$. Post-hoc analysis by Tukey's HSD method: Female Control and Male Control: 0.8723. Female Control and Female Late task: 0.9123. Male Control and Male Late task: 0.0579

**Figure 7k: Cluster shifts (Control vs. Alcohol)**

Statistical significance was determined by Chi-squared test. The significance of difference in population in cluster 1, 2 and 3 is 0.0232, 0.1753 and 0.0003, respectively.

**Figure 7o: Baseline and alcohol individual rat Euclidian distance**

Statistical significance $p = 8.9133e−06$, determined by two-sample Kolmogorov–Smirnov test. (Control $N = 23$, Alcohol $N = 20$).

**Figure 7p: Baseline and alcohol early vs. late bins Euclidian distance**

Statistical significance $p = 0.00105$, determined by two-sample Kolmogorov–Smirnov test. (Control $N = 37$, Alcohol $N = 20$)

**Supplemental Fig. 1**

**Figure S.1g: Average time to learn task (FvM)**

statistical significance was determined by paired $t$-test using SPSS software package ($F = 12$, $M = 11$). $p$-vale for sex difference: 0.01.

**Supplemental** Fig. 2

**Figure S.2i: Control distance traveled, approach only (FvM)**

Statistical significance was determined by Repeated measures analysis of variance. (Female $N = 12$, Male $N = 11$). Effect of concentration: d.f. = 3, $F = 7.9201$, $p = 1.6745e−04$. Effect of sex: d.f. = 1, $F = 10.8854$, $p = 0.0038$. kstest2 results: $h = 1$, $p = 7.1912e−04$, ks2stat = 0.4103 (overall sex difference). Post-hoc analysis: 0.5%: 1.2870e−02, 2%: 2.5797e−01, 5%: 2.5247e−02, 9%: 1.7835e−02.

KStest2 and Wilcoxon rank sum test Results (complementary to post-hoc analysis): KStest2: Conc1: $h = 1$, $p = 0.0121$, ks2stat = 0.6364. RStest: Conc1: $h = 1$, $p = 0.0104$. KStest2: Conc2: $h = 0$, $p = 0.2270$, ks2stat = 0.4167. RStest: Conc2: $h = 0$, $p = 0.3734$. KStest2: Conc2: $h = 1$, $p = 0.0452$, ks2stat = 0.5500. RStest: Conc2: $h = 1$, $p = 0.0192$. KStest2: Conc4: $h = 1$, $p = 0.0289$, ks2stat = 0.5682. RStest: Conc4: $h = 1$, $p = 0.0074$.

**Figure S.2j: Control number of stopping points, approach only (FvM)**

Statistical significance was determined by Repeated measures analysis of variance. (Female $N = 12$, Male $N = 11$). Effect of concentration: d.f. = 3, $F = 5.8431$, $p = 1.4948e−03$. Effect of sex: d.f. = 1, $F = 0.5058$, $p = 0.4856$.

kstest2 results: $h = 0$, $p = 6.2873e-02$, ks2stat = 0.2710 (overall sex difference) post-hoc analysis: 0.5%: 5.8083e−02 2%: 8.8005e−01, 5%: 5.5419e−01, 9%: 1.6653e−01. KStest2 and Wilcoxon rank sum test Results (complementary to post-hoc analysis). KStest2: Conc1: $h = 0$, $p = 0.1473$, ks2stat = 0.4545. RStest: Conc1: $h = 0$, $p = 0.0878$. KStest2: Conc2: $h = 0$, $p = 0.4896$, ks2stat = 0.3333. RStest: Conc2: $h = 0$, $p = 0.5310$. KStest2: Conc2: $h = 0$, $p = 0.2689$, ks2stat = 0.4000. RStest: Conc2: $h = 0$, $p = 0.1985$. KStest2: Conc4: $h = 0$, $p = 0.0915$, ks2stat = 0.4848. RStest: Conc4: $h = 0$, $p = 0.0905$.

**Figure S.2k: Control number of high sp. runs, approach only (FvM)**
Statistical significance was determined by Repeated measures analysis of variance. (Female $N = 12$, Male $N = 11$). Effect of concentration: d.f. = 3, $F = 33.6668$, $p = 1.1834e-12$. Effect of sex: d.f. = 1, $F = 1.6147$, $p = 0.2192$. kstest2 results: $h = 0$, $p = 4.3505e-01$, ks2stat = 0.1793 (overall sex difference). Post-hoc analysis: 0.5%: 1.9769e−01, 2%: 5.3803e−01, 5%: 4.1293e−01, 9%: 1.8367e−01.

KStest2 and Wilcoxon rank sum test Results (complementary to post-hoc analysis): KStest2: Conc1: $h = 0$, $p = 0.3744$, ks2stat = 0.3636 RStest: Conc1: $h = 0$, $p = 0.1486$ KStest2: Conc2: $h = 0$, $p = 0.8286$, ks2stat = 0.2500 RStest: Conc2: $h = 0$, $p = 0.9212$ KStest2: Conc2: $h = 0$, $p = 0.4268$, ks2stat = 0.3500 RStest: Conc2: $h = 0$, $p = 0.1985$ KStest2: Conc4: $h = 0$, $p = 0.1328$, ks2stat = 0.4545 RStest: Conc4: $h = 0$, $p = 0.1661$.

**Figure S.2l: Control prop. of trial out. all reward zones, approach only (FvM)**
Statistical significance was determined by Repeated measures analysis of variance. (Female $N = 12$, Male $N = 11$). Effect of concentration: d.f. = 3, $F = 7.3360$, $p = 3.0532e-04$. Effect of sex: d.f. = 1, $F = 0.0233$, $p = 0.8803$. kstest2 results: $h = 0$, $p = 7.5063e-01$, ks2stat = 0.1393 (overall sex difference). Post-hoc analysis: 0.5%: 6.6119e−01, 2%: 4.6695e−01, 5%: 5.8985e−01, 9%: 6.5657e−01.

KStest2 and Wilcoxon rank sum test Results (complementary to post-hoc analysis): KStest2: Conc1: $h = 0$, $p = 0.9852$, ks2stat = 0.1818. RStest: Conc1: $h = 0$, $p = 0.8422$. KStest2: Conc2: $h = 0$, $p = 0.8848$, ks2stat = 0.2333. RStest: Conc2: $h = 0$, $p = 0.3834$. KStest2: Conc2: $h = 0$, $p = 0.5564$, ks2stat = 0.3167. RStest: Conc2: $h = 0$, $p = 0.4681$. KStest2: Conc4: $h = 0$, $p = 0.9610$, ks2stat = 0.1970. RStest: Conc4: $h = 0$, $p = 0.5588$.

**Figure S.2m. Control distance traveled, reject only (FvM)**
Statistical significance was determined by Repeated measures analysis of variance. (Female $N = 12$, Male $N = 11$). Effect of concentration: d.f. = 3, $F = 2.7997$, $p = 4.7581e-02$. Effect of sex: d.f. = 1, $F = 7.3378$, $p = 0.0135$. kstest2 results: $h = 1$, $p = 2.2098e-04$, ks2stat = 0.4351 (overall sex difference). Post-hoc analysis: 0.5%: 0.0048, 2%: 0.0168, 5%: 0.0162, 9%: 0.1742.

KStest2 and Wilcoxon rank sum test Results (complementary to post-hoc analysis): KStest2: Conc1: $h = 1$, $p = 0.0289$, ks2stat = 0.5682. RStest: Conc1: $h = 1$, $p = 0.0074$. KStest2: Conc2: h = 1, $p = 0.0361$, ks2stat = 0.5530, RStest: Conc2: $h = 1$, $p = 0.0127$. KStest2: Conc2: $h = 1$, $p = 0.0403$, ks2stat = 0.5455. RStest: Conc2: $h = 1$, $p = 0.0089$. KStest2: Conc4: $h = 0$, $p = 0.2270$, ks2stat = 0.4167. RStest: Conc4: $h = 0$, $p = 0.2766$.

**Figure S.2n. Control number of stopping points, reject only (FvM)**
Effect of concentration: d.f. = 3, $F = 1.4054$, $p = 2.5002e-01$. Effect of sex: d.f. = 1, $F = 1.5795$, $p = 0.2233$. kstest2 results: $h = 1$, $p = 2.3013e-04$, ks2stat = 0.4341 (overall sex difference). Post-hoc analysis: 0.5%: 0.0987, 2%: 0.2915, 5%: 0.1825, 9%: 0.3455.

KStest2 and Wilcoxon rank sum test Results (complementary to post-hoc analysis): KStest2: Conc1: $h = 1$, $p = 0.0289$, ks2stat = 0.5682 RStest: Conc1: $h = 1$, $p = 0.0062$ KStest2: Conc2: $h = 0$, $p = 0.1006$, ks2stat = 0.4773 RStest: Conc2: $h = 0$, $p = 0.1316$ KStest2: Conc2: $h = 1$, $p = 0.0361$, ks2stat = 0.5530 RStest: Conc2: $h = 1$, $p = 0.0089$ KStest2: Conc4: $h = 0$, $p = 0.3162$, ks2stat = 0.3833 RStest: Conc4: $h = 0$, $p = 0.3390$.

**Figure S.2o. Control number of high sp. runs, reject only (FvM)**
Effect of concentration: d.f. = 3, $F = 1.7438$, $p = 1.6766e-01$. Effect of sex: d.f. = 1, F = 12.3666, p = 0.0022. kstest2 results: $h = 1$, $p = 4.1576e-07$, ks2stat = 0.5654 (overall sex difference). Post-hoc analysis: 0.5%: 0.0003, 2%: 0.0025, 5%: 0.0054, 9%: 0.0742.

KStest2 and Wilcoxon rank sum test Results (complementary to post-hoc analysis): KStest2: Conc1: $h = 1$, $p = 0.0003$, ks2stat = 0.8258 RStest: Conc1: $h = 1$, $p = 0.0006$ KStest2: Conc2: $h = 1$, $p = 0.0059$, ks2stat = 0.6667 RStest: Conc2: $h = 1$, $p = 0.0028$ KStest2: Conc2: $h = 1$, $p = 0.0098$, ks2stat = 0.6364 RStest: Conc2: $h = 1$, $p = 0.0062$ KStest2: Conc4: $h = 0$, $p = 0.1902$, ks2stat = 0.4333 RStest: Conc4: $h = 0$, $p = 0.0806$

**Figure S.2p. Control prop. of trial out. all reward zones, reject only (FvM)**
Effect of concentration: d.f. = 3, $F = 1.0197$, $p = 3.9035e-01$. Effect of sex: d.f. = 1, $F = 1.2575$, $p = 0.2754$. kstest2 results: h=0, $p = 6.1613e-02$, ks2stat=0.2689 (overall sex difference). Post-hoc analysis: 0.5%: 0.4147, 2%: 0.7963, 5%: 0.0463, 9%: 0.7952

KStest2 and Wilcoxon rank sum test Results (complementary to post-hoc analysis): KStest2: Conc1: $h = 0$, $p = 0.7136$, ks2stat = 0.2727. RStest: Conc1: $h = 0$, $p = 0.4060$. KStest2: Conc2: $h = 0$, $p = 0.9094$, ks2stat = 0.2197. RStest: Conc2: $h = 0$, $p = 0.7582$. KStest2: Conc2: $h = 1$, $p = 0.0289$, ks2stat = 0.5682. RStest: Conc2: $h = 1$, $p = 0.0310$. KStest2: Conc4: $h = 0$, $p = 0.1582$, ks2stat = 0.4500. RStest: Conc4: $h = 0$, $p = 0.3551$.

**Supplemental** Fig. 4
**Figure S.4c: Shape comparison of psychometric function**
statistical significance was determined by chi-squared test using SPSS software package ($F = 12$, $M = 11$). $p$-value for Sigmoidal and U-shape for initial 1–3 months: 0.016. $p$-value for Sigmoidal and U-shape after a year: 0.0009.

**Supplemental** Fig. 5
**Figure S.5a. Control vs. FD Distance traveled, approach only**
Effect of condition: d.f. = 1, $F = 10.2599$, $p = 0.0033$.
**Figure S.5b. Control vs. FD Number of stopping points, approach only**
Effect of condition: d.f. = 1, $F = 5.9745$, $p = 0.0208$.
**Figure S.5c. Control vs. FD Number of high-speed runs, approach only**
Effect of condition: d.f. = 1, $F = 0.6510$, $p = 0.4263$.
**Figure S.5d. Control vs. FD Proportion of trials outside all reward zone, approach only**
Effect of condition: d.f. = 1, $F = 3.2717$, $p = 0.0809$.
**Figure S.5e. Control vs. FD Distance traveled, reject only**
Effect of condition: d.f. = 1, $F = 17.8994$, $p = 0.0002$.
**Figure S.5f. Control vs. FD Number of stopping points, reject only**
Effect of condition: d.f. = 1, $F = 2.2523$, $p = 0.1424$.
**Figure S.5g. Control vs. FD Number of high-speed runs, reject only**
Effect of condition: d.f. = 1, $F = 9.8747$, $p = 0.0034$.
**Figure S.5h. Control vs. FD Proportion of trials outside all reward zone, reject only**
Effect of condition: d.f. = 1, $F = 7.0077$, $p = 0.0121$.
**Figure S.5i: FD approach rate (FvM)**
Statistical significance was determined by Repeated measures analysis of variance. (Female $N = 12$, Male $N = 10$). Effect of concentration: d.f. = 3, $F = 182.0357$, $p = 4.3911e-30$. Effect of sex: d.f. = 1, $F = 2.1437$, $p = 0.1587$. kstest2 results: $h = 0$, $p = 9.3097e-01$, ks2stat = 0.1125 (overall sex difference). Post-hoc analysis: 0.5%: 7.8880e−01, 2%: 2.2787e−01, 5%: 2.6929e−01, 9%: 7.6084e−01.

KStest2 and Wilcoxon rank sum test Results (complementary to post-hoc analysis): KStest2: Conc1: $h = 0$, $p = 0.9636$, *ks2stat* = 0.2000. RStest: Conc1: h = 0, p = 0.5631. KStest2: Conc2: $h = 0$, $p = 0.8286$, ks2stat = 0.2500. RStest: Conc2: $h = 0$, p = 0.2840. KStest2: Conc2: h = 0, p = 0.6961, ks2stat = 0.2833. RStest: Conc2: $h = 0$, p = 0.2892. KStest2: Conc4: $h = 0$, p = 0.8848, ks2stat = 0.2333. RStest: Conc4: h = 0, $p = 0.8391$.

**Control vs. FD Female:** Statistical significance was determined by Repeated measures analysis of variance. (Control $N = 12$, FD $N = 12$). $p$-value for Control vs. FD of female: 0.0042028. kstest2 results: $h = 1$, $p = 2.6487e-02$, ks2stat = 0.2917. Post-hoc analysis: 0.5%: 6.2877e−01, 2%: 4.9942e−01, 5%: 9.7466e−03, 9%: 4.6536e−02.

KStest2 and Wilcoxon rank sum test Results (complementary to post-hoc analysis): KStest2: Conc1: $h = 0$, $p = 0.4333$, ks2stat = 0.3333. RStest:

Conc1: $h = 0$, $p = 0.4504$. KStest2: Conc2: $h = 0$, $p = 0.1862$, ks2stat = 0.4167. RStest: Conc2: $h = 0$, $p = 0.3354$. KStest2: Conc2: $h = 0$, $p = 0.0656$, ks2stat = 0.5000. RStest: Conc2: $h = 1$, $p = 0.0140$. KStest2: Conc4: $h = 1$, $p = 0.0002$, ks2stat = 0.8333. RStest: Conc4: $h = 1$, $p = 0.0028$.

**Male:** Statistical significance was determined by Repeated measures analysis of variance. (Control $N = 11$, FD $N = 10$). $p$-value for Control vs. FD of male: 0.0027308. kstest2 results: $h = 0$, $p = 2.9608e{-}01$, ks2stat = 0.2068. Post-hoc analysis: 0.5%: 7.4398e$-$01, 2%: 9.6105e$-$01, 5%: 1.0232e$-$02, 9%: 4.9382e$-$02.

KStest2 and Wilcoxon rank sum test Results (complementary to post-hoc analysis): KStest2: Conc1: $h = 0$, $p = 0.2890$, ks2stat = 0.4000. RStest: Conc1: $h = 0$, $p = 0.3734$. KStest2: Conc2: $h = 0$, $p = 0.9884$, ks2stat = 0.1818. RStest: Conc2: $h = 0$, $p = 0.9716$. KStest2: Conc2: $h = 0$, $p = 0.0551$, ks2stat = 0.5455. RStest: Conc2: $h = 1$, $p = 0.0166$. KStest2: Conc4: $h = 0$, $p = 0.0697$, ks2stat = 0.5273. RStest: Conc4: $h = 1$, $p = 0.0301$.

**3-way ANOVA results**

| Source | Sum sq. | d.f. | Singular? | Mean Sq. | F | Prob > F |
|---|---|---|---|---|---|---|
| Sex | 0.0895 | 1 | 0 | 0.0895 | 4.7591 | 0.0305 |
| Condition | 0.3417 | 1 | 0 | 0.3417 | 18.158 | 0 |
| Concentration | 18.0205 | 3 | 0 | 6.0068 | 319.2341 | 0 |
| Sex*Condition | 0.0076 | 1 | 0 | 0.0076 | 0.4055 | 0.5251 |
| Sex*Concentration | 0.1528 | 3 | 0 | 0.0509 | 2.707 | 0.047 |
| Condition*Concentration | 0.549 | 3 | 0 | 0.183 | 9.7255 | 0 |
| Error | 3.1423 | 167 | 0 | 0.0188 | NaN | NaN |
| Total | 22.2751 | 179 | 0 | NaN | NaN | NaN |

**Figure S.5j: FD Distance traveled (FvM)**
Statistical significance was determined by Repeated measures analysis of variance. (Female $N = 12$, Male $N = 10$).Effect of concentration: d.f. = 3, $F = 12.1087$, $p = 2.6791e{-}06$. Effect of sex: d.f. = 1, $F = 21.4749$, $p = 0.0002$. kstest2 results: $h = 1$, $p = 1.3139e{-}08$, ks2stat = 0.6375 (overall sex difference). Post-hoc analysis: 0.5%: 4.0070e$-$04, 2%: 1.4910e$-$05, 5%: 9.2678e$-$03, 9%: 2.5464e$-$03.

KStest2 and Wilcoxon rank sum test Results (complementary to post-hoc analysis): KStest2: Conc1: $h = 1$, $p = 0.0076$, ks2stat = 0.6667. RStest: Conc1: $h = 1$, $p = 0.0033$. KStest2: Conc2: $h = 1$, $p = 0.0001$, ks2stat = 0.9167. RStest: Conc2: $h = 1$, $p = 0.0003$. KStest2: Conc2: $h = 0$, $p = 0.0567$, ks2stat = 0.5333. RStest: Conc2: $h = 1$, $p = 0.0229$. KStest2: Conc4: $h = 1$, $p = 0.0076$, ks2stat = 0.6667. RStest: Conc4: $h = 1$, $p = 0.0051$.

**Control vs. FD Female:** Statistical significance was determined by Repeated measures analysis of variance. (Control $N = 12$, FD $N = 12$). $p$-value for Control vs. FD of female: 0.0011858. kstest2 results: $h = 1$, $p = 1.7074e{-}05$, ks2stat = 0.4792. Post-hoc analysis: 0.5%: 5.1267e$-$04, 2%: 4.0099e$-$05, 5%: 1.0567e$-$01, 9%: 1.7728e$-$02.

KStest2 and Wilcoxon rank sum test Results (complementary to post-hoc analysis): KStest2: Conc1: $h = 1$, $p = 0.0046$, ks2stat = 0.6667. RStest: Conc1: $h = 1$, $p = 0.0017$. KStest2: Conc2: $h = 1$, $p = 0.0009$, ks2stat = 0.7500. RStest: Conc2: $h = 1$, $p = 0.0005$. KStest2: Conc2: $h = 0$, $p = 0.0656$, ks2stat = 0.5000. RStest: Conc2: $h = 0$, $p = 0.0531$. KStest2: Conc4: $h = 0$, $p = 0.0656$, ks2stat = 0.5000. RStest: Conc4: $h = 1$, $p = 0.0226$.

**Male:** Statistical significance was determined by Repeated measures analysis of variance. (Control $N = 11$, FD $N = 10$). $p$-value for Control vs. FD of male: 0.016738. kstest2 results: $h = 1$, $p = 2.1901e{-}04$, ks2stat = 0.4523. Post-hoc analysis: 0.5%: 4.2001e$-$03, 2%: 8.3414e$-$02, 5%: 2.0879e$-$02, 9%: 8.0063e$-$02.

KStest2 and Wilcoxon rank sum test Results (complementary to post-hoc analysis): KStest2: Conc1: h = 1, $p = 0.0198$, ks2stat = 0.6182. RStest: Conc1: $h = 1$, p = 0.0067. KStest2: Conc2: $h = 0$, $p = 0.2006$, ks2stat = 0.4364. RStest: Conc2: $h = 0$, $p = 0.1300$. KStest2: Conc2: $h = 0$, $p = 0.0978$, ks2stat =

0.5000. RStest: Conc2: $h = 1$, $p = 0.0317$. KStest2: Conc4: $h = 0$, $p = 0.2205$, ks2stat = 0.4273. RStest: Conc4: $h = 0$, $p = 0.0845$.

**3-way ANOVA results**

| Source | Sum sq. | d.f. | Singular? | Mean sq. | F | Prob > F |
|---|---|---|---|---|---|---|
| Sex | 27.3951 | 1 | 0 | 27.3951 | 106.6057 | 0 |
| Condition | 16.098 | 1 | 0 | 16.098 | 62.6439 | 0 |
| Concentration | 2.7766 | 3 | 0 | 0.9255 | 3.6016 | 0.0148 |
| Sex*Condition | 1.6504 | 1 | 0 | 1.6504 | 6.4225 | 0.0122 |
| Sex*Concentration | 0.5568 | 3 | 0 | 0.1856 | 0.7222 | 0.5401 |
| Condition*Concentration | 1.4535 | 3 | 0 | 0.4845 | 1.8854 | 0.134 |
| Error | 42.9151 | 167 | 0 | 0.257 | NaN | NaN |
| Total | 94.4813 | 179 | 0 | NaN | NaN | NaN |

**Figure S.5k: FD Number of stopping points (FvM)**
Statistical significance was determined by Repeated measures analysis of variance. (Female $N = 12$, Male $N = 10$). Effect of concentration: d.f. = 3, $F = 3.9968$, $p = 1.1616e{-}02$. Effect of sex: d.f. = 1, $F = 8.8810$, $p = 0.0074$. kstest2 results: $h = 1$, $p = 1.8518e{-}05$, ks2stat = 0.5000 (overall sex difference).

Post-hoc analysis: 0.5%: 1.8703e$-$02, 2%: 4.5455e$-$04, 5%: 2.3150e$-$01, 9%: 1.1937e$-$02.

KStest2 and Wilcoxon rank sum test Results (complementary to post-hoc analysis): KStest2: Conc1: $h = 1$, $p = 0.0076$, ks2stat = 0.6667 RStest: Conc1: $h = 1$, $p = 0.0111$. KStest2: Conc2: $h = 1$, $p = 0.0003$, ks2stat = 0.8333. RStest: Conc2: $h = 1$, $p = 0.0014$. KStest2: Conc2: $h = 0$, $p = 0.2270$, ks2stat = 0.4167. RStest: Conc2: $h = 0$, $p = 0.4098$. KStest2: Conc4: $h = 1$, $p = 0.0452$, ks2stat = 0.5500. RStest: Conc4: $h = 1$, $p = 0.0092$.

**Control vs. FD Female:** Statistical significance was determined by Repeated measures analysis of variance. (Control $N = 12$, FD $N = 12$). $p$-value for Control vs. FD of female: 0.010394. kstest2 results: $h = 1$, $p = 4.6080e{-}05$, ks2stat = 0.4583.

Post-hoc analysis: 0.5%: 3.0300e$-$03, 2%: 7.5065e$-$03, 5%: 1.1942e$-$01, 9%: 1.3927e$-$02.

KStest2 and Wilcoxon rank sum test Results (complementary to post-hoc analysis): KStest2: Conc1: $h = 1$, $p = 0.0191$, ks2stat = 0.5833. RStest: Conc1: $h = 1$, $p = 0.0043$. KStest2: Conc2: $h = 1$, $p = 0.0191$, ks2stat = 0.5833. RStest: Conc2: $h = 1$, $p = 0.0024$. KStest2: Conc2: $h = 0$, $p = 0.1862$, ks2stat = 0.4167. RStest: Conc2: $h = 0$, $p = 0.2366$. KStest2: Conc4: $h = 0$, $p = 0.0656$, ks2stat = 0.5000. RStest: Conc4: $h = 1$, $p = 0.0194$.

**Male:** Statistical significance was determined by Repeated measures analysis of variance. (Control $N = 11$, FD $N = 10$). $p$-value for Control vs FD of male: 0.1438. kstest2 results: $h = 1$, $p = 4.8245e{-}06$, ks2stat = 0.5386.

Post-hoc analysis: 0.5%: 9.9947e$-$02, 2%: 1.9939e$-$01, 5%: 1.1572e$-$01, 9%: 1.9212e$-$01.

KStest2 and Wilcoxon rank sum test Results (complementary to post-hoc analysis): KStest2: Conc1: $h = 1$, $p = 0.0009$, ks2stat = 0.8000. RStest: Conc1: $h = 1$, p = 0.0022. KStest2: Conc2: $h = 0$, $p = 0.0697$, ks2stat = 0.5273. RStest: Conc2: $h = 0$, $p = 0.0725$. KStest2: Conc2: $h = 1$, $p = 0.0173$, ks2stat = 0.6273. RStest: Conc2: $h = 1$, $p = 0.0028$. KStest2: Conc4: $h = 1$, $p = 0.0198$, ks2stat = 0.6182. RStest: Conc4: $h = 1$, $p = 0.0448$.

**3-way ANOVA results**

| Source | Sum sq. | d.f. | Singular? | Mean sq. | F | Prob > F |
|---|---|---|---|---|---|---|
| Sex | 290,797.7063 | 1 | 0 | 290,797.7063 | 14.1124 | 0.0002 |
| Condition | 403,580.2146 | 1 | 0 | 403,580.2146 | 19.5858 | 0 |
| Concentration | 2269.2494 | 3 | 0 | 756.4165 | 0.0367 | 0.9906 |

| | | | | | | |
|---|---|---|---|---|---|---|
| Sex*Condition | 73,903.6627 | 1 | 0 | 73,903.6627 | 3.5865 | 0.06 |
| Sex*Concentration | 5914.6548 | 3 | 0 | 1971.5516 | 0.0957 | 0.9623 |
| Condition*Concentration | 2016.996 | 3 | 0 | 672.332 | 0.0326 | 0.9921 |
| Error | 3,441,168.5604 | 167 | 0 | 20,605.7998 | NaN | NaN |
| Total | 4,223,019.6835 | 179 | 0 | | NaN | NaN | NaN |

**Figure S.5l: FD Number of high sp. runs (FvM)**

Statistical significance was determined by Repeated measures analysis of variance. (Female $N$ = 12, Male $N$ = 10). Effect of concentration: d.f. = 3, $F$ = 33.4826, $p$ = 7.6444e−13. Effect of sex: d.f. = 1, $F$ = 5.9221, $p$ = 0.0245. kstest2 results: $h$ =1, $p$ = 6.8336e−03, ks2stat = 0.3500 (overall sex difference)

Post-hoc analysis: 0.5%: 1.1326e−02, 2%: 1.2677e−04, 5%: 9.3261e−01, 9%: 1.6538e−01.

KStest2 and Wilcoxon rank sum test Results (complementary to post-hoc analysis): KStest2: Conc1: $h$ = 1, $p$ = 0.0358, ks2stat = 0.5667. RStest: Conc1: $h$ = 1, $p$ = 0.0111. KStest2: Conc2: $h$ = 1, $p$ = 0.0001, ks2stat = 0.9167. RStest: Conc2: $h$ = 1, $p$ = 0.0003. KStest2: Conc2: $h$ = 0, $p$ = 0.9989, ks2stat = 0.1500. RStest: Conc2: $h$ = 0, $p$ = 0.9212. KStest2: Conc4: $h$ = 0, $p$ = 0.5564, ks2stat = 0.3167. RStest: Conc4: $h$ = 0, $p$ = 0.2485

**Control vs. FD Female:** Statistical significance was determined by Repeated measures analysis of variance. (Control N = 12, FD $N$ = 12). $p$-value for Control vs. FD of female: 0.13035. kstest2 results: $h$ = 0, $p$ = 8.3415e−02, ks2stat = 0.2500.

Post-hoc analysis: 0.5%: 1.5506e − 02, 2%: 1.7542e−02, 5%: 5.6165e−01, 9%: 1.9334e−01.

KStest2 and Wilcoxon rank sum test Results (complementary to post-hoc analysis): KStest2: Conc1: $h$ = 1, $p$ = 0.0191, ks2stat = 0.5833. RStest: Conc1: $h$ = 1, $p$ = 0.0194. KStest2: Conc2: $h$ = 1, p = 0.0046, ks2stat = 0.6667. RStest: Conc2: $h$ = 1, $p$ = 0.0102. KStest2: Conc2: $h$ = 0, $p$ = 0.7864, ks2stat = 0.2500. RStest: Conc2: $h$ = 0, $p$ = 0.5834. KStest2: Conc4: $h$ = 0, $p$ = 0.4333, ks2stat = 0.3333. RStest: Conc4: $h$ = 0, $p$ = 0.2145.

**Male:** Statistical significance was determined by Repeated measures analysis of variance. (Control $N$ = 11, FD $N$ = 10). $p$-value for Control vs. FD of male: 0.025096. kstest2 results: $h$ = 1, $p$ = 3.0081e−03, ks2stat = 0.3818.

Post-hoc analysis: 0.5%: 4.9411e−04, 2%: 4.0639e−01, 5%: 1.5711e−02, 9%: 2.2641e−01.

KStest2 and Wilcoxon rank sum test Results (complementary to post-hoc analysis): KStest2: Conc1: $h$ = 1, $p$ = 0.0046, ks2stat = 0.7091. RStest: Conc1: $h$ = 1, $p$ = 0.0028. KStest2: Conc2: $h$ = 0, $p$ = 0.5376, ks2stat = 0.3273. RStest: Conc2: $h$ = 0, $p$ = 0.3418. KStest2: Conc2: $h$ = 0, $p$ = 0.0551, $ks2stat$ = 0.5455. RStest: Conc2: $h$ = 1, $p$ = 0.0448. KStest2: Conc4: $h$ = 0, $p$ = 0.4339, ks2stat = 0.3545. RStest: Conc4: $h$ = 0, $p$ = 0.2178.

**3-way ANOVA results**

| Source | Sum Sq. | d.f. | Singular? | Mean Sq. | F | Prob > F |
|---|---|---|---|---|---|---|
| Sex | 448.5973 | 1 | 0 | 448.5973 | 52.871 | 0 |
| Condition | 217.1167 | 1 | 0 | 217.1167 | 25.589 | 0 |
| Concentration | 476.1899 | 3 | 0 | 158.73 | 18.7077 | 0 |
| Sex*Condition | 47.5647 | 1 | 0 | 47.5647 | 5.6059 | 0.019 |
| Sex*Concentration | 29.9264 | 3 | 0 | 9.9755 | 1.1757 | 0.3207 |
| Condition*Concentration | 41.8773 | 3 | 0 | 13.9591 | 1.6452 | 0.1809 |
| Error | 1416.9547 | 167 | 0 | 8.4848 | NaN | NaN |
| Total | 2703.9688 | 179 | 0 | | NaN | NaN | NaN |

**Figure E.5m: FD Prop. of trial out. all reward zones (FvM)**

Statistical significance was determined by Repeated measures analysis of variance. (Female $N$ = 12, Male $N$ = 10). Effect of concentration: d.f. = 3, $F$ = 45.0059, $p$ = 2.3011e−15. Effect of sex: d.f. = 1, $F$ = 0.7749, $p$ = 0.3892. kstest2 results: $h$ = 0, $p$ = 2.6677e−01, ks2stat = 0.2083 (overall sex difference)

Post-hoc analysis: 0.5%: 2.8309e−01, 2%: 1.3761e−02, 5%: 8.4288e−01, 9%: 5.6653e−01.

KStest2 and Wilcoxon rank sum test Results (complementary to post-hoc analysis): KStest2: Conc1: $h$ = 0, $p$ = 0.4896, ks2stat = 0.3333. RStest: Conc1: $h$ = 0, $p$ = 0.3891. KStest2: Conc2: $h$ = 0, $p$ = 0.0567, ks2stat = 0.5333. RStest: Conc2: $h$ = 1, $p$ = 0.0149. KStest2: Conc2: $h$ = 0, $p$ = 0.9636, ks2stat = 0.2000. RStest: Conc2: $h$ = 0, $p$ = 0.8940. KStest2: Conc4: $h$ = 0, $p$ = 0.3689, ks2stat = 0.3667. RStest: Conc4: $h$ = 0, $p$ = 0.5716.

**Control vs. FD Female:** Statistical significance was determined by Repeated measures analysis of variance. (Control $N$ = 12, FD $N$ = 12). $p$-value for Control vs FD of female: 0.23688. kstest2 results: $h$ = 1, $p$ = 4.8054e−02, ks2stat = 0.2708. Post-hoc analysis: 0.5%: 2.8018e−02, 2%: 8.2300e−03, 5%: 3.6018e−01, 9%: 7.4730e−01.

KStest2 and Wilcoxon rank sum test Results (complementary to post-hoc analysis): KStest2: Conc1: $h$ = 0, $p$ = 0.1862, ks2stat = 0.4167. RStest: Conc1: $h$ = 1, $p$ = 0.0399. KStest2: Conc2: $h$ = 0, $p$ = 0.0656, ks2stat = 0.5000. RStest: Conc2: $h$ = 1, $p$ = 0.0163. KStest2: Conc2: $h$ = 0, $p$ = 0.1862, ks2stat = 0.4167. RStest: Conc2: $h$ = 0, $p$ = 0.1651. KStest2: Conc4: $h$ = 0, $p$ = 0.7864, ks2stat = 0.2500. RStest: Conc4: $h$ = 0, $p$ = 0.4510.

**Male:** Statistical significance was determined by Repeated measures analysis of variance. (Control $N$ = 11, FD $N$ = 10). $p$-value for Control vs. FD of male: 0.19499. kstest2 results: $h$ = 0, $p$ = 7.2583e−02, ks2stat = 0.2727.

Post-hoc analysis: 0.5%: 5.5760e−02, 2%: 4.2774e−01, 5%: 6.6086e−01, 9%: 2.2116e−01

KStest2 and Wilcoxon rank sum test Results (complementary to post-hoc analysis): KStest2: Conc1: $h$ = 1, $p$ = 0.0227, ks2stat = 0.6091. RStest: Conc1: $h$ = 1, $p$ = 0.0342. KStest2: Conc2: $h$ = 0, $p$ = 0.4339, ks2stat = 0.3545. RStest: Conc2: $h$ = 0, $p$ = 0.2872. KStest2: Conc2: $h$ = 0, $p$ = 0.8603, ks2stat = 0.2455. RStest: Conc2: $h$ = 0, $p$ = 0.6961. KStest2: Conc4: $h$ = 0, $p$ = 0.0876, ks2stat = 0.5091. RStest: Conc4: $h$ = 0, $p$ = 0.1675.

**3-way ANOVA results**

| Source | Sum Sq. | d.f. | Singular? | Mean Sq. | F | Prob > F |
|---|---|---|---|---|---|---|
| Sex | 0.1799 | 1 | 0 | 0.1799 | 6.1638 | 0.014 |
| Condition | 0.2727 | 1 | 0 | 0.2727 | 9.3414 | 0.0026 |
| Concentration | 1.7647 | 3 | 0 | 0.5882 | 20.1509 | 0 |
| Sex*Condition | 0.0005 | 1 | 0 | 0.0005 | 0.0178 | 0.8939 |
| Sex*Concentration | 0.0584 | 3 | 0 | 0.0195 | 0.6672 | 0.5733 |
| Condition*Concentration | 0.2834 | 3 | 0 | 0.0945 | 3.2362 | 0.0237 |
| Error | 4.8749 | 167 | 0 | 0.0292 | NaN | NaN |
| Total | 7.4668 | 179 | 0 | | NaN | NaN | NaN |

**Figure S.5q: Baseline and food deprivation distance traveled individual rat Euclidian distance**

Statistical significance $p$ = 1.3043e−22, determined by two-sample Kolmogorov–Smirnov test. (Control $N$ = 23, FD $N$ = 22).

**Figure S.5r: Baseline and food deprivation stopping points individual rat Euclidian distance**

Statistical significance $p$ = 2.2672e−47, determined by two-sample Kolmogorov-Smirnov test. (Control $N$ = 23, FD $N$ = 22).

**Supplemental** Fig. 6

**Figure S.6a: Self admin oxycodone Approach time (FvM)**

Statistical significance was determined by Repeated measures analysis of variance. (Female $N$ = 5, Male $N$ = 5). Effect of concentration: d.f. = 3, $F$ = 2.4534, $p$ = 8.7784e−02. Effect of sex: d.f. = 1, $F$ = 6.7698, $p$ = 0.0315. kstest2 results: $h$ = 1, $p$ = 2.3213e−02, ks2stat = 0.4500 (overall sex difference).

Post-hoc analysis: 0.5%: 3.1246e−03, 2%: 5.8901e−01, 5%: 3.6169e−02, 9%: 9.2633e−01. KStest2 and Wilcoxon rank sum test Results (complementary to post-hoc analysis). KStest2: Conc1: $h$ = 1, $p$ = 0.0038, ks2stat = 1.0000. RStest: Conc1: $h$ = 1, $p$ = 0.0079. KStest2: Conc2: $h$ = 0, $p$ = 0.9996,

ks2stat = 0.2000. RStest: Conc2: $h = 0$, $p = 0.6905$. KStest2: Conc2: $h = 1$, $p = 0.0361$, ks2stat = 0.8000. RStest: Conc2: $h = 0$, $p = 0.0556$. KStest2: Conc4: $h = 0$, $p = 0.6974$, ks2stat = 0.4000. RStest: Conc4: $h = 0$, $p = 1.0000$.

**Control vs. Self admin. Oxy Female:** Statistical significance was determined by Repeated measures analysis of variance. (Control $N = 12$, Self admin. Oxy $N = 5$). $p$-value for Control vs. Self admin. Oxy of female: 0.95625. kstest2 results: $h = 0$, $p = 8.3454e{-}01$, ks2stat = 0.1596.

Post-hoc analysis: 0.5%: $2.0591e{-}01$, 2%: $9.2070e{-}01$, 5%: $6.9608e{-}01$, 9%: $1.7620e{-}01$. KStest2 and Wilcoxon rank sum test Results (complementary to post-hoc analysis). KStest2: Conc1: $h = 0$, $p = 0.1707$, ks2stat = 0.5455. RStest: Conc1: $h = 0$, $p = 0.2674$. KStest2: Conc2: $h = 0$, $p = 0.9887$, ks2stat = 0.2167. RStest: Conc2: $h = 0$, $p = 0.7990$. KStest2: Conc2: $h = 0$, $p = 0.4046$, ks2stat = 0.4333. RStest: Conc2: $h = 0$, $p = 0.6461$. KStest2: Conc4: $h = 0$, $p = 0.5074$, ks2stat = 0.4000. RStest: Conc4: $h = 0$, $p = 0.3284$.

**Male:** Statistical significance was determined by Repeated measures analysis of variance. (Control $N = 11$, Self admin. Oxy $N = 5$). p-value for Control vs Self admin. Oxy of male: 0.11333. kstest2 results: $h = 0$, $p = 6.9831e{-}02$, ks2stat=0.3381

Post-hoc analysis: 0.5%: $2.1225e{-}01$, 2%: $2.2855e{-}01$, 5%: $7.6955e{-}02$, 9%: $5.4354e{-}01$. KStest2 and Wilcoxon rank sum test Results (complementary to post-hoc analysis). KStest2: Conc1: $h = 0$, $p = 0.6450$, ks2stat = 0.3636. RStest: Conc1: $h = 0$, $p = 0.3773$. KStest2: Conc2: $h = 0$, $p = 0.5402$, ks2stat = 0.4000. RStest: Conc2: $h = 0$, $p = 0.4396$. KStest2: Conc2: $h = 1$, $p = 0.0388$, ks2stat = 0.7000. RStest: Conc2: $h = 0$, $p = 0.0553$. KStest2: Conc4: $h = 0$, $p = 0.4648$, ks2stat = 0.4182. RStest: Conc4: $h = 0$, $p = 0.4409$.

**3-way ANOVA results**

| Source | Sum Sq. | d.f. | Singular? | Mean sq. | F | Prob > F |
|---|---|---|---|---|---|---|
| Sex | 23.5048 | 1 | 0 | 23.5048 | 4.8286 | 0.03 |
| Condition | 8.2571 | 1 | 0 | 8.2571 | 1.6963 | 0.1954 |
| Concentration | 96.2147 | 3 | 0 | 32.0716 | 6.5884 | 0.0004 |
| Sex*Condition | 5.0089 | 1 | 0 | 5.0089 | 1.029 | 0.3125 |
| Sex*Concentration | 10.1859 | 3 | 0 | 3.3953 | 0.6975 | 0.5554 |
| Condition*Concentration | 21.9818 | 3 | 0 | 7.3273 | 1.5052 | 0.2169 |
| Error | 564.6711 | 116 | 0 | 4.8679 | NaN | NaN |
| Total | 758.4433 | 128 | 0 | NaN | NaN | NaN |

**Figure S.6b: Self-admin oxycodone Prop. of trial out. all reward zones (FvM)**

Statistical significance was determined by Repeated measures analysis of variance. (Female $N = 5$, Male $N = 5$). Effect of concentration: d.f. = 3, $F = 0.5016$, $p = 6.8473e{-}01$. Effect of sex: d.f. = 1, $F = 2.4942$, $p = 0.1529$. kstest2 results: $h = 1$, $p = 2.3213e{-}02$, ks2stat = 0.4500 (overall sex difference).

Post-hoc analysis: 0.5%: $3.2600e{-}01$, 2%: $1.2515e{-}01$, 5%: $3.5598e{-}01$, 9%: $1.0995e{-}01$. KStest2 and Wilcoxon rank sum test Results (complementary to post-hoc analysis). KStest2: Conc1: $h = 0$, $p = 0.2090$, ks2stat = 0.6000. RStest: Conc1: $h = 0$, $p = 0.3095$. KStest2: Conc2: $h = 0$, $p = 0.2090$, ks2stat = 0.6000. RStest: Conc2: h = 0, $p = 0.2222$. KStest2: Conc2: $h = 0$, $p = 0.2090$, ks2stat = 0.6000. RStest: Conc2: $h = 0$, $p = 0.3968$. KStest2: Conc4: $h = 0$, $p = 0.2090$, ks2stat = 0.6000. RStest: Conc4: $h = 0$, $p = 0.2222$.

**Control vs. Self admin. Oxy Female:** Statistical significance was determined by Repeated measures analysis of variance. (Control $N = 12$, Self admin. Oxy $N = 5$). $p$-value for Control vs. Self admin. Oxy of female: 0.096217. kstest2 results: $h = 1$, $p = 4.6360e{-}03$, ks2stat = 0.4458

Post-hoc analysis: 0.5%: $2.1543e{-}01$, 2%: $3.5559e{-}02$, 5%: $2.7408e{-}01$, 9%: $2.8956e{-}01$. KStest2 and Wilcoxon rank sum test Results (complementary to post-hoc analysis). KStest2: Conc1: $h = 0$, $p = 0.3153$, ks2stat = 0.4667. RStest: Conc1: $h = 0$, $p = 0.1542$. KStest2: Conc2: $h = 1$, $p = 0.0259$, ks2stat = 0.7167. RStest: Conc2: $h = 1$, $p = 0.0343$. KStest2: Conc2: $h = 0$, $p = 0.7348$, ks2stat = 0.3333. RStest: Conc2: $h = 0$, $p = 0.6299$. KStest2: Conc4: $h = 0$, $p = 0.0671$, ks2stat = 0.6333. RStest: Conc4: $h = 0$, $p = 0.1503$.

**Male:** Statistical significance was determined by Repeated measures analysis of variance. (Control $N = 11$, Self admin. Oxy $N = 5$). p-value for Control vs Self admin. Oxy of male: 0.2521. kstest2 results: $h = 0$, $p = 1.3724e{-}01$, ks2stat = 0.3000.

Post-hoc analysis: 0.5%: $3.6395e{-}01$, 2%: $5.1022e{-}01$, 5%: $1.7972e{-}01$, 9%: $3.6020e{-}02$. KStest2 and Wilcoxon rank sum test Results (complementary to post-hoc analysis). KStest2: Conc1: $h = 0$, $p = 0.7677$, ks2stat = 0.3273. RStest: Conc1: $h = 0$, $p = 0.7628$. KStest2: Conc2: h = 0, $p = 0.7072$, ks2stat = 0.3455. RStest: Conc2: $h = 0$, $p = 0.4592$. KStest2: Conc2: $h = 0$, $p = 0.4648$, ks2stat = 0.4182. RStest: Conc2: $h = 0$, $p = 0.2088$. KStest2: Conc4: $h = 1$, $p = 0.0101$, ks2stat = 0.8000. RStest: Conc4: $h = 0$, $p = 0.0641$.

**3-way ANOVA results**

| Source | Sum Sq. | d.f. | Singular? | Mean Sq. | F | Prob > F |
|---|---|---|---|---|---|---|
| Sex | 0.0435 | 1 | 0 | 0.0435 | 1.7476 | 0.1887 |
| Condition | 0.0069 | 1 | 0 | 0.0069 | 0.2763 | 0.6001 |
| Concentration | 0.1103 | 3 | 0 | 0.0368 | 1.4758 | 0.2246 |
| Sex*Condition | 0.3152 | 1 | 0 | 0.3152 | 12.6488 | 0.0005 |
| Sex*Concentration | 0.0352 | 3 | 0 | 0.0117 | 0.4706 | 0.7033 |
| Condition*Concentration | 0.1621 | 3 | 0 | 0.054 | 2.1684 | 0.0953 |
| Error | 2.9652 | 119 | 0 | 0.0249 | NaN | NaN |
| Total | 3.7529 | 131 | 0 | NaN | NaN | NaN |

**Figure S.6c: Self admin oxycodone Number of stopping points (FvM)**

Statistical significance was determined by Repeated measures analysis of variance. (Female $N = 5$, Male $N = 5$). Effect of concentration: d.f. = 3, $F = 0.3109$, $p = 8.1732e{-}01$. Effect of sex: d.f. = 1, $F = 2.9124$, $p = 0.1263$. kstest2 results: $h = 1$, $p = 8.1617e{-}03$, ks2stat = 0.5000 (overall sex difference). Post-hoc analysis: 0.5%: $2.1755e{-}01$, 2%: $9.1185e{-}02$, 5%: $1.8516e{-}01$, 9%: $9.4228e{-}02$.

KStest2 and Wilcoxon rank sum test Results (complementary to post-hoc analysis). KStest2: Conc1: $h = 0$, $p = 0.6974$, ks2stat = 0.4000. RStest: Conc1: $h = 0$, $p = 0.2222$. KStest2: Conc2: $h = 1$, $p = 0.0361$, ks2stat = 0.8000. RStest: Conc2: $h = 1$, p = 0.0317. KStest2: Conc2: $h = 0$, $p = 0.2090$, ks2stat = 0.6000. RStest: Conc2: $h = 0$, $p = 0.1508$. KStest2: Conc4: $h = 0$, $p = 0.2090$, ks2stat = 0.6000. RStest: Conc4: $h = 0$, $p = 0.0952$.

**Control vs. Self admin. Oxy Female:**

Statistical significance was determined by Repeated measures analysis of variance. (Control $N = 12$, Self admin. Oxy $N = 5$). $p$-value for Control vs. initial task of female: 0.013864. kstest2 results: $h = 1$, $p = 1.6985e{-}04$, ks2stat = 0.5542.

Post-hoc analysis: 0.5%: $6.5797e{-}03$, 2%: $2.4568e{-}02$, 5%: $3.1498e{-}02$, 9%: $1.3492e{-}02$. KStest2 and Wilcoxon rank sum test Results (complementary to post-hoc analysis). KStest2: Conc1: $h = 0$, $p = 0.0950$, ks2stat = 0.6000. RStest: Conc1: $h = 1$, $p = 0.0485$. KStest2: Conc2: $h = 0$, $p = 0.0671$, ks2stat = 0.6333. RStest: Conc2: $h = 1$, $p = 0.0365$. KStest2: Conc2: $h = 0$, $p = 0.3153$, ks2stat = 0.4667. RStest: Conc2: $h = 0$, $p = 0.1037$. KStest2: Conc4: $h = 0$, $p = 0.1545$, ks2stat = 0.5500. RStest: Conc4: $h = 0$, $p = 0.1037$.

**Male:** Statistical significance was determined by Repeated measures analysis of variance. (Control $N = 11$, Self admin. Oxy $N = 5$). $p$-value for Control vs. initial task of male: 0.38653. kstest2 results: $h = 1$, $p = 4.2681e{-}03$, ks2stat = 0.4545.

Post-hoc analysis: 0.5%: $4.9541e{-}01$, 2%: $3.2649e{-}01$, 5%: $3.7954e{-}01$, 9%: $3.6035e{-}01$. KStest2 and Wilcoxon rank sum test Results (complementary to post-hoc analysis). KStest2: Conc1: $h = 0$, $p = 0.1019$, ks2stat = 0.6000. RStest: Conc1: $h = 0$, $p = 0.3196$. KStest2: Conc2: $h = 0$, $p = 0.2342$, ks2stat = 0.5091. RStest: Conc2: $h = 0$, $p = 0.1451$. KStest2: Conc2: $h = 0$, $p = 0.2005$, ks2stat = 0.5273. RStest: Conc2: $h = 0$, $p = 0.2212$. KStest2: Conc4: $h = 0$, $p = 0.1019$, ks2stat = 0.6000. RStest: Conc4: $h = 0$, $p = 0.1451$.

**3-way ANOVA results**

| Source | Sum Sq. | d.f. | Singular? | Mean Sq. | F | Prob>F |
|---|---|---|---|---|---|---|
| Sex | 63,068.0185 | 1 | 0 | 63,068.0185 | 1.5922 | 0.2095 |
| Condition | 79,667.9163 | 1 | 0 | 79,667.9163 | 2.0112 | 0.1588 |
| Concentration | 2413.7358 | 3 | 0 | 804.5786 | 0.0203 | 0.996 |
| Sex*Condition | 794,212.3436 | 1 | 0 | 794,212.3436 | 20.0501 | 0 |
| Sex*Concentration | 20,606.0544 | 3 | 0 | 6868.6848 | 0.1734 | 0.9142 |
| Condition*Concentration | 2792.4651 | 3 | 0 | 930.8217 | 0.0235 | 0.9951 |
| Error | 4,713,750.3113 | 119 | 0 | 39,611.3472 | NaN | NaN |
| Total | 5,631,827.6514 | 131 | 0 | NaN | NaN | NaN |

**Figure S.6d: Abstinence approach time (FvM)**

Statistical significance was determined by Repeated measures analysis of variance. (Female $N = 6$, Male $N = 6$).Effect of concentration: d.f. = 3, $F = 8.5277$, $p = 3.0232e$-04. Effect of sex: d.f. = 1, $F = 0.5135$, $p = 0.4900$. kstest2 results: $h = 0$, $p = 8.6076e$−01, ks2stat = 0.1667 (overall sex difference). Post-hoc analysis: 0.5%: 6.9253e−01, 2%: 6.3994e−01, 5%: 6.0961e−01, 9%: 7.4470e−01.

KStest2 and Wilcoxon rank sum test Results (complementary to post-hoc analysis). KStest2: Conc1: $h = 0$, $p = 0.8096$, ks2stat = 0.3333. RStest: Conc1: $h = 0$, $p = 0.5887$. KStest2: $Conc2$: $h = 0$, $p = 0.8096$, ks2stat = 0.3333. RStest: Conc2: $h = 0$, $p = 0.6991$. KStest2: Conc3: $h = 0$, $p = 0.3180$, ks2stat = 0.5000. RStest: Conc3: $h = 0$, $p = 0.4848$. KStest2: Conc4: $h = 0$, $p = 0.8096$, ks2stat = 0.3333. RStest: Conc4: $h = 0$, $p = 0.8182$.

**Control vs. Abstinence Female:** Statistical significance was determined by Repeated measures analysis of variance. (Control $N = 12$, Abstinence $N = 6$). $p$-value for Control vs initial task of female: 0.8104. kstest2 results: $h = 0$, $p = 5.1065e$−01, ks2stat = 0.1986. Post-hoc analysis: 0.5%: 9.1448e−01, 2%: 8.9431e−01, 5%: 7.9708e−01, 9%: 6.4577e−01.

KStest2 and Wilcoxon rank sum test Results (complementary to post-hoc analysis). KStest2: $Conc1$: $h = 0$, $p = 0.2971$, ks2stat = 0.4545. RStest: Conc1: $h = 0$, $p = 0.7325$. KStest2: Conc2: $h = 0$, $p = 0.9290$, ks2stat = 0.2500. RStest: Conc2: $h = 0$, $p = 0.7503$. KStest2: $Conc3$: $h = 0$, $p = 0.9290$, ks2stat = 0.2500. RStest: Conc3: $h = 0$, $p = 0.8916$. KStest2: Conc4: $h = 0$, $p = 0.9290$, ks2stat = 0.2500. RStest: Conc4: $h = 0$, $p = 0.6820$.

**Male:** Statistical significance was determined by Repeated measures analysis of variance. (Control $N = 11$, Abstinence $N = 6$). $p$-value for Control vs. initial task of male: 0.8535. kstest2 results: $h = 0$, $p = 7.5068e$−01, ks2stat = 0.1667. Post-hoc analysis: 0.5%: 8.1127e−01, 2%: 9.0195e−01, 5%: 9.8759e−01, 9%: 7.9737e−01.

KStest2 and Wilcoxon rank sum test Results (complementary to post-hoc analysis). KStest2: $Conc1$: $h = 0$, $p = 0.7395$, ks2stat = 0.3182. RStest: Conc1: $h = 0$, $p = 0.4623$. KStest2: Conc2: $h = 0$, $p = 0.8163$, ks2stat = 0.3000. RStest: Conc2: $h = 0$, $p = 0.7925$. KStest2: $Conc3$: $h = 0$, $p = 0.4725$, ks2stat = 0.4000. RStest: Conc3: $h = 0$, $p = 0.7925$. KStest2: $Conc4$: $h = 0$, $p = 0.2971$, ks2stat = 0.4545. RStest: Conc4: $h = 0$, $p = 0.6605$.

**3-way ANOVA results**

| Source | Sum sq. | d.f. | Singular? | Mean Sq. | F | Prob >F |
|---|---|---|---|---|---|---|
| Sex | 5.7742 | 1 | 0 | 5.7742 | 1.4499 | 0.2308 |
| Condition | 0.8863 | 1 | 0 | 0.8863 | 0.2225 | 0.6379 |
| Concentration | 140.1993 | 3 | 0 | 46.7331 | 11.7345 | 0 |
| Sex*Condition | 0.1205 | 1 | 0 | 0.1205 | 0.0303 | 0.8622 |
| Sex*Concentration | 2.97 | 3 | 0 | 0.99 | 0.2486 | 0.8622 |
| Condition*Concentration | 0.9126 | 3 | 0 | 0.3042 | 0.0764 | 0.9726 |
| Error | 493.8364 | 124 | 0 | 3.9826 | NaN | NaN |
| Total | 667.9834 | 136 | 0 | NaN | NaN | NaN |

**Figure S.6e: Abstinence Prop. of trial out. all reward zones (FvM)**

Statistical significance was determined by Repeated measures analysis of variance. (Female $N = 6$, Male $N = 6$).Effect of concentration: d.f. = 3, $F = 7.6904$, $p = 5.9005e$−04. Effect of sex: d.f. = 1, $F = 2.5630$, $p = 0.1405$. kstest2 results: $h = 0$, $p = 2.1598e$−01, ks2stat = 0.2917 (overall sex difference). Post-hoc analysis: 0.5%: 3.2825e−01, 2%: 4.1626e−01, 5%: 5.1375e−02, 9%: 3.6430e−01.

KStest2 and Wilcoxon rank sum test Results (complementary to post-hoc analysis). KStest2: Conc1: $h = 0$, $p = 0.3180$, ks2stat = 0.5000. RStest: Conc1: $h = 0$, $p = 0.3939$. KStest2: Conc2: $h = 0$, $p = 0.8096$, ks2stat = 0.3333. RStest: Conc2: $h = 0$, $p = 0.5887$. KStest2: Conc3: $h = 0$, $p = 0.0766$, $ks2stat = 0.6667$. RStest: Conc3: $h = 0$, $p = 0.0649$. KStest2: Conc4: $h = 0$, $p = 0.8096$, ks2stat = 0.3333. RStest: Conc4: $h = 0$, $p = 0.4848$.

**Control vs. Abstinence Female:**

Statistical significance was determined by Repeated measures analysis of variance. (Control $N = 12$, Abstinence $N = 6$). $p$-value for Control vs. initial task of female: 0.42819. kstest2 results: $h = 0$, $p = 6.9487e$−02, ks2stat = 0.3125. Post-hoc analysis: 0.5%: 3.6147e−01, 2%: 8.1315e−01, 5%: 5.1536e−01, 9%: 4.0850e−01.

KStest2 and Wilcoxon rank sum test Results (complementary to post-hoc analysis). KStest2: $Conc1$: $h = 0$, $p = 0.6693$, ks2stat = 0.3333. RStest: Conc1: $h = 0$, $p = 0.3731$. KStest2: $Conc2$: $h = 0$, $p = 0.6693$, ks2stat = 0.3333. RStest: Conc2: $h = 0$, $p = 0.9462$. KStest: Conc3: $h = 0$, $p = 0.1877$, ks2stat = 0.5000. RStest: Conc3: $h = 0$, $p = 0.2579$. KStest2: Conc4: $h = 0$, $p = 0.3842$, ks2stat = 0.4167. RStest: Conc4: $h = 0$, $p = 0.3704$.

**Male:** Statistical significance was determined by Repeated measures analysis of variance. (Control $N = 11$, Abstinence $N = 6$). $p$-value for Control vs. initial task of male: 0.50339. kstest2 results: $h = 0$, $p = 4.1694e$−01, ks2stat = 0.2159. Post-hoc analysis: 0.5%: 3.3679e−01, 2%: 8.9578e−01, 5%: 4.9842e−01, 9%: 3.3823e−01.

KStest2 and Wilcoxon rank sum test Results (complementary to post-hoc analysis). KStest2: Conc1: $h = 0$, $p = 0.4722$, ks2stat = 0.3939. RStest: Conc1: $h = 0$, $p = 0.2455$. KStest2: Conc2: $h = 0$, $p = 0.9857$, ks2stat = 0.2121. RStest: Conc2: $h = 0$, $p = 0.8641$. KStest2: Conc3: $h = 0$, $p = 0.1106$, ks2stat = 0.5606. RStest: Conc3: $h = 0$, $p = 0.1708$. KStest2: Conc4: $h = 0$, $p = 0.4722$, ks2stat = 0.3939. RStest: Conc4: $h = 0$, $p = 0.3108$.

**3-way ANOVA results**

| Source | Sum Sq. | d.f. | Singular? | Mean Sq. | F | Prob > F |
|---|---|---|---|---|---|---|
| Sex | 0.151 | 1 | 0 | 0.151 | 7.7287 | 0.0063 |
| Condition | 0.0602 | 1 | 0 | 0.0602 | 3.0791 | 0.0817 |
| Concentration | 0.4706 | 3 | 0 | 0.1569 | 8.0279 | 0.0001 |
| Sex*Condition | 0.0002 | 1 | 0 | 0.0002 | 0.0092 | 0.9237 |
| Sex*Concentration | 0.0293 | 3 | 0 | 0.0098 | 0.5001 | 0.6829 |
| Condition*Concentration | 0.0196 | 3 | 0 | 0.0065 | 0.3347 | 0.8003 |
| Error | 2.4814 | 127 | 0 | 0.0195 | NaN | NaN |
| Total | 3.3006 | 139 | 0 | NaN | NaN | NaN |

**Figure S.6f: Abstinence Number of stopping points (FvM)**

Statistical significance was determined by repeated measures analysis of variance. (Female $N = 6$, Male $N = 6$). Effect of concentration: d.f. = 3, $F = 0.9612$, $p = 4.2378e$−01. Effect of sex: d.f. = 1, $F = 1.1444$, $p = 0.3099$. kstest2 results: $h = 1$, $p = 9.3124e$−04, ks2stat = 0.5417 (overall sex difference). Post-hoc analysis: 0.5%: 3.2581e−01, 2%: 3.9929e−01, 5%: 2.7256e−01, 9%: 2.6367e−01.

KStest2 and Wilcoxon rank sum test results (complementary to post-hoc analysis): KStest2: Conc1: $h = 0$, $p = 0.0766$, ks2stat = 0.6667. RStest: Conc1: $h = 0$, $p = 0.1320$. KStest2: Conc2: $h = 1$, $p = 0.0122$, ks2stat

= 0.8333RStest: Conc2: $h = 0$, $p = 0.0649$. KStest2: *Conc3*: $h = 0$, $p = 0.3180$, ks2stat = 0.5000. RStest: Conc3: $h = 0$, $p = 0.3095$. KStest2: Conc4: $h = 0$, $p = 0.3180$, *ks2stat* = 0.5000. RStest: Conc4: $h = 0$, $p = 0.2403$.

**Control vs. Abstinence Female:** Statistical significance was determined by Repeated measures analysis of variance. (Control $N = 12$, Abstinence $N = 6$). *p*-value for Control vs. initial task of female: 0.23108. kstest2 results: $h = 0$, $p = 4.4421$e−01, ks2stat = 0.2083.

Post-hoc analysis: 0.5%: 2.0036e−01, 2%: 3.5578e−01, 5%: 2.1478e−01, 9%: 1.7829e−01. KStest2 and Wilcoxon rank sum test Results (complementary to post-hoc analysis). KStest2: *Conc1*: $h = 0$, $p = 0.9290$, ks2stat = 0.2500. RStest: Conc1: $h = 0$, $p = 0.8201$. KStest2: *Conc2*: $h = 0$, $p = 0.6693$, ks2stat = 0.3333. RStest: Conc2: $h = 0$, $p = 0.8201$. KStest2: Conc3: $h = 0$, $p = 0.3842$, *ks2stat* = 0.4167. RStest: Conc3: $h = 0$, $p = 0.2129$. KStest2: Conc4: $h = 0$, $p = 0.6693$, *ks2stat* = 0.3333. RStest: Conc4: $h = 0$, $p = 0.3355$.

**Male:** Statistical significance was determined by Repeated measures analysis of variance. (Control $N = 11$, Abstinence $N = 6$).*p*-value for Control vs. initial task of male: 0.38794. kstest2 results: $h=1$, $p=7.4300$e−03, ks2stat=0.4091.

Post-hoc analysis: 0.5%: 5.3937e−01, 2%: 3.3053e−01, 5%: 4.3710e−01, 9%: 2.8029e−01. KStest2 and Wilcoxon rank sum test Results (complementary to post-hoc analysis). KStest2: *Conc1*: $h = 0$, $p = 0.4238$, ks2stat = 0.4091. RStest: Conc1: $h = 0$, $p = 0.5908$. KStest2: Conc2: $h = 0$, $p = 0.4238$, ks2stat = 0.4091. RStest: Conc2: $h = 0$, $p = 0.2161$. KStest2: Conc3: $h = 0$, $p = 0.4238$, ks2stat = 0.4091. RStest: Conc3: $h = 0$, $p = 0.5249$. KStest2: Conc4: $h = 0$, $p = 0.1106$, ks2stat = 0.5606. RStest: Conc4: $h = 0$, $p = 0.1215$.

**3-way ANOVA results**

| Source | Sum sq. | d.f. | Singular? | Mean sq. | F | Prob > F |
|---|---|---|---|---|---|---|
| Sex | 581,128.0431 | 1 | 0 | 581,128.0431 | 12.7578 | 0.0005 |
| Condition | 352,607.5709 | 1 | 0 | 352,607.5709 | 7.7409 | 0.0062 |
| Concentration | 4575.5728 | 3 | 0 | 1525.1909 | 0.0335 | 0.9917 |
| Sex*Condition | 6646.8434 | 1 | 0 | 6646.8434 | 0.1459 | 0.7031 |
| Sex*Concentration | 8228.0996 | 3 | 0 | 2742.6999 | 0.0602 | 0.9806 |
| Condition*Concentration | 5712.0473 | 3 | 0 | 1904.0158 | 0.0418 | 0.9886 |
| Error | 5,784,971.7033 | 127 | 0 | 45,550.9583 | NaN | NaN |
| Total | 6,780,383.6044 | 139 | 0 | NaN | NaN | NaN |

**Figure S.6h: Oxycodone I.V. vs. fraction of sigmoid**

Correlation coefficient between amount of oxycodone administered and fraction of sigmoid in all sessions for each animal was determined by MATLAB 'corrcoef' function.

**Figure S.6l: Baseline and oxy early vs. late bins Euclidian distance**

Statistical significance $p = 0.0011$, determined by two-sample Kolmogorov–Smirnov test. (Control $N = 37$, Alcohol $N = 13$).

## Reporting summary

Further information on research design is available in the Nature Portfolio Reporting Summary linked to this article.

## Data availability

Database links go here All data used to produce the figures presented, including raw Excel Ethovision output, PostgreSQL database backups, and CSV data, are provided on the Harvard Dataverse[69]: https://doi.org/10.7910/DVN/QADUKS.

## Code availability

Repo links go here All code made for the RECORD system and related to data analysis has been made available through various GitHub repositories. All depositable RECORD components, including microcontroller firmware, printed circuit board design files, CAD and STL files, Ethovision experiments, Bonsai workflows, Python library, and neuroeconomic analysis code, are available in the main RECORD repository (https://GitHub.com/rjibanezalcala/RECORD). The behavioural data parser along with GUI, and shape fit and clustering analysis code have been made available in the "Databases and Serendipity App repository" (https://GitHub.com/lddavila/UTEP-Brain-Computation-Lab-Remote-Databases-and-Serendipity-App/tree/main). Feature that is used for data analysis are contained in our 'Feature Extraction' repository (https://GitHub.com/atanugiri/Feature-Extraction), and all code made to analyze calcium traces is reported in our calcium trace repository (https://GitHub.com/lrakocev/inscopix). A detailed summary of what each repository contains, along with a direct link to each is reported in Supplementary Table 1.

**Figure generation scripts**

Figures 2a, 2b, 2d, 2e, 2f, 2g, 2h,

5a, 5c, 5d, 5e, 5f, 5g, 5h,

6a, 6b, 6c, 6d, 6e, 6f, 6g, 6h, 6i, 6j, 6k, 6l, 6m, 6n, 6o, 6p, 6q, 6r, 6s,

7b, 7c, 7d, 7e, 7f, 7g, 7h, 7i, 7j,

SF2a, SF2b, SF2c, SF2d, SF2e, SF2f, SF2g, SF2h, SF2i, SF2j, SF2k, SF2l, SF2m, SF2n, SF2o, SF2p, SF2q, SF2r

SF5a, SF5b, SF5c, SF5d, SF5e, SF5f, SF5g, SF5h, SF5i, SF5j, SF5k, SF5l, SF5m,

SF6a, SF6b, SF6c, SF6d, SF6e, SF6f, SF6g, SF6h

can be reproduced with code contained within our Data-Analysis repository (https://GitHub.com/atanugiri/Data-Analysis/tree/main/Data%20Analysis) where instructions and parameters are also reported (https://GitHub.com/atanugiri/Data-Analysis/blob/main/Figure%20generation%20scripts.pdf).

Figure 2i,

5i,

SF5n

which denotes psychometric function dynamics can be reproduced using code contained in https://GitHub.com/WhiteHatArnav/RECORDFiguresCode/blob/main.

Figure 3a, which describes our neuroeconomic analysis, can be reproduced with code contained in https://GitHub.com/rjibanezalcala/RECORD/blob/main/data_analysis/neuroeconomic_analysis/fig6A.m.

Additionally, code for the left panel of Fig. 3b is made available at https://GitHub.com/rjibanezalcala/RECORD/blob/main/data_analysis/neuroeconomic_analysis/fig6B.m.

Figures 3e, 3f, 3g, and 3h were produced by code in https://GitHub.com/lrakocev/inscopix/blob/main.

Figures 4a, 4b, 4c, and 4d,4e,4f were produced by code in https://github.com/lddavila/UTEP-Brain-Computation-Lab-Remote-Databases-and-Serendipity-App/tree/main/Data%20Analysis

Figures: 3c, SF3a, SF3b, SF3c, and SF3d, were produced by the code reported in https://GitHub.com/rjibanezalcala/RECORD/blob/main/data_analysis/neuroeconomic_analysis/OnebyOneAnimalColorMap.m.

Additionally, Fig. 3d was taken from https://GitHub.com/rjibanezalcala/RECORD/blob/main/data_analysis/neuroeconomic_analysis/fig6D.fig.

Figure SF3e, SF3f was produced by https://GitHub.com/rjibanezalcala/RECORD/blob/main/data_analysis/neuroeconomic_analysis/SupFig.m.

Figures 2c, 5b, 6s, SF2g are created from excel files deposited in Harvard Dataverse at the following link, the excel files are in the "Figures Excel Files" directory, which is viewable in the tree view. https://doi.org/10.7910/DVN/QADUKS

Finally, Figs. 4d, 4e, 4f,

5j, 5k, 5l,5m,

6t,6u,6v,

7k,7l,7m,7n,7o,7p,

SF4a, SF4b, SF4c, SF4d,

SF5o, SF5p, SF5q, SF5r,

SF6i, SF6j, SF6k, SF6l,

were produced by the scripts described in supplemental note 5. The GitHub described in the note is located at the following link. (https://github.com/lddavila/UTEP-Brain-Computation-Lab-Remote-Databases-and-Serendipity-App).

### Resource availability
Materials used for the RECORD system are available through various vendors and are reported in Supplementary Table 2.

### Supplementary information
Supplementary notes 1–6 found in Supplementary Information.

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

## Acknowledgements

We greatly appreciate G. Schoenbaum, Y. Shaham, and E. Boyden for their constructive criticism and helpful discussions for improving the system. This project was supported by the NSF-CAREER (2235858), NIH-NIDA (R01DA058653), and NIH (DA045764).

## Author contributions

R.J.I., K.A.G., and A.F. conceptualized the behavioral system. The components were 3D printed by R.J.I., M.M.O., and G.E.M. Electronics were set up by R.J.I. following A.F.'s designs and guidance. The software was developed by R.J.I, D.W.B, L.D.D, A.G, L.I.R, and A.F. Methodologies was developed by K.A.G. and A.F. Thereafter, behavioral data was collected by R.J.I., A.A.S., C.N.H., K.V.R., S.B.H., N.F.R., S.A.B., A.Y.M., P.V., A.J.F., B.J.H., F.Y.R., N.L., A.R., L.L., P.L.C., A.A.A., S.N.V., J.I.A., O.Q., F.M., P.M.O., A.E.M., G.M.N., and R.E.P. Surgery for in vivo imaging was done by A.A.S. and N.F.R. Data was preprocessed and uploaded to our database by R.J.I., L.D.D. A.J.F. Data analysis was done by B.W.D., L.D.D., A.G., C.N.H., L.I.R., S.M.D., K.N., Q.Z., K.A.G., and A.F. Modeling was done by D.W.B., S.M.D., and A.F. Figure creation, layout, and editing was accomplished by C.N.H., S.A.B., P.V., K.A.G., and A.F. The manuscript was written by C.N.H., S.U.B., K.A.G., and A.F. The manuscript was edited by R.J.I., C.N.H., P.V., L.E.O., T.M.M., K.A.G., and A.F.

## Competing interests

The authors declare no competing interests.

## Additional information

[1]Department of Biological Sciences, University of Texas at El Paso, El Paso, TX, USA. [2]Computational Science Program, University of Texas at El Paso, El Paso, TX, USA. [3]Artificial Intelligence Laboratory, Department of Computer Science, Massachusetts Institute of Technology, Cambridge, MA, USA. [4]National Institute on Drug Abuse, Baltimore, MD, USA. [5]Department of Biomedical Informatics, Harvard Medical School, Cambridge, MA, USA. [6]Department of Psychology, University of Texas at El Paso, El Paso, TX, USA. [7]Department of Psychiatry, Center for Translational Medicine and Pharmacology, Friedman Brain Institute, Icahn School of Medicine at Mount Sinai, New York, NY, USA. [8]These authors contributed equally: Raquel J. Ibáñez Alcalá, Dirk W. Beck, Alexis A. Salcido, Luis D. Davila, Atanu Giri, Cory N. Heaton, Kryssia Villarreal Rodriguez, Lara I. Rakocevic, Safa B. Hossain, Neftali F. Reyes, Serina A. Batson, Andrea Y. Macias, Sabrina M. Drammis, Kenichiro Negishi, Qingyang Zhang, Shreeya Umashankar Beck, Paulina Vara. ✉e-mail: Ki.goosens@mssm.edu; Afriedman@utep.edu

