## [Peer Review File · Communications Biology]

Reviewers' comments:

Reviewer #1 (Remarks to the Author):

Ibanez-Alcala et al present a new hardware/software method for quantifying decision making in rodents. It is a potentially powerful system, having many technical and analytical upsides. I commend the authors as well for the thorough documentation on the setup, and suggest that they make a bigger 'deal' out of this resource within the manuscript itself. The custom parsers are also an excellent tool. I think this paper is appropriate for the journal, although some considerable tightening up of the ideas will be required before it is ready for publication. Specific comments are below, but I found both the figures and text to need of more focus and revision. There is plenty of good work here, and for the benefit of clarity, I'd almost prefer fewer of Dr Friedman's many ideas to be shown, but with more clarity and thoughtfulness - especially if other groups are to adopt this potentially very powerful tool. That being said, I am optimistic that the tweaks to the manuscript can be made and can be done without further experiments.

Major:

1- The introduction is a bit all over the place. While I appreciate the very broad view of the opening paragraph in substantiating DM as a field, this is probably not necessary. Then in the 4th paragraph, they abruptly shift what is being discussed without a rationale that I could understand (multiple trade-offs to multiple animals in line 75). Perhaps starting with the objectives laid out in line 90 and then substantiating each would help? I assume that is what the authors are in part aiming to do, but the through-line is difficult to pull out in its current state. Instead, consider instead helping yourself "sell" the tool to the large body of researchers who study DM.

Lastly, some of the generalizations are also a bit over the top. For instance, CPP is very popular DM-task that I cannot recall being used in conjunction with food/water restriction, so the claim that restriction is 'predominant' may be somewhat overstated. In other places, some terms, like 'profound' and 'for the first time' should likely be avoided, even if in the eyes of the author they are correct.

In summary, the authors would benefit from a simplified refocusing of what specifically their system can offer rather than a more general account of deficits in the field which are applicable to varying degrees to the method at hand.

2- As far as I can tell, extended fig 1 a-h is covered in much better detail in the supplemental manual and it is unclear what it adds here. While I appreciate the thoroughness, it made the manuscript / figure harder to parse.

3- 2k has no x axis label and I cannot tell for certain what it should be. My best guess is 2 animals (one trained) over 4 training sessions? if so, put them together on the same plot and label sessions 1-4. if it's 4 animals, please don't connect the points.

In fig 2 e,f,g, the *** significance is all over the place. $p < 0.0001$ can be three or four stars? similarly 3 stars can be $p < 0.001$ or ostensibly $p < 0.001$, aka 0.0008. Please be consistent, especially within a figure. It may help to just say $* < 0.05$, $** < 0.01$, $*** < 0.001$ as it is unclear to me what comparisons

benefit beyond this range. I leave it up to the authors, but it really has to be consistent. If this is a typo, it's just as bad. Extended fig 2 has similar issues. single * in n is undefined and I honestly can't keep track of # vs *. This issue continues in latter figures of the paper and the authors should really have caught/been more focused this, although it can be difficult with such a large team. If there is a better rationale for **, ***, ++, #, etc, please make it clear.

4- Line 139 - Are the behavioral definitions fixed? I assume not, but how difficult is it to change? The standards used here are also a bit confusing. Is a turn of >360 degrees considered two rotations? 3 seconds is long for a stop, even by Noldus standards.

More importantly, it is unclear if a user would get fundamental measures like speed, position, and heading. I'd assume so – especially given extended figure 2 - but it is not clear to me from the text. (also minor, but velocity is used a few times. I believe the authors should use the term speed, which is not a vector like velocity.)

5- Figure 3 has several issues. The legend begins with an explanatory paragraph. Please remove / move to results.

It is unclear why there are only 3 points in 3a 'low sensitivity', when there are 4 in all others.

it is also unclear why there are no error bars on plots with dots on them?

Also, it is never stated what the dashed demarcation is in 3b,c.

The line $y=0.5$ is used sometimes but not others. Please remove.

I have no idea what the lines are in 3e, nor to what particular element the significance start refers to. I'd also have these colors be different than 3f,g to make it more clear that the measure is different (it also seems to have a dark brown streak at $x = 0.96$?). Similarly, I'd say it makes more sense to combine 3f,g – but I understand that it makes decent use of filling up the space.

Many of these comments are also appropriate for Ext Fig 3. Additionally in ext fig 3a, why are there no points beyond $x=2$?

6- While many behaviorists like myself are likely able to code up something similar to the parser or even the hardware/software, several do not have the confidence in modeling or expertise of Dr Friedman. If RECORD is to be an easily utilized, one-stop-shopping tool, is it possible to include the modeling code? Apologies if I missed this, but even so, it's a great opportunity to show off all the work that has been done and increase the chances for wider adoption.

7- Fig 4 – the points in a-c, e are impossible to see. While I appreciate the team approach that is likely given the list of authors, they would be greatly helped to have considerably more top-down consistency, e.g. Fig 3 itself looks like it came from two individuals with vastly different levels of experience, while Fig 4 appears to be from yet another person and lacks the easy-to-read points and fits of 3a or b.

8- I found it exceedingly difficult to understand the radar plot in 4g. Couldn't this be shown on a regular axis and it would relay the same information? It's also quite cramped together as it is now. There is no possible way I can see the difference between rat similar / not similar... they look identical in the key – and also I still don't quite get what it's referring to after having read the text. Finally, are there error bars on these measures? I'm further confused because there's 23 rats and 3

clusters but 4 points here.

It's not clear to me that 4a belongs in the main text, so maybe move this out and give 4g the proper space to elaborate?

9- The ellipses in 4d are not defined. It is also unclear why the major/minor axes must be along x and y, especially when many distributions of points lie on the diagonal.

10- I am so lost in 5h. Why are there %'s on a plot of spatial location? Also, "the rat moves more during food deprivation" was established in 'distance traveled' in the previous panel, right? I guess it's ok to show an individual animal, but the motivation for showing this is left unclear. Also, and potentially instead of this plot, I would be much more interested in RECORD's ability and the actual values for how distance travels changes over TIME. Sessions may be too short, but changes locomotor output has been shown by many to be most pronounced only early in a session. From an experimental design and scientific question standpoint, I would be curious to know if RECORD can capture this readily and what the results are.

This actually brings up a bigger question: does the parser provide a timestamp for each measure, where applicable (e.g. effective speed of bouts starting at 30.5s, 46.7s,)?

Minor

Line 192 – it may be helpful to group the features that are / are not significant, or at least rank order them. Also, while it seems like the authors mostly write out 4 sig figs, this is not consistent, nor necessary. $P = 0.4348$ or 0.233 can just be 0.43 or 0.23 .

Lastly, it would greatly help the reader to state the statistical tests used here rather than the legend – including whether trial/session/or mouse averages were used for the ANOVA, and remove the p values and test from the legends.

Also, for proportions, ANOVA is typically not appropriate. I would recommend a chi squared test for all comparisons of %'s, here and elsewhere (e.g. fig 5).

Line 128. Maybe a subjective comment, but "immense" doesn't tell me much. Technically, there also shouldn't be a comma before 'and'.

Lux is capitalized in the figures and figure legends, but it should not be (as is the case in the text).

It's distance travel~~ED~~ in 4c.

Fig 5 and elsewhere: plots with m/f bars will be easier to read if they are grouped together. This is actually done in ext fig 4d, but I'd suggest going further and have the bars abutting each other.

There is quite a bit of redundant information embedding in the panels. If all plots have 23 rats or 12 oxy sessions(?), just tell us this once – ideally in the legend text.

Similarly, for fig 6 A-L, just put the label for the row off to the side. Using different colors for the lines

may also help.

The y axis in panel 6f does not make any sense. Typo?

In several plots, e.g. 6ghr, ext 5e, the sig marker is floating above an empty space where no measures exist. I do not know what this means. Are the values from the first and second sets of datapoints sig, but not the 3rd and fourth?

Reviewer #2 (Remarks to the Author):

The manuscript “RECORD: A high-throughput system for complex naturalistic decision-making in rodents” by Ibanez-Alcala et al., aims at proposing a new decision-making task in rodents that does not require food or water deprivation. The lack of interesting, ethological decision-making tasks in the literature generated for rodents makes this paper very interesting for the scientific community. Indeed, this new test could be a very interesting tool to study decision-making in rodents of cognitive deficits as well as the potential beneficial effect of a treatment on these deficits. The manuscript is well written, all control experiments are well conducted, and the author even test their new protocol in three different environmental modifications: food deprivation, effect of 14 days of self-administration of oxycodone or after abstinence and sucrose vs alcohol intake. The food deprivation led to enhancing risky decision through an insensitivity of cost, oxycodone self-administration and abstinence led to a desensitization of the reward and the sucrose vs alcohol trade-off enable the study vulnerability to alcohol exposure.

However, some point still needs to be clarified or corrected:

- A part of the literature about inter-individual variability in decision-making in rodent as well as the vulnerability to develop trait like psychiatric disorders in rodents need to be added to the introduction and the discussion (Rivalan et al., 2013; 2009; Pittaras et al., 2022).
- Ambiguity is also an important factor of risky decision-making as, in real life, we are not always aware of the exact consequence of a choice, in the long and short term. RECORD requires a long time of training (9 weeks) and for the high low-cost cost-benefice, it is a real learning over time. The author should discuss this lack of ambiguity in the test and maybe a way to incorporate more ambiguity in a modified version of the combination of high/low-cost cost-benefit task.
- In the entire manuscript, the author used the word “offer”. This led to confusion sometimes in the text like in page 6, line 180. Maybe using the term “reward” would be better.
- The author used several times “related to Figure x” in the legends of the Figure. It is not necessary to say it.
- In general, there are a lot of figures. The Extended Figures are not usual and should be put in supplementary data.
- The radar plots are interesting data to study behavioral change, but they are small, and the lines are similar which makes them very difficult to read. It could be very interesting to improve those graphs.
- In Figure 4g, there are even some data points green/yellow without legend.
- In extended Figure 3, an example of the corner profile as well as the vertical are given but no example of the curve one is shown.

- The word “trial” is not always used the same way in the manuscript (for example Figure 1f and Extended Figure 7a). This leads to confusion and needs to be clarified.
- By looking carefully at the protocols used for the food deprivation, oxycodone, and sucrose vs alcohol trade-off test, we can see that they all differ. The light intensity is different for the oxycodone task, for example, but there is no explanation of why. Or it seems that there are only 3 options in the sucrose vs alcohol trade-off instead on 4. It is important to clarify all parameters of the behavioral tests used to be able to compare them. A table or a detailed Figure explaining the parameters of all tests would be very interesting (duration and number of trials per day and in weeks, what is exactly the high cost, low cost, the different rewards, their associations...). Indeed, if a reader is interested in using RECORD they would want to be sure to understand how the system is working as well as all behavioral experiments in detail.
- In general, the author never explained or discussed what it could mean for a rat to have a decision-making profile described in the manuscript. For example, what does that mean to show a sigmoidal curve for the approach rate depending on the sucrose concentration compared to a parabolic curve or even more compared to an undefined curve? Or what does it mean, in the behavioral point of view, to have no distinct cluster emerging for the U-shaped psychometric function? Hypothesis or propositions on the behavioral explanation of the mathematical observation of the behavior is lacking.
- In Extended Figure 1, j the ITI is 10 sec but, in the text, page 5, line 162 it is 5 sec. This needs to be clarified.
- In Extended Figure 1, j we can only see the low-cost cost-benefit task learning but not the high-cost cost-benefit task learning, why? Is it one week duration for each?
- Figure 2e, the y axe needs to be corrected.
- Page 6, line 180, there is twice the word “more.”
- Some rats were “incapable” of doing the test after self-administration of oxycodone, why?
- There is no Figure 7 and directly Extended Figure 7
- Extended Figure 2: graphs i to l should have the same order as m to p. This would help the comparison.

Response to Reviewers

Reviewer #1

1- The introduction is a bit all over the place. While I appreciate the very broad view of the opening paragraph in substantiating DM as a field, this is probably not necessary. Then in the 4th paragraph, they abruptly shift what is being discussed without a rationale that I could understand (multiple trade-offs to multiple animals in line 75). Perhaps starting with the objectives laid out in line 90 and then substantiating each would help? I assume that is what the authors are in part aiming to do, but the through-line is difficult to pull out in its current state. Instead, consider instead helping yourself "sell" the tool to the large body of researchers who study DM. Lastly, some of the generalizations are also a bit over the top. For instance, CPP is very popular DM-task that I cannot recall being used in conjunction with food/water restriction, so the claim that restriction is 'predominant' may be somewhat overstated. In other places, some terms, like 'profound' and 'for the first time' should likely be avoided, even if in the eyes of the author they are correct. In summary, the authors would benefit from a simplified refocusing of what specifically their system can offer rather than a more general account of deficits in the field which are applicable to varying degrees to the method at hand.

We thank the reviewer for this comment. We have reworked the introduction to be more concise. We shifted the focus of the introduction to highlighting RECORD's features and strengths.

2- As far as I can tell, extended fig 1 a-h is covered in much better detail in the supplemental manual and it is unclear what it adds here. While I appreciate the thoroughness, it made the manuscript / figure harder to parse.

We thank the reviewer for this feedback. To make Supplemental Figure 1 more concise we removed panels a,b,c,f, and h, since they are repetitive. We kept Supplemental Figure 1 panels d,e, and g (now a,b, and c) to have every component of the RECORD system represented in the manuscript figures.

3- 2k has no x axis label and I cannot tell for certain what it should be. My best guess is 2 animals (one trained) over 4 training sessions? if so, put them together on the same plot and label sessions 1-4. if it's 4 animals, please don't connect the points. In fig 2 e,f,g, the * significance is all over the place. $p < 0.0001$ can be three or four stars? similarly 3 stars can be $p < 0.001$ or ostensibly $p < 0.001$, aka 0.0008. Please be consistent, especially within a figure. It may help to just say $* < 0.05$, $** < 0.01$, $*** < 0.001$ as it is unclear to me what comparisons benefit beyond this range. I leave it up to the authors, but it really has to be consistent. If this is a typo, it's just as bad. Extended fig 2 has similar issues. single * in n is undefined and I honestly can't keep track of # vs *. This issue continues in latter figures of the paper and the authors should really have caught/been more focused this, although it can be difficult with such a large team. If there is a better rationale for **, ***, ++, #, etc, please make it clear.**

We thank the reviewer for these observations. Extended Figure 2k, now Supplemental Figure 2f, is comparing a rat who has performed the task and learned to approach higher sucrose concentrations to a rat who has recently started training and may not have learned to associate the quadrants to sucrose reward. We added the x-axis (sucrose %) to the figure panel. We have also cleaned up and ensured that all p-values in the legends, main text and figures are now consistent. The different symbols represent different significant effects, with * meaning significant

effect of concentration, # being used for significant effect of sex, and + signifying a significant effect of conditions (food deprivation, oxycodone, and alcohol). To clarify the meaning of these symbols, we added text to describe this in the figure legend of the figure where the symbol is first used.

4- Line 139 - Are the behavioral definitions fixed? I assume not, but how difficult is it to change? The standards used here are also a bit confusing. Is a turn of >360 degrees considered two rotations? 3 seconds is long for a stop, even by Noldus standards. More importantly, it is unclear if a user would get fundamental measures like speed, position, and heading. I'd assume so – especially given extended figure 2 - but it is not clear to me from the text. (also minor, but velocity is used a few times. I believe the authors should use the term speed, which is not a vector like velocity.)

We thank the reviewer for these important questions. The behavioral definitions used throughout the manuscript are not fixed and can be changed in the Matlab codes accessible through a GitHub linked in methods: 'Spatiotemporal behavioral dynamics'. Fundamental measures like speed, position and heading are extracted and used to calculate the "behavioral features" used throughout the manuscript. To make this clear in the manuscript, we have added a sentence into methods: 'Spatiotemporal behavioral dynamics' in line 975.

5- Figure 3 has several issues. The legend begins with an explanatory paragraph. Please remove / move to results. It is unclear why there are only 3 points in 3a 'low sensitivity', when there are 4 in all others. it is also unclear why there are no error bars on plots with dots on them? Also, it is never stated what the dashed demarcation is in 3b,c. The line $y=0.5$ is used sometimes but not others. Please remove. I have no idea what the lines are in 3e, nor to what particular element the significance start refers to. I'd also have these colors be different than 3f,g to make it more clear that the measure is different (it also seems to have a dark brown streak at $x = 0.96?$). Similarly, I'd say it makes more sense to combine 3f,g – but I understand that it makes decent use of filling up the space. Many of these comments are also appropriate for Ext Fig 3. Additionally in ext fig 3a, why are there no points beyond $x=2$?

We thank the reviewer for these points. We moved the explanatory paragraph to introduce the neuroeconomic modeling methods. We added additional clarification in the figure legend. We also defined what the dashed demarcations mean in the figure legends of Figure 3b, line 3497. The dark streak of Figure 3e seems to have been an exporting error and is not visible in the original figure. We tried different exporting methods to find the one with the truest representations of the colors. We removed the points from Supplemental Figure 3a; they were the raw data points used to fit the model and when all the points were present, it impeded the clarity of the function along with the different shades being used to delineate different costs/rewards.

6- While many behaviorists like myself are likely able to code up something similar to the parser or even the hardware/software, several do not have the confidence in modeling or expertise of Dr Friedman. If RECORD is to be an easily utilized, one-stop-shopping tool, is it possible to include the modeling code? Apologies if I missed this, but even so, it's a great opportunity to show off all the work that has been done and increase the chances for wider adoption.

We thank the reviewer for this question. All RECORD tools will be readily available. All software, including modeling, analysis, and interfacing tools are available in GitHub and are linked in the supplemental notes. We mention this availability in the manuscript in line 134, line 200, and in

code availability section starting with line 3102. Also, in Supplemental table 2 we provide all the links to all the codes that were developed for the project.

7- Fig 4 – the points in a-c, e are impossible to see. While I appreciate the team approach that is likely given the list of authors, they would be greatly helped to have considerably more top-down consistency, e.g. Fig 3 itself looks like it came from two individuals with vastly different levels of experience, while Fig 4 appears to be from yet another person and lacks the easy-to-read points and fits of 3a or b.

We thank the reviewer for these observations. We made the points in Figure 4 more apparent, along with other minor aesthetic changes to some panels to improve the visual clarity of the figure.

8- I found it exceedingly difficult to understand the radar plot in 4g. Couldn't this be shown on a regular axis and it would relay the same information? It's also quite cramped together as it is now. There is no possible way I can see the difference between rat similar / not similar... they look identical in the key – and also I still don't quite get what it's referring to after having read the text. Finally, are there error bars on these measures? I'm further confused because there's 23 rats and 3 clusters but 4 points here. It's not clear to me that 4a belongs in the main text, so maybe move this out and give 4g the proper space to elaborate?

We thank the reviewer for these observations. To address this, we have removed figure 4g and have added examples of radar plots across the different conditions. These new radar plots show shifts across behavioral clusters between baseline and food deprivation/oxycodone/alcohol for individual rats. Since the new radar plots only depict two lines the differences should be clearer, and the plots are less cramped.

9- The ellipses in 4d are not defined. It is also unclear why the major/minor axes must be along x and y, especially when many distributions of points lie on the diagonal.

We thank the reviewer for this important comment. We removed the ellipses from Figure 4d; they denoted the center of each cluster. In Figures 5 and 6, we reworked the cluster plots to have the ellipses represent one standard deviation away from the center of the cluster. An explanation for this was added into the figure legend of Figure 5 line 3570.

10- I am so lost in 5h. Why are there %'s on a plot of spatial location? Also, "the rat moves more during food deprivation" was established in 'distance traveled' in the previous panel, right? I guess it's ok to show an individual animal, but the motivation for showing this is left unclear. Also, and potentially instead of this plot, I would be much more interested in RECORD's ability and the actual values for how distance travels changes over TIME. Sessions may be too short, but changes locomotor output has been shown by many to be most pronounced only early in a session. From an experimental design and scientific question standpoint, I would be curious to know if RECORD can capture this readily and what the results are. This actually brings up a bigger question: does the parser provide a timestamp for each measure, where applicable (e.g. effective speed of bouts starting at 30.5s, 46.7s,)?

We thank the reviewer for this important inquiry. The percentages on the plot are denoting where the different reward values were administered. RECORD can capture specific time frames for each measure. To demonstrate this, we added new panels to Supplemental Figure 2 (Supplemental Fig. 2q,r, line 3755) to show spikes in acceleration across a twenty second time

interval which were then mapped to show the spatial location of the rodent in the maze at the time of the spike in line 186.

Line 192 – it may be helpful to group the features that are / are not significant, or at least rank order them. Also, while it seems like the authors mostly write out 4 sig figs, this is not consistent, nor necessary. $P = 0.4348$ or 0.233 can just be 0.43 or 0.23 . Lastly, it would greatly help the reader to state the statistical tests used here rather than the legend – including whether trial/session/or mouse averages were used for the ANOVA, and remove the p values and test from the legends. Also, for proportions, ANOVA is typically not appropriate. I would recommend a chi squared test for all comparisons of %'s, here and elsewhere (e.g. fig 5).

We thank the reviewer for these suggestions. We grouped features to be aligned throughout the manuscript since it would allow for easier condition between conditions, especially those that have two sets of plots, and as was requested by reviewer 2 in his last comment.

Line 128. Maybe a subjective comment, but “immense” doesn’t tell me much. Technically, there also shouldn’t be a comma before ‘and’.

We thank the reviewer for these observations. We removed the word immense and removed the incorrect comma in what is now line 124.

Lux is capitalized in the figures and figure legends, but it should not be (as is the case in the text).

We thank the reviewer for this observation. We addressed this in the figures and figure legends.

It's distance traveled in 4c.

We thank the reviewer for this comment. We corrected the figure to be Distance traveled.

Fig 5 and elsewhere: plots with m/f bars will be easier to read if they are grouped together. This is actually done in ext fig 4d, but I'd suggest going further and have the bars abutting each other.

We thank the reviewer for this suggestion. We abutted bar plots in many instances and otherwise tried to keep male and female plots grouped close together.

There is quite a bit of redundant information embedding in the panels. If all plots have 23 rats or 12 oxy sessions(?), just tell us this once – ideally in the legend text.

We thank the reviewer for this observation. We have lessened the information embedded in the panels unless the information changes for the next panel/sets of panels.

Similarly, for fig 6 A-L, just put the label for the row off to the side. Using different colors for the lines may also help.

We thank the reviewer for this advice. We shifted the titles to be on the leftmost panel of the row and removed the redundant titles. We also changed the color of the plots to keep baseline consistent with condition having separate colors.

The y axis in panel 6f does not make any sense. Typo?

We thank the reviewer for this question, we have ensured that the y axis is labeled, "Number of stopping points" and will ensure that no errors arise during exporting the figures.

In several plots, e.g. 6ghr, ext 5e, the sig marker is floating above an empty space where no measures exist. I do not know what this means. Are the values from the first and second sets of datapoints sig, but not the 3rd and fourth?

We thank the reviewer for this question. We added lines adjacent to each significance marker to make it clear which main factor is associated with the marker (for example, group or sucrose concentration).

Reviewer #2

1- A part of the literature about inter-individual variability in decision-making in rodent as well as the vulnerability to develop trait like psychiatric disorders in rodents need to be added to the introduction and the discussion (Rivalan et al., 2013; 2009; Pittaras et al., 2022).

We thank the reviewer for bringing our attention to this important literature. We have added a brief description of rodent versions of the Iowa gambling task into the introduction which has been reworked in line 64 and discuss the collection of individual traits being linked with variable decision-making in rodents within the discussion line 444.

2- Ambiguity is also an important factor of risky decision-making as, in real life, we are not always aware of the exact consequence of a choice, in the long and short term. RECORD requires a long time of training (9 weeks) and for the high low-cost cost-benefice, it is a real learning over time. The author should discuss this lack of ambiguity in the test and maybe a way to incorporate more ambiguity in a modified version of the combination of high/low-cost cost-benefit task.

We would like to thank the reviewer for this important question as addressing it highlights the versatility of the RECORD system and how it can be potentially modified to explore many different types of decision-making. Starting in line 478, we discuss three different ways that ambiguity could be introduced into the task environment with relative ease.

3- In the entire manuscript, the author used the word “offer”. This led to confusion sometimes in the text like in page 6, line 180. Maybe using the term “reward” would be better.

We thank the reviewer for this suggestion. We switched offer to terms such as reward/cost combinations, trade-offs, or reward, depending on the context, to improve clarity.

4- The author used several times “related to Figure x” in the legends of the Figure. It is not necessary to say it.

We thank the reviewer for this observation. “Related to Figure x” phrases were removed.

5- In general, there are a lot of figures. The Extended Figures are not usual and should be put in supplementary data.

We thank the reviewer for this comment. We have moved all extended figures to supplementary data.

6- The radar plots are interesting data to study behavioral change, but they are small, and the lines are similar which makes them very difficult to read. It could be very interesting to improve those graphs.

We thank the reviewer for this observation. To address this, we provided more radar plots comparing individual rats to themselves after manipulating the condition. This allows radar plots to clearly depict differences in behavioral cluster preference while not being overly crowded and hard to see.

7- In Figure 4g, there are even some data points green/yellow without legend.

We thank the reviewer for bringing up this important point. We removed Figure 4g altogether and instead added radar plots depicting examples of behavioral cluster shifts between an individual rat's performance at baseline and after a manipulation. These plots have two distinct lines which makes the difference more apparent.

8- In extended Figure 3, an example of the corner profile as well as the vertical are given but no example of the curve one is shown.

We thank the reviewer for this point. We added an example of a curve profile in Supplemental Figure 3d.

9- The word "trial" is not always used the same way in the manuscript (for example Figure 1f and Extended Figure 7a). This leads to confusion and needs to be clarified.

We thank the reviewer for this important point. We changed the figure panel for Extended Figure 7a, now Figure 7a, to remove the word trial altogether. Figure 7a is only meant to show the four different options available for a rat to approach during an alcohol task behavioral session. Each trial will offer one of the four sucrose-alcohol combinations with a light intensity. We have also expanded this idea in the figure legends line 3639.

10- By looking carefully at the protocols used for the food deprivation, oxycodone, and sucrose vs alcohol trade-off test, we can see that they all differ. The light intensity is different for the oxycodone task, for example, but there is no explanation of why. Or it seems that there are only 3 options in the sucrose vs alcohol trade-off instead on 4. It is important to clarify all parameters of the behavioral tests used to be able to compare them. A table or a detailed Figure explaining the parameters of all tests would be very interesting (duration and number of trials per day and in weeks, what is exactly the high cost, low cost, the different rewards, their associations...). Indeed, if a reader is interested in using RECORD they would want to be sure to understand how the system is working as well as all behavioral experiments in detail.

We thank the reviewer for these suggestions. We expanded upon the behavioral methods section to include more details and clarify the specific tasks used. For alcohol, four options were presented, and this has been made clearer in line 402 and in methods: 'Sucrose vs. alcohol trade-off'. For oxycodone, we added a section to the methods section discussing the hypersensitivity to cost after oxycodone. It is for this reason that we found it necessary to reduce the light intensities used. Overall, we also specified that the high-cost task included 320 lux light intensities while the low-cost task only presented solutions with 15 lux light intensities. This has all been expanded upon within the behavioral methods sections.

11- In general, the author never explained or discussed what it could mean for a rat to have a decision-making profile described in the manuscript. For example, what does that mean to show a sigmoidal curve for the approach rate depending on the sucrose concentration compared to a parabolic curve or even more compared to an undefined curve? Or what does it mean, in the behavioral point of view, to have no distinct cluster emerging for the U-shaped psychometric function? Hypothesis or propositions on the behavioral explanation of the mathematical observation of the behavior is lacking.

We thank the reviewer for these important observations. More discussion about potential behavioral outcomes from modeling ideas was added starting line 219 and is expanded upon throughout the paragraph.

- In Extended Figure 1, j the ITI is 10 sec but, in the text, page 5, line 162 it is 5 sec. This needs to be clarified.

We thank the reviewer for these important observations. We clarified this in the resubmitted manuscript. We have changed the values for the example trial to accurately depict a general timeframe that a trial occurs in. There are some variances depending on system delay and we explain this variance in the RECORD user manual provided in supplementary information along with including a note about it in the related figure legend for extended figure 1j/nw Supplemental Figure 1e.

- In Extended Figure 1, j we can only see the low-cost cost-benefit task learning but not the high-cost cost-benefit task learning, why? Is it one week duration for each?

We thank the reviewer for this important comment. We habituated the rats to handlers for about 2 weeks, then taught the rats where rewards and costs were located in the maze with the reward/cost association task which also usually took about 2 weeks, then rats typically spent 2 weeks learning the low-cost cost-benefit task, and finally rats were trained on both high-cost and low-cost cost-benefit tasks for 3 weeks thus leading to approximately nine weeks of task training. After psychometric functions are established, we gradually introduce a high cost. We are running high cost-benefit tasks 2 times a week and low-cost benefit 3 times a week. Rats display stable performance 4 weeks after we start the task. A better explanation for this has been added in Methods: 'Adaptable decision-making task batteries'.

- Figure 2e, the y axe needs to be corrected.

We thank the reviewer for the observation we corrected y axes within the original document. Some axes errors seem to have arisen from PDF exporting errors. We changed the export settings to minimize visual errors.

- Page 6, line 180, there is twice the word "more."

We thank the reviewer for the observation. We have corrected the error in the text.

- Some rats were "incapable" of doing the test after self-administration of oxycodone, why?

We thank the reviewer for this question because it is important for understanding why the population sizes fluctuate across tasks. The rats who did not participate in behavioral sessions tried to bite the experimenters as they were removed from the oxycodone self-administration chambers and would also chew and bite the LEDs and floor ports within the RECORD system. Due to the excessive difficulty in handling these rats, they were not run in DM behavioral sessions. This explanation was added into methods 'Oxycodone behavioral task'.

- There is no Figure 7 and directly Extended Figure 7

We thank the reviewer for this observation. We made Extended Figure 7 into Figure 7.

- Extended Figure 2: graphs i to l should have the same order as m to p. This would help the comparison.

We thank the reviewer for this suggestion and rearranged the panels.

Reviewers' comments:

Reviewer #1 (Remarks to the Author):

Considerable work, particularly on the hodgepodge of figures originally, has been done. issues are addressed. my comments are fairly aesthetic and would not require further follow up for my part. This may be a powerful behavioural tool for several groups.

The introduction still lacks direction but more importantly, we don't get a sense of what the problem or need is. I'd ask the authors to put something in the first paragraph that sets up the statement in line 90. Even something akin to starting with 'Decision-making is a fundamental responsibility of the brain. Although several methods exist to explore particular aspects of these computations, their limitations prevent a more comprehensive view of how decisions are made". Similarly, line 26 (abstract) leaves something to be desired.

subjective, but for readability, the authors might consider replacing DM with decision making (or decision-making, as is also in vogue) line 488. I believe it is written B-SO_iD, rather than B-SoiD. The recently published A-SO_iD may be a good addition as well. Similarly, in the following line, I believe LEAP has been discontinued in favour of SLEAP, even for single animals.

spacing between the formulae in fig 3 could be increased. it currently reads as if it were all one long formula

in fig 7, panel A is not required. Moreover, the plots might be more readable with the initial and late panels aligned, which would be possible with panel a's omission

Reviewer #2 (Remarks to the Author):

The manuscript "RECORD: A high-throughput system for complex naturalistic decision-making in rodents" by Ibanez-Alcala et al., aims at proposing a new decision-making task in rodents that does not require food or water deprivation.

They did answer all my suggestions, point by point, and did improve the manuscript.

In my opinion, the manuscript seems to be ready for publication except a few minors' points:

- The supplementary Table 1 is missing all the literature about the adaptation of the IGT in rodents,
- What does the * mean in this table?
- The difference between male and female are never discussed or compared to previous publications.
- The Fig 2, o, does not need the legend again.
- Figure 2i seems a bit confusing. It seems that some mice (top right) would prefer to approach even more when the light intensity is higher than if it is lower at a 9% cc of sucrose?
- In Fig 5k, the meaning of the arrows is missing.

Response to Reviewers

Reviewer #1 (Remarks to the Author):

1- The introduction still lacks direction but more importantly, we don't get a sense of what the problem or need is. I'd ask the authors to put something in the first paragraph that sets up the statement in line 90. Even something akin to starting with 'Decision-making is a fundamental responsibility of the brain. Although several methods exist to explore particular aspects of these computations, their limitations prevent a more comprehensive view of how decisions are made'.

We thank the reviewer for this comment. We worked on rephrasing certain sections of the introduction to make it clearer where we believe RECORD solves a problem, rather than simply listing the problems.

2- Similarly, line 26 (abstract) leaves something to be desired.

We thank the reviewer for their thorough overview of our abstract. We acknowledge the opportunity to include detail in line 26. We made some minor changes to focus on RECORD goals and expanded on key points.

3- subjective, but for readability, the authors might consider replacing DM with decision making (or decision-making, as is also in vogue)

Thank you for this comment we have replaced DM with decision-making.

4- line 488. I believe it is written B-SOiD, rather than B-SoiD. The recently published A-SOiD may be a good addition as well.

We thank the reviewer for catching this error. We have corrected the spelling in line 488 and added A-SOiD for readers interested in leveraging these tools.

5- Similarly, in the following line, I believe LEAP has been discontinued in favor of SLEAP, even for single animals.

We thank the reviewer for their comment. We have removed LEAP from the list.

6- spacing between the formulae in fig 3 could be increased. it currently reads as if it were all one long formula.

We thank the reviewer for the suggestion. We have incorporated the needed space to enhance readability and general understanding.

7- in fig 7, panel A is not required. Moreover, the plots might be more readable with the initial and late panels aligned, which would be possible with panel a's omission

We thank the reviewer for considering the necessity of panel A in Main Figure 7. For a general audience, we feel that panel A may be helpful for a basic understanding of the task if a reader is skimming the manuscript or only looking through the figures. Each pair of initial/late panels is side-by-side in our revised figure.

Reviewer #2 (Remarks to the Author):

1- The supplementary Table 1 is missing all the literature about the adaptation of the IGT in rodents.

We thank the reviewer for bringing this to our attention. We have addressed this by adding rodent IGT to our supplementary table.

2- What does the * mean in this table?

We thank the reviewer for their commitment to the readability of our paper. We have included a legend under the table which defines “ * “ within the supplementary table as indicating important cases that are not representative of the average task implementation.

3- The difference between male and female are never discussed or compared to previous publications.

We thank the reader for this comment. We addressed differences between males and females within our own task, however, we have included additional commentary concerning sex differences in previous publications. **Please see lines 342 and 449**, where we have incorporated this feedback.

4- The Fig 2, o, does not need the legend again.

We thank the reviewer for their attention to detail. We have removed the repetitive legend but left the symbol reflecting significance.

5- Figure 2i seems a bit confusing. It seems that some mice (top right) would prefer to approach even more when the light intensity is higher than if it is lower at a 9% cc of sucrose?

We thank the reviewer for this important idea as it addresses the versatility of RECORD's capabilities to parse individual differences. Like humans, rodents have preference thresholds that will impact their decision-making. The top right of Figure 2i represents a rodent whose preference is to approach despite high light levels/costs. In this instance, this rodent is not representative of rodents on average but instead seeks to represent a light resilient individual. We have added a brief explanation in **line 195**.

6- In Fig 5k, the meaning of the arrows is missing.

We thank the reviewer for this observation. We have included the meaning of the arrow, highlighting greatest differences in cluster participation, within the legend. This applies to all radar plots with arrows.